# A Provably Robust Algorithm for Differentially Private Clustered Federated Learning

## Abstract

Federated Learning (FL), which is a decentralized machine learning (ML) approach, often incorporates differential privacy (DP) to enhance data privacy guarantees. However, differentially private federated learning (DPFL) introduces performance disparities across clients, particularly affecting minority groups. Some recent works have attempted to address large data heterogeneity in vanilla FL settings through clustering clients, but these methods remain sensitive and prone to errors further exacerbated by the DP noise, making them inappropriate for DPFL settings. We propose an algorithm for differentially private clustered FL, which is robust to the DP noise in the system and identifies clients' clusters correctly. To this end, we propose to cluster clients based on both their model updates and training loss values. Furthermore, when clustering clients' model updates, our proposed approach addresses the server's uncertainties by employing large batch sizes as well as Gaussian Mixture Models (GMM) to reduce the impact of DP and stochastic noise and avoid potential clustering errors. This idea is efficient especially in privacy-sensitive scenarios with more DP noise. We provide theoretical analysis justifying our approach, and evaluate it extensively across diverse data distributions and privacy budgets. Our experimental results show its effectiveness in addressing large data heterogeneity in DPFL systems with a small computational cost.

## 1 Introduction

Federated learning (FL) (McMahan et al., 2017) is a collaborative ML paradigm, which allows multiple clients to train a shared global model without sharing their data. However, in order for FL algorithms to ensure rigorous privacy guarantees against data privacy attacks (Hitaj et al., 2017; Rigaki & García, 2020; Wang et al., 2019; Zhu et al., 2019; Geiping et al., 2020), they are reinforced with DP (Dwork et al., 2006b;a; Dwork, 2011; Dwork & Roth, 2014). This is done in the presence of a trusted server (McMahan et al., 2018; Geyer et al., 2017) and in its absence (Zhao et al., 2020; Duchi et al., 2013; 2018). In the latter case and for record-level DP, each client adds noise to its stochastic gradients locally and shares its noisy model update with the server at the end of each round.

A key challenge in FL settings is ensuring a similar performance across clients under heterogeneous data distributions, where several existing works focus on accuracy parity across clients with a single common model (Mohri et al., 2019; Michieli & Ozay, 2021). However, a single global model often fails to adapt to the data heterogeneity across clients (Chu et al., 2023), *especially with extreme covariate and label shifts*. To address this, multiple methods were proposed to achieve performance parity *in non-DP FL settings*: agnostic federated learning (Mohri et al., 2019), client reweighting (Li et al., 2020b;a; Zhang et al., 2023), multi-task learning (Smith et al., 2017; Li et al., 2021; Marfoq et al., 2021; Wu et al., 2023), transfer learning (Li & Wang, 2019; Liu et al., 2020) and clustered FL (Ghosh et al., 2020; Mansour et al., 2020; Ruan & Joe-Wong, 2021; Sattler et al., 2019; Werner et al., 2023; Briggs et al., 2020), where the latter is the focus of this work. On the other hand, when augmenting FL with DP for getting rigorous privacy guarantees, DP can have disparate impacts on the accuracy of different subgroups of clients - even with small imbalances and loose privacy guarantees (Farrand et al., 2020; Fioretto et al., 2022; Bagdasaryan & Shmatikov, 2019). In fact, groups with minority data experience a larger drop in model utility (larger privacy cost). Being due to the inequitable gradient clipping in DPSGD (Abadi et al., 2016; Bagdasaryan & Shmatikov, 2019; Xu et al., 2021; Esipova et al., 2022), this behavior has become increasingly important to be addressed.

As mentioned, clustered FL was proposed as an efficient personalization technique in vanilla FL for performance parity under extreme data heterogeneity across clusters of clients: subsets of clients are grouped together by the server based on their loss values (Ghosh et al., 2020; Mansour et al., 2020; Ruan & Joe-Wong, 2021; Chu et al., 2023; Liu et al., 2022) or their gradients (model updates) (Sattler et al., 2019; Werner et al., 2023; Briggs et al., 2020). As discussed in (Werner et al., 2023) in details, the aforementioned two categories of clustered FL approaches are vulnerable to errors in clustering due to their sensitivity to: 1. model initialization 2. randomness in clients' model updates due to stochastic noise. DP noise exacerbates this vulnerability, especially in the first few rounds of FL training. To address this, we propose a clustered DPFL algorithm which uses both clients' model updates and losses values to cluster them, making it more robust to DP/stochastic noise.

A correct clustering of clients results in equity of privacy cost between the client groups (Esipova et al., 2022; Tran et al., 2020). Justified by our theoretical analysis, our proposed algorithm uses a full batch size in the first round to reduce the noise in clients' model updates at the end of this round. Then, the server soft clusters clients based on these less noisy model updates using a Gaussian Mixture Model (GMM). Depending on the "confidence" of the learned GMM, the server keeps using it to soft cluster clients during the next few rounds. Finally, the server switches the clustering strategy to *local* clustering of clients based on their loss values in the remaining rounds. These altogether make our method effective and robust. The highlights of our contributions are as follows:

- We propose a DP clustered FL algorithm, which combines information from both clients' model updates and their loss values. The algorithm is robust and achieves high-quality clustering of clients, even in the presence of DP noise in the system.
- We theoretically prove that increasing clients' batch sizes, particularly in the initial communication round, consistently improves the server's ability to cluster clients based on their model updates at the end of the first round.
- We theoretically prove that using sufficiently large client batch sizes in the first round, enables super-linear convergence rate for learning a GMM on clients' model updates, which leads to fast and accurate clustering of clients with low computational overhead.
- Extensive evaluation across diverse and heterogeneous datasets and scenarios demonstrates the effectiveness of our robust clustered DPFL (RC-DPFL) algorithm in detecting the clustering structure of clients, which leads to a utility improvement for minority clusters.

## 2 RELATED WORK

**Performance parity in FL**: Performance parity of the final trained model across clients is an important goal in FL. Addressing this goal, Mohri et al. (2019) proposed Agnostic FL (AFL) by using a min-max optimization approach. TERM (Li et al., 2020a) used tilted losses to up-weight clients with large losses. Finally, Li et al. (2020b) and Zhang et al. (2023) proposed $q$-FFL and PropFair, inspired by $\alpha$-fairness (Lan et al., 2010) and proportional fairness (Bertsimas et al., 2011), respectively. Generating one common model for all clients, these techniques do not perform well when the data distribution across clients is highly heterogeneous, leading to low overall performance in the system. This leads us to use stronger personalization techniques, e.g., client clustering.

**Clustered FL:** Clustered FL has been originally proposed for personalization *in vanilla non-DP FL with highly heterogeneous data*, where clients can be naturally partitioned into clusters. Existing clustered FL algorithms cluster clients based on their loss values (Mansour et al., 2020; Ghosh et al., 2020; Ruan & Joe-Wong, 2021) or their model updates (based on e.g., their euclidean distance (Werner et al., 2023; Briggs et al., 2020) or cosine similarity (Sattler et al., 2019)). As studied in (Werner et al., 2023), the algorithms are prone to clustering errors in the early rounds of FL training (due to gradient stochasticity, model initialization or the form of loss functions far from their optima), which can even propagate in the subsequent rounds. This vulnerability is exacerbated in DPFL systems, due to the extra DP noise. Without addressing this vulnerability, Luo et al. (2024) proposed a clustered DPFL algorithm with a limited applicability, which clusters clients based on the labels that they do not have in their local data, and is inapplicable when clients have all possible labels.

**Differential privacy, group fairness and performance parity:** Gradient clipping and random noise addition used in DPSGD disproportionately affect underrepresented groups. Some works tried to address the tension between group fairness and DP in centralized settings (Tran et al., 2020)

(by using Lagrangian duality) and `FL` settings (Pentyala et al., 2022) (by using Secure Multiparty Computation (`MPC`)). Another work tried to remove the disparate impact of `DP` on model performance of minority groups in centralized settings (Esipova et al., 2022), by preventing gradient misalignment across different groups of data. Unlike the previous works on group fairness, this work adopts cross-model fairness, where the performance cost of adding privacy to a non-private model must be fairly distributed between different groups. We adopt the same notion - which is also used in (Chu et al., 2023). Considering a highly heterogeneous data split, the mentioned approaches are not appropriate due to generating one single model for all groups. In contrast, we propose a robust "clustered" `DPFL` algorithm, which identifies different groups of clients and learns a model for each.

## 3    DEFINITIONS, NOTATIONS AND ASSUMPTIONS

There are multiple definitions of `DP`. We adopt the following definition in this work:

**Definition 3.1** (($\epsilon, \delta$)-`DP` (Dwork et al., 2006a))**.** *A randomized mechanism $\mathcal{M} : \mathcal{D} \to \mathcal{R}$ with domain $\mathcal{D}$ and range $\mathcal{R}$ satisfies ($\epsilon, \delta$)-`DP` if for any two adjacent inputs $d, d' \in \mathcal{D}$, which differ only by a single record (by removal), and for any measurable subset of outputs $\mathcal{S} \subseteq \mathcal{R}$ it holds that*

$$Pr[\mathcal{M}(d) \in \mathcal{S}] \le e^\epsilon Pr[\mathcal{M}(d') \in \mathcal{S}] + \delta.$$

Gaussian mechanism randomizes the output of a query $f$ as $\mathcal{M}(d) \triangleq f(d) + \mathcal{N}(0, \sigma^2)$. The randomized output of the Gaussian mechanism satisfies ($\epsilon, \delta$)-`DP` for a continuum of pairs ($\epsilon, \delta$): it is ($\epsilon, \delta$)-`DP` for all $\epsilon < 1$ and $\sigma > \frac{\sqrt{2 \ln(1.25/\delta)}}{\epsilon} \Delta_2 f$, where $\Delta_2 f \triangleq \max_{d,d'} \| f(d) - f(d') \|_2$ is the $l_2$-sensitivity of the query $f$ with respect to its input dataset. Also, the $\epsilon$ and $\delta$ privacy parameters resulting from running Gaussian mechanism depend on the quantity $z = \frac{\sigma}{\Delta_2 f}$ (called "noise scale"). We consider a `DPFL` system (see Figure 1, left), where there are $n$ clients with the same desired privacy parameters ($\epsilon, \delta$), and each runs `DPSGD`. In the context of Definition 3.1, we consider record-level ($\epsilon, \delta$)-`DP` for every client $i$: the set of model updates sent by client $i$ to the server satisfies ($\epsilon, \delta$)-`DP` (Definition 3.1) for all adjacent datasets $\mathcal{D}_i$ and $\mathcal{D}'_i$ differing in one record (by removal).

Let $x \in \mathcal{X} \subseteq \mathbb{R}^d$ and $y \in \mathcal{Y} = \{1, \dots, C\}$ denote an input data point and its target label. Client $i$ holds dataset $\mathcal{D}_i$ with $N_i$ samples from distribution $P_i(x, y) = P_i(y|x)P_i(x)$. Let $h : \mathcal{X} \times \boldsymbol{\theta} \to \mathbb{R}^C$ be the predictor function, which is parameterized by $\boldsymbol{\theta} \in \mathbb{R}^p$. Also, let $\ell : \mathbb{R}^C \times \mathcal{Y} \to \mathbb{R}_+$ be the loss function used (cross-entropy loss). Client $i$ in the system has empirical train loss $f_i(\boldsymbol{\theta}) = \frac{1}{N_i} \sum_{(x,y) \in \mathcal{D}_i} [\ell(h(x, \boldsymbol{\theta}), y)]$, with minimum value $f_i^*$. There are $E$ communication rounds indexed by $e$. During each round $e$, client $i$ runs $K$ local epochs with learning rate $\eta_l$. There are $M$ clusters of clients indexed by $m$, and the server holds $M$ cluster models $\{\boldsymbol{\theta}_m^e\}_{m=1}^M$ for them at the beginning of round $e$. Clients $i$ and $j$ belonging to the same cluster have the same data distributions, while there is high data heterogeneity across clusters. $s(i)$ denotes the true cluster of client $i$ and $R^e(i)$ denotes the cluster assigned to it at the beginning of round $e$. Let's assume the batch size that client $i$ uses in the first round $e = 1$ is $b_i^1$, which may be different from the batch size $b_i^{>1}$ that it uses in the rest of the rounds $e > 1$. At the $t$-th gradient update during the round $e$, and given a current model $\boldsymbol{\theta}$, client $i$ uses batch $\mathcal{B}_i^{e,t}$ with size $b_i^e$, and computes the following `DP` noisy batch gradient:

$$\tilde{g}_i^{e,t}(\boldsymbol{\theta}) = \frac{1}{b_i^e} \left[ \Big( \sum_{j \in \mathcal{B}_i^{e,t}} \bar{g}_{ij}(\boldsymbol{\theta}) \Big) + \mathcal{N}(0, \sigma_{i,\text{DP}}^2 \mathbb{I}_p) \right], \tag{1}$$

where $\bar{g}_{ij}(\boldsymbol{\theta}) = \texttt{clip}(\nabla \ell(h(x_{ij}, \boldsymbol{\theta}), y_{ij}), c)$, and $c$ is a clipping threshold: for a given vector $\mathbf{v}$, $\texttt{clip}(\mathbf{v}, c) = \min\{\|\mathbf{v}\|, c\} \cdot \frac{\mathbf{v}}{\|\mathbf{v}\|}$. Also, $\mathcal{N}$ is the Gaussian noise distribution with variance $\sigma_{i,\text{DP}}^2$, where $\sigma_{i,\text{DP}} = c \cdot z_i(\epsilon, \delta, b_i^1, b_i^{>1}, N_i, K, E)$. $z_i$ is the noise scale needed for achieving ($\epsilon, \delta$)$-$`DP` by client $i$, which can be determined with a privacy accountant, e.g., the Renyi-DP accountant (Mironov et al., 2019) used in this work, which is capable of accounting composition of *heterogeneous* `DP` mechanisms (Mironov, 2017). The privacy parameter $\delta$ is fixed to $10^{-4}$ in this work. For an arbitrary random $\mathbf{v} = (v_1, \dots, v_p)^\top \in \mathbb{R}^{p \times 1}$, we define $\texttt{Var}(\mathbf{v}) := \sum_{j=1}^p \mathbb{E}[(v_j - \mathbb{E}[v_j])^2]$, i.e., variance of $\mathbf{v}$ is the sum of the variances of its elements. Table 1 in the appendix summarizes the used notations. Finally, we have the following assumption:

**Assumption 3.2.** *The stochastic gradient $g_i^{e,t}(\boldsymbol{\theta}) = \frac{1}{b_i^e} \sum_{j \in \mathcal{B}_i^{e,t}} g_{ij}(\boldsymbol{\theta})$ is an unbiased estimate of $\nabla f_i(\boldsymbol{\theta})$ with a bounded variance: $\forall \boldsymbol{\theta} \in \mathbb{R}^p : \texttt{Var}(g_i^{e,t}(\boldsymbol{\theta})) \le \sigma_{i,g}^2(b_i^e)$. The tight bound $\sigma_{i,g}^2(b_i^e)$ is a constant depending only on the used batch size $b_i^e$: the larger the batch size $b_i^e$, the smaller $\sigma_{i,g}^2(b_i^e)$.*

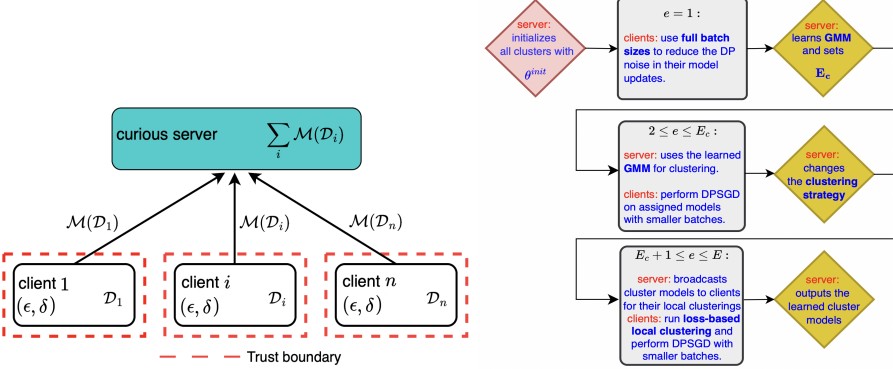

Figure 1: **Left:** Considered threat model in this work, where client $i$ has local train data $\mathcal{D}_i$ and DP privacy parameters $(\epsilon, \delta)$, and does not trust any external parties, including the server. **Right:** Three main stages of the proposed RC-DPFL algorithm, with the key components being highlighted.

## 4 MOTIVATION, METHODOLOGY AND PROPOSED ALGORITHM

We start with the shortcomings of the existing *non-DP* clustered FL algorithms. As discussed in (Werner et al., 2023), algorithms clustering clients based on their loss values (Mansour et al., 2020; Ghosh et al., 2020; Ruan & Joe-Wong, 2021), i.e., assign client $i$ to cluster $R^e(i) = \arg\min_m f_i(\boldsymbol{\theta}_m^e)$ at the beginning of round $e$, are prone to clustering errors in the first few rounds, mainly due to random initialization of cluster models $\{\boldsymbol{\theta}_m^e\}_{m=1}^M$. On the other hand, clustering clients based on their model updates (gradients) (Werner et al., 2023; Briggs et al., 2020; Sattler et al., 2019) makes sense only when the updates are obtained on the same model initialization. Additionally, even if we assume these algorithms can initially cluster clients perfectly in each round $e$, the clients' model updates (gradients) will approach zero as the clusters' models converge to their optimum parameters. Hence, clients from different clusters may appear to belong to the same cluster, which results in clustering mistakes. To illustrate these shortcomings, we provide a more detailed example in Appendix C.

For the above mentioned reasons, we next propose an algorithm which starts with clustering clients based on their model updates for the first several rounds and then switches its strategy to cluster clients based on their loss values. We also augment this idea with some other non-obvious techniques to enhance the clustering accuracy in the first few rounds, when the most clustering uncertainty exists.

### 4.1 RC-DPFL ALGORITHM

Considering the points above, which were overlooked in the existing *non-private* algorithms, we propose our *differentially private* RC-DPFL algorithm with the following steps (see Figure 1, right):

- Initializing clusters uniformly ($\forall m \in [M] : \boldsymbol{\theta}_m^1 = \boldsymbol{\theta}^{init}$), clients use **full batch sizes** in the first round to make their model updates $\{\Delta\tilde{\boldsymbol{\theta}}_i^1\}_{i=1}^n$ less noisy. Then, the server **soft clusters them by running GMM on their model updates**. The remaining clustering uncertainties are incorporated in the probabilities returned by GMM ($\pi_{i,m}$).

- During the subsequent rounds $e \in \{2, \ldots, E_c\}$, the server uses the learned GMM to soft-cluster clients: client $i$ contributes to the training of each cluster ($m$) model proportional to the probability of its assignment to that cluster ($\pi_{i,m}$). **The duration $E_c$ for using the GMM depends on the "confidence level" of the GMM.**

- After the first $E_c$ rounds, some progress has been made in the training of the cluster models $\{\boldsymbol{\theta}_m^{E_c}\}_{m=1}^M$. Now, is the right time to **hard cluster clients based on their *loss values*** in the remaining rounds to build more personalized models per cluster: $R^e(i) = \arg\min_m f_i(\boldsymbol{\theta}_m^e)$.

In Section 4.2.1, we provide theoretical justification for why using full batch sizes in the initial round improves the clustering quality of GMM considerably. Also, in Section 4.3 we analyze the convergence rate for learning the GMM and show that the computational overhead of using GMM is also low. Note that even when clients have a limited memory budget, they can still perform DPSGD with full batch size using gradient accumulation technique (see Appendix I). The technique causes no extra computational overhead, as it just accumulates multiple gradient updates into one update.

---

**Algorithm 1:** RC-DPFL

---

**Input:** Initial parameter $\boldsymbol{\theta}^{init}$, number of clusters $M$, batch size $b$, dataset sizes $\{N_1, \ldots, N_n\}$, noise scales $\{z_1, \ldots, z_n\}$, gradient norm bound $c$, local epochs $K$, global round $E$.

**Output:** cluster models $\{\boldsymbol{\theta}_m^E\}_{m=1}^M$

1   **Initialize** $\boldsymbol{\theta}_1^1 = \ldots = \boldsymbol{\theta}_m^1 = \boldsymbol{\theta}^{init}$ ;         // "uniform" initializations

2   **for** $e \in \{1, \ldots, E\}$ **do**

3     **if** $e = 1$ **then**

4       **for** *each client* $i \in \{1, .., n\}$ *in parallel* **do**

5         $b_i^1 \leftarrow N_i$ ;                   // full batch size

6         $\Delta\tilde{\boldsymbol{\theta}}_i^1 \leftarrow \texttt{DPSGD}\,(\boldsymbol{\theta}_i^1, b_i^1, N_i, K, z_i, c)$

7       on server:

8       **if** $M$ *is unknown* **then**

9         $M = \arg\max_m \texttt{MSS}\Big(\mathbf{GMM}(\Delta\tilde{\boldsymbol{\theta}}_1^1, \ldots, \Delta\tilde{\boldsymbol{\theta}}_n^1; m)\Big)$ ;     // Appendix F.2

10       $\{\pi_1, \ldots, \pi_n, \texttt{MPO}\} = \mathbf{GMM}(\Delta\tilde{\boldsymbol{\theta}}_1^1, \ldots, \Delta\tilde{\boldsymbol{\theta}}_n^1; M)$ ;     // **1st stage:** GMM

11       set $E_c(\texttt{MPO})$ ;               // $E_c$ is set based on MPO

12       continue ;                    // go to next round ($e = 2$)

13     **else if** $e \in \{2, \ldots, E_c\}$ **then**

14       **for** *each client* $i \in \{1, \ldots, n\}$ **do**

15         $R^e(i) \leftarrow m$ *with probability* $\pi_i[m]$ ;   // **2nd stage:** soft clustering

16     **else**

17       on server: broadcast cluster models $\{\boldsymbol{\theta}_m^e\}_{m=1}^M$ to all clients

18       **for** *each client* $i \in \{1, \ldots, n\}$ **do**

19         $R^e(i) = \arg\min_m f_i(\boldsymbol{\theta}_m^e)$ ;     // **3rd stage:** "local" clustering

20     **for** *each client* $i \in \{1, .., n\}$ *in parallel* **do**

21       $b_i^e \leftarrow b$ ;                    // batch size $b$

22       $\Delta\tilde{\boldsymbol{\theta}}_i^e \leftarrow \texttt{DPSGD}\,(\boldsymbol{\theta}_{R^e(i)}^e, b_i^e, N_i, K, z_i, c)$

23     on server:

24     **for** *each client* $i \in \{1, \ldots, n\}$ **do**

25       $w_i^e \leftarrow \frac{N_i}{\sum_{j=1}^n \mathbb{1}_{R^e(j)=R^e(i)} N_j}$

26     **for** $m \in \{1, \ldots, M\}$ **do**

27       $\boldsymbol{\theta}_m^{e+1} \leftarrow \boldsymbol{\theta}_m^e + \sum_{i \in \{1, \ldots, n\}} \mathbb{1}_{R^e(i)=m} w_i^e \Delta\tilde{\boldsymbol{\theta}}_i^e$ ;     // $i$ contributes to $R^e(i)$

---

## 4.2 REDUCING GMM UNCERTAINTY VIA USING FULL BATCH SIZE IN THE FIRST ROUND

The $\texttt{DP}$ noise in the model updates $\{\Delta\tilde{\boldsymbol{\theta}}_i^1\}_{i=1}^n$ makes it harder for the server to cluster clients by running $\texttt{GMM}$ on the model updates. Thus, an efficient clustering algorithm should be robust to this extra $\texttt{DP}$ noise. The following lemma, which is an extension of a similar result in (Malekmohammadi et al., 2024), shows that the noise in model update $\Delta\tilde{\boldsymbol{\theta}}_i^e$ at the of round $e$, including stochastic and $\texttt{DP}$ noise, heavily drops with the batch size $b_i^e$ that client $i$ uses during round $e$. This suggests to use large batch sizes in the first round to improve the quality of clustering on the server side.

**Lemma 4.1.** *Let us assume $\boldsymbol{\theta}_i^{e,0}$ is the model parameter that client $i$ is assigned at the beginning of round $e$. At the end of round, the client generates the noisy $\texttt{DP}$ model update $\Delta\tilde{\boldsymbol{\theta}}_i^e(b_i^e)$ after $K$ local epochs with step size $\eta_l$. The amount of noise in the resulting model update can be found as:*

$$\sigma_i^{e^2}(b_i^e) := \mathit{Var}(\Delta\tilde{\boldsymbol{\theta}}_i^e(b_i^e)|\boldsymbol{\theta}_i^{e,0}) \approx K \cdot N_i \cdot \eta_l^2 \cdot \frac{pc^2 z_i^2(\epsilon, \delta, b_i^1, b_i^{>1}, N_i, K, E)}{b_i^{e^3}}. \tag{2}$$

*We have shown $b_i^e$ as an argument of $\sigma_i^{e^2}(b_i^e)$ to emphasize on its dependence on $b_i^e$. The lemma means that the noise level in $\Delta\tilde{\boldsymbol{\theta}}_i^e$ decreases fast with $b_i^e$* (Malekmohammadi et al., 2024; Räisä et al.,

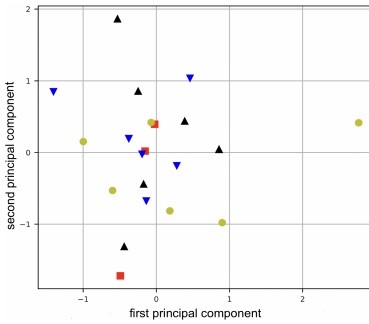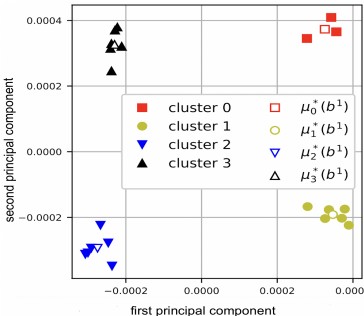

Figure 2: PCA visualization of updates $\{\Delta\tilde{\boldsymbol{\theta}}_i^1\}_{i=1}^n$ on 2D space. **Left:** $\epsilon_i = 10$, $b_i^e = 32$ for all $i$ and $e$. **Right:** $\epsilon_i = 10$, $b_i^1 = b^1 = N = 6600$, i.e., full batch size (assuming $N_i = N = 6600$ for all clients), and $b_i^{>1} = 32$ for all $i$. The empty markers show the centers of the Gaussian components. The model updates are obtained from running DPFedAvg on CIFAR10 with covariate shift (rotation) between clusters, and under the same values as in Figure 3.

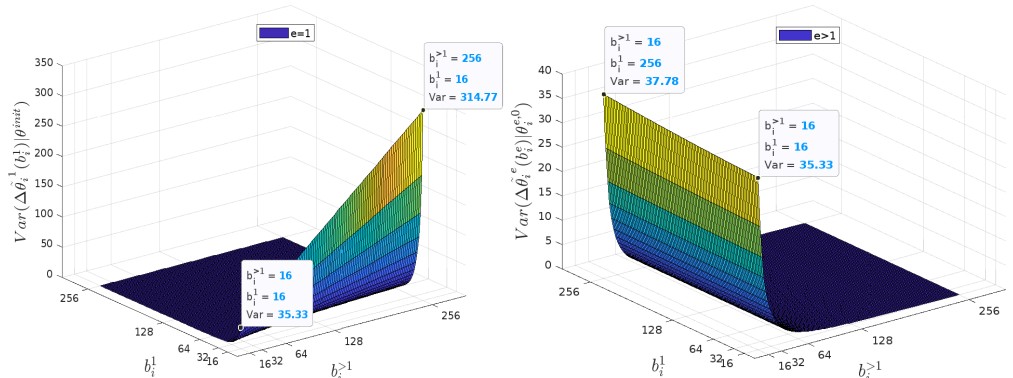

Figure 3: Plot of $\mathtt{Var}(\Delta\tilde{\boldsymbol{\theta}}_i^1(b_i^1)|\boldsymbol{\theta}_i^{init})$ **(left)** and $\mathtt{Var}(\Delta\tilde{\boldsymbol{\theta}}_i^e(b_i^e)|\boldsymbol{\theta}_i^{e,0})$ ($e > 1$) **(right)** v.s. both $b_i^1$ and $b_i^{>1}$ obtained from Equation (2) and Renyi-$\mathtt{DP}$ Accountant (Mironov et al., 2019) in a setting with $N_i = 6600, \epsilon = 5, \delta = 10^{-4}, c = 3, K = 1, E = 200, p = 11,181,642, \eta_l = 5 \times 10^{-4}$. There are two clear messages: 1) for all $e \in \{1, \cdots, E\}$, $\mathtt{Var}(\Delta\tilde{\boldsymbol{\theta}}_i^e(b_i^e)|\boldsymbol{\theta}_i^{e,0})$ decreases with $b_i^e$ quickly. **This was observed in Lemma 4.1**. 2) The effect of $b_i^{>1}$ in the left figure is more than the effect of $b_i^1$ in the right figure. The reason is that $b_i^{>1}$ is used in $E-1$ rounds, while $b_i^1$ is used only in the first round. Also, see Figure 8 in the appendix for the plot of $z_i(\epsilon, \delta, b_i^1, b_i^{>1}, N_i, K, E)$ v.s. $b_i^1$ and $b_i^{>1}$.

2024). *Let us consider $e = 1$ especially*: If a client $i$ can increase its batch size 10 times by using its full batch size in round $e = 1$, the variance of the noise in its model update $\Delta\tilde{\boldsymbol{\theta}}_i^1(b_i^1)$ drops almost 1000 times. If all clients do so, it becomes much easier for the server to cluster them at the end of the first round, by learning a $\mathtt{GMM}$ on $\{\Delta\tilde{\boldsymbol{\theta}}_i^1\}_{i=1}^n$, as their updates become more separable. An illustration of the considerable effect of using full batch sizes in the first round (i.e., $b_i^1 = N_i$) on the noise level in model updates $\{\Delta\tilde{\boldsymbol{\theta}}_i^1\}_{i=1}^n$ is shown in Figure 2. Furthermore, instead of fixing $b_i^{>1}$ to some value, we have also demonstrated the effect of *both* batch sizes $b_i^1$ and $b_i^{>1}$ on the noise levels $\mathtt{Var}(\Delta\tilde{\boldsymbol{\theta}}_i^1|\boldsymbol{\theta}_i^{init})$ ($e = 1$) and $\mathtt{Var}(\Delta\tilde{\boldsymbol{\theta}}_i^e|\boldsymbol{\theta}_i^{e,0})$ ($e > 1$) separately, in Figure 3. As a take away, Figure 3 left, suggests that **in order to make $\{\Delta\tilde{\boldsymbol{\theta}}_i^1\}_{i=1}^n$ less noisy, we have to make $\{b_i^1\}_{i=1}^n$ larger and make $\{b_i^{>1}\}_{i=1}^n$ smaller**, similar to what done in Figure 2 right. These interesting results are consistent with the observations in (De et al., 2022; Anil et al., 2021; Dörmann et al., 2021; Hoory et al., 2021; Li et al., 2022; Luo et al., 2021) that increasing the batch size can significantly improve the privacy-utility trade-off of $\mathtt{DPSGD}$. In the next section, we will provide a theoretical justification for these observations, especially Figure 2.

### 4.2.1 Effect of batch sizes $\{b_i^1\}_{i=1}^n$ on model updates $\{\Delta\tilde{\boldsymbol{\theta}}_i^1\}_{i=1}^n$

Hereafter, we focus on round $e = 1$, and show theoretically why increasing batch sizes $\{b_i^1\}_{i=1}^n$ improves the distinguishability of the model updates $\{\Delta\tilde{\boldsymbol{\theta}}_i^1\}_{i=1}^n$. *For simplicity, we assume clients have the same dataset sizes and batch sizes:* $\forall i : N_i = N, b_i^1 = b^1$. Also, remember that $\boldsymbol{\theta}_i^{1,0} = \boldsymbol{\theta}^{init}$. According to Equation (2) and having uniform privacy parameters $(\epsilon, \delta)$, we have: $\forall i : \sigma_i^{1^2}(b^1) :=$ $\mathrm{Var}[\Delta\tilde{\boldsymbol{\theta}}_i^1(b^1)|\boldsymbol{\theta}^{init}] = \sigma^{1^2}(b^1)$. Hence, we can consider the model updates $\{\Delta\tilde{\boldsymbol{\theta}}_i^1(b^1)\}_{i=1}^n$ as the samples from a mixture of $M$ Gaussian distributions with mean, covariance matrix, prior probability parameters: $\psi^*(b^1) = \{\mu_m^*(b^1), \Sigma_m^*(b^1), \alpha_m^*\}_{m=1}^M$, where $\forall m : \alpha_m^* > 0$ and $\mu_m^*(b^1) \neq \mu_{m'}^*(b^1)$ $(m \neq m')$. Also, model update $\Delta\tilde{\boldsymbol{\theta}}_i^1(b^1)$ comes from component $m = s(i)$:

$$\mu_m^*(b^1) := \mathbb{E}\left[\Delta\tilde{\boldsymbol{\theta}}_i^1(b_i^1)\bigg|\boldsymbol{\theta}^{init}, b_i^1 = b^1, s(i) = m\right], \tag{3}$$

$$\Sigma_m^*(b^1) := \mathbb{E}\left[\left(\Delta\tilde{\boldsymbol{\theta}}_i^1(b_i^1) - \mu_m^*(b^1)\right)\left(\Delta\tilde{\boldsymbol{\theta}}_i^1(b_i^1) - \mu_m^*(b^1)\right)^\top\bigg|\boldsymbol{\theta}^{init}, b_i^1 = b^1, s(i) = m\right] = \frac{\sigma^{1^2}(b^1)}{p}\mathbb{I}_p, \tag{4}$$

where the last equality is from $\mathrm{Var}[\Delta\tilde{\boldsymbol{\theta}}_i^1|\boldsymbol{\theta}^{init}, b_i^1 = b^1] = \mathbb{E}[\|\Delta\tilde{\boldsymbol{\theta}}_i^1 - \mu_{s(i)}^*(b^1)\|^2] = \sigma^{1^2}(b^1)$ and that the noises existing in each of the $p$ elements of $\Delta\tilde{\boldsymbol{\theta}}_i^1$ are *i.i.d* (hence, $\Sigma_m^*(b^1)$ is a diagonal covariance matrix with equal diagonal elements). Intuitively, we expect more separation between the true Gaussian components $\{\mathcal{N}\left(\mu_m^*(b^1), \Sigma_m^*(b^1)\right)\}_{m=1}^M$, from which clients' updates $\{\Delta\tilde{\boldsymbol{\theta}}_i^1\}_{i=1}^n$ are sampled, to make the model updates more distinguishable for server. In the following, we show that the overlap between the Gaussian components $\{\mathcal{N}(\mu_m^*(b^1), \Sigma_m^*(b^1))\}_{m=1}^M$ decreases fast with $b^1$.

**Lemma 4.2.** *Let us assume* $\Delta_{m,m'}(b^1) := \|\mu_m^*(b^1) - \mu_{m'}^*(b^1)\|$ *when* $\forall i : b_i^1 = b^1$. *Then, the overlap between the pair* $\mathcal{N}\left(\mu_m^*(b^1), \Sigma_m^*(b^1)\right)$ *and* $\mathcal{N}\left(\mu_{m'}^*(b^1), \Sigma_{m'}^*(b^1)\right)$ *is* $O_{m,m'} = 2Q(\frac{\sqrt{p}\Delta_{m,m'}(b^1)}{2\sigma^1(b^1)})$, *where* $\sigma^{1^2}(b^1) := \mathrm{Var}[\Delta\tilde{\boldsymbol{\theta}}_i^1|\boldsymbol{\theta}^{init}, b_i^1 = b^1]$ *and* $Q(\cdot)$ *is the tail distribution function of the standard normal distribution. Furthermore, if we increase* $b_i^1 = b^1$ *to* $b_i^1 = kb^1 \leq N$ *(for all* $i$*), we have* $O_{m,m'} \leq 2Q(\frac{\sqrt{kp}\Delta_{m,m'}(b^1)}{2\rho\sigma^1(b^1)})$, *where* $1 \leq \rho \in \mathcal{O}(1)$ *is a small constant.*

The lemma states that using a large batch size in the first round results in a *fast* reduction of the overlap between the underlying components, which leads to more distinguishability for $\{\Delta\tilde{\boldsymbol{\theta}}_i^1\}_{i=1}^n$ on the server side (see Figure 2, right). *One of the beneficial consequences of this well separation is that RC-DPFL becomes robust to the initialization of the GMM model.* Furthermore, note that for a fixed batch size $b^1$, the terms $\Delta_{m,m'}(b^1)$ and $\sigma^1(b^1)$ represent the "data heterogeneity level across clusters $m$ and $m'$" and "privacy sensitivity of their clients", respectively. We define the "separation score" $\mathrm{SS}(m, m') = \frac{\sqrt{p}\Delta_{m,m'}(b^1)}{2\sigma^1(b^1)} = \frac{\Delta_{m,m'}(b^1)}{2\sigma^1(b^1)/\sqrt{p}}$ between two components $m$ and $m'$ as a measure of their separability. The larger $\mathrm{SS}(m, m')$, the smaller their overlap $O_{m,m'} = 2Q(\mathrm{SS}(m, m'))$. Based on the form of the Q function, an $\mathrm{SS}(m, m')$ above 3 can be considered as a complete separation.

### 4.2.2 Confidence of GMM

As we observed in Lemma 4.2, the separation score $\mathrm{SS}(m, m')$ (the overlap $O_{m,m'}$) increases (decreases) as $b^1$ increases. Remember that $\mathrm{SS}(m, m') = \frac{\Delta_{m,m'}(b^1)}{2\sigma^1(b^1)/\sqrt{p}}$, and note that $\sigma^{1^2}(b^1)/p$ is the value of diagonal elements of covariance matrices of Gausssian components (Equation (4)), *which the GMM aims to learn.* Therefore, when the GMM is learned, we can use its parameters to get an estimate $\hat{\mathrm{SS}}(m, m')$ for every cluster pair $m$ and $m'$. Then, we can define the **"minimum pairwise separation score"** as $\mathrm{MSS} = \min_{m,m'} \hat{\mathrm{SS}}(m, m') \in [0, +\infty)$ **as a measure of confidence** of the learned GMM in its clusterings. The larger the MSS of a learned GMM, the more "confident" it is in its clustering decisions. For instance, if we learn a GMM on Figure 2 left, it will have a much smaller MSS than when we learn a GMM on Figure 2 right. We can similarly define the estimated **"maximum pairwise overlap"** for a learned GMM as $\mathrm{MPO} = 2Q(\mathrm{MSS}) \in [0, 1)$, **as a measure of uncerntainty of the learned GMM** (the smaller the better. Q is a decreasing function).

### 4.3 CONVERGENCE RATE OF EM FOR LEARNING GMM

Let us define the maximum pairwise overlap in $\psi^*(b^1) = \{\mu_m^*(b^1), \Sigma_m^*(b^1), \alpha_m^*\}_{m=1}^M$, as $O^{\max}(\psi^*(b^1)) = \max_{m,m'} O_{m,m'}(\psi^*(b^1))$. According to Lemma 4.2, when $b^1$ is large enough, $O^{\max}(\psi^*(b^1))$ decreases (like in Figure 2, right) and we can expect EM to converge to the true GMM parameters $\psi^*(b^1)$. Next, we analyze the local convergence rate of EM around the true solution.

**Theorem 4.3.** *(Ma et al., 2000) Given model updates $\{\Delta\tilde{\boldsymbol{\theta}}_i^1(b^1)\}_{i=1}^n$, which are samples from a true mixture of Gaussians $\{\mathcal{N}(\mu_m^*(b^1), \Sigma_m^*(b^1)), \alpha_m^*\}_{m=1}^M$, if $O^{\max}(\psi^*(b^1))$ is small enough, then:*

$$\lim_{r\to\infty} \frac{\|\psi^{r+1} - \psi^*(b^1)\|}{\|\psi^r - \psi^*(b^1)\|} = o\left(\left[O^{max}(\psi^*(b^1))\right]^{0.5-\gamma}\right), \tag{5}$$

*as $n$ increases. $\psi^r$ is the GMM parameters returned by EM after $r$ iterations. $\gamma$ is an arbitrary small positive number, and $o(x)$ means it is a higher order infinitesimal as $x \to 0 : \lim_{x\to 0} \frac{o(x)}{x} = 0$.*

This means that convergence rate of EM around the true solution $\psi^*(b^1)$ is faster than how $\left[O^{\max}(\psi^*(b^1))\right]^{0.5-\gamma}$ decreases with $b^1$. In Lemma 4.2, we showed that $O^{\max}(\psi^*(b^1))$ indeed drops fast as $b^1$ increases. Therefore, if clients have a large enough dataset size and use full batch sizes in the first round, convergence rate of EM approaches approximately 0. *Hence, as an important consequence, the computational complexity of learning the GMM in the first round decreases fast.*

### 4.4 APPLICABILITY OF RC-DPFL

Even when the number of the underlying clusters $(M)$ is not known beforehand, we can find it with high accuracy based on the confidence metric MSS $\in (0, +\infty)$ defined above (line 9 of Algorithm 1). Intuitively, we choose the $M$ which yields to the largest confidence level MSS for the resulting GMM. We have provided further details about how to find $M$ in these scenarios in Appendix F.2.

The strategy switching time $E_c$ can also be set using the uncertainty metric MPO $\in [0, 1)$. Intuitively, if the learned GMM is not certain about its clustering decisions, RC-DPFL should not rely on its decisions for a large $E_c$, and vice versa. Hence, we can set $E_c$ as a decreasing function of MPO. For instance, $E_c = (1 - \text{MPO})\frac{E}{2}$ means that if a GMM is completely confident about its clusterings, e.g., what happens in Figure 2 right, the server changes the clustering strategy to loss-based after the first half of the training time. This change happens earlier as the uncertainty increases (e.g., when $\epsilon$ is small), and RC-DPFL slowly gets close to the completely loss-based clustering.

Furthermore, we already know that in order to have a quality client clustering at the end of the first round, $\{b_i^{>1}\}_{i=1}^n$ should be small (from Figure 3. Also, see Appendix F.1 for a more detailed discussion). Finally, note that after the training progress made in the first $E_c$ rounds, the loss-based hard clustering is performed "locally" at clients' side (Ghosh et al., 2020) (line 17 in Algorithm 1). Also, at this stage, the sensitivity of the local model selection of a client $i$ to adding/removing a data point to its local dataset is effectively zero. Therefore, there is no privacy concern regarding the local loss-based clusterings performed in the last stage of RC-DPFL (see Appendix H for a formal privacy proof). These important features altogether make RC-DPFL a robust and applicable algorithm.

## 5 EVALUATION

### 5.1 EXPERIMENTAL SETUP

**Datasets, models and baseline algorithms:** We evaluate our proposed method on three benchamark datasets, including: MNIST (Deng, 2012), FMNIST (Xiao et al., 2017) and CIFAR10 (Krizhevsky, 2009), with heterogeneous data distributions from covariate shift (rotation) (Kairouz et al., 2021; Werner et al., 2023) and concept shift (label flip) (Werner et al., 2023), which are the commonly used data splits in the literature (see Appendix B). We consider four clusters of clients indexed by $m \in \{0, 1, 2, 3\}$ with $\{3, 6, 6, 6\}$ clients, where the smallest cluster is considered as the minority group. To the best of our knowledge, there was no prior work on DP clustered FL, so we compared to the DP version of existing algorithms. More specifically, we compare with the following baseline

algorithms, which are combined with DPSGD: 1. DPFedAvg (Noble et al., 2021): clients run DPSGD locally and send their model updates to the server 2. KM-CDPFL (Werner et al., 2023): FL with myopic clustering, in which server clusters clients at the end of each round using running K-means clustering on their DP model updates 3. f-CDPFL (Ghosh et al., 2020; Mansour et al., 2020): FL with loss clustering, which clusters clients based on their train loss values on existing cluster models 4. Oracle-CDPFL: an oracle algorithm which has the knowledge of true clusters from the first round.

**Evaluation metrics and baselines:** Given the set of $n$ clients, fairness in a DPFL system can be measured in terms of the disparate impact of DP on utility (performance drop) of different groups (Chu et al., 2023; Bagdasaryan & Shmatikov, 2019; Tran et al., 2021; Esipova et al., 2022): $\mathcal{F}_{acc} = \max_{i,j \in [n]} |\Delta acc_i(\boldsymbol{\theta}_i) - \Delta acc_j(\boldsymbol{\theta}_j)|$, where $\boldsymbol{\theta}_i$ is the model assigned to agent $i$ at the end of DP training, and $\Delta acc_i(\boldsymbol{\theta}_i) = \max_{\boldsymbol{\theta}^*} acc_i(\boldsymbol{\theta}^*) - acc_i(\boldsymbol{\theta}_i)$, where $\boldsymbol{\theta}^*$ is any possible model. Similarly, we can measure fairness in terms of the increment enforced to clients' train loss (Tran et al., 2021; Esipova et al., 2022; Chu et al., 2023): $\mathcal{F}_{loss} = \max_{i,j \in [n]} |\xi_i(\boldsymbol{\theta}_i) - \xi_j(\boldsymbol{\theta}_j)|$, where $\xi_i(\boldsymbol{\theta}_i) = f_i(\boldsymbol{\theta}_i) - \min_{\boldsymbol{\theta}^*} f_i(\boldsymbol{\theta}^*)$. These notions of fairness compare the cost of adding differential privacy on different clients, and define client-level fairness as the equality of "performance drop" across clients. Following (Chu et al., 2023), we estimate the model $\boldsymbol{\theta}^*$ for each cluster by centrally training a model with SGD based on the data of the clients belonging to that cluster. We also consider the following evaluation metrics: average test accuracy (overall, majority, minority), worst accuracy across clients (Mohri et al., 2019), and maximum accuracy disparity across clients: $\max_{i,j} |acc_i(\boldsymbol{\theta}_i) - acc_j(\boldsymbol{\theta}_j)|$.

## 5.2 RESULTS

In our experiments, we aim to 1) compare RC-DPFL with other clustering approaches, 2) analyze its robustness to noise; and 3) evaluate its robustness to different types of data heterogeneity.

**RQ1: How does RC-DPFL perform compared to other algorithms?** We first explore how RC-DPFL performs in comparison with the defined baseline algorithms. Figure 4 shows the performance for MNIST and FMNIST in terms of per cluster performance and fairness metric $\mathcal{F}_{acc}$. Through these results, it is clear that RC-DPFL performance on-par with the oracle-DPFL baseline, which constitutes the ideal case. This is mainly attributed to the accurate clustering obtained in RC-DPFL as seen in subfigure (c), which compares the success rate of clustering using loss function (f-CDPFL) versus our approach. Also, KM-CDPFL incurs the highest unfairness $\mathcal{F}_{acc}$, due to clustering errors.

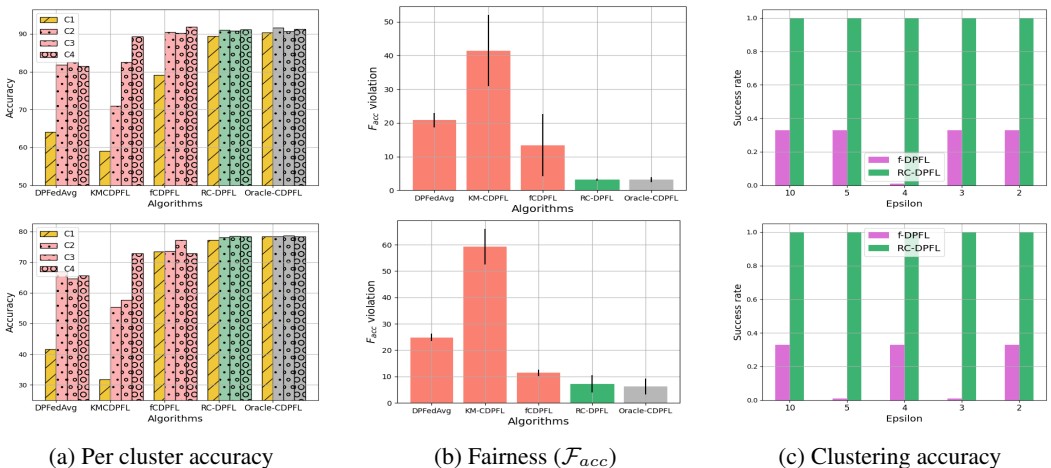

(a) Per cluster accuracy      (b) Fairness ($\mathcal{F}_{acc}$)      (c) Clustering accuracy

Figure 4: Comparison of RC-DPFL with the defined baselines on MNIST (Top row) and FMNIST (bottom row) with C1 being the minority cluster. All results are for $\epsilon = 5$.

**RQ2: How does RC-DPFL perform under different levels of noise?** Figure 5 on CIFAR10, shows the effect of varying levels of DP noise on the fairness of different algorithms ($\delta = 10^{-4}$) in terms of three different metrics. RC-DPFL performs close to the oracle algorithm in terms of all the three metrics, which shows its robustness to the DP noise in the system. RC-DPFL has the smallest gap between majority and minority groups in terms of the three disparity metrics and outperforms the two

other baseline algorithms for improving the minority cluster. The gap is even larger on smaller values of $\epsilon$, for instance, the minimum accuracy for $\epsilon = 2$ using RCDPFL is 5% and 9% higher than the best performing benchmark algorithm on FMNIST and MNIST respectively, while unfairness ($\mathcal{F}_{acc}$) is 6% and 13% lower. Detailed results for other datasets can be found in Tables 4–9 in the appendix, which include results for accuracy across groups and various fairness metrics.

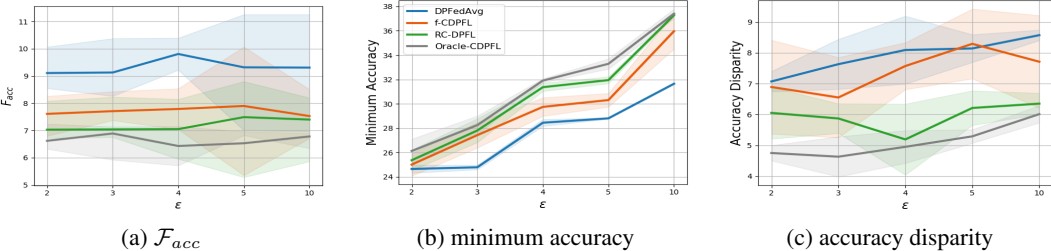

(a) $\mathcal{F}_{acc}$        (b) minimum accuracy        (c) accuracy disparity

Figure 5: Effect of $\epsilon$ on fairness (legends are common between the figures). Results are obtained on CIFAR10. RC-DPFL performs very close to the oracle method and outperforms the baselines. Due to its low performance, we have not shown KM-CDPFL, for better visibility. See Table 9 for details.

**RQ3: How does RC-DPFL perform under different types of data heterogeneity across clients?**
We evaluate how different types of distribution shift across client groups affect the clustered `DPFL` algorithms. To do so, we compare covariate shift and concept shift on CIFAR10 dataset. Concept shift, which can also be viewed as a label flipping attack, has a more significant impact on performance in the single model case, as labels vary across client groups. Figure 6 shows results with $\epsilon = 5.0$ in terms of per-cluster performance and fairness, as well as clustering accuracy for different values of $\epsilon$. Through these results, we notice that it is easier to detect minorities with the loss values in the case of concept-shift. Nonetheless, we also note that mistakes become more costly in this case. Table 11, Table 10 in the appendix show a high variance for f-CPFL across experiments, especially smaller values of $\epsilon$, while RC-DPFL is more consistent. Additionally, in terms of fairness metrics, and across different $\epsilon$ values, RC-DPFL still outperforms the baselines, and remains closer to the oracle case.

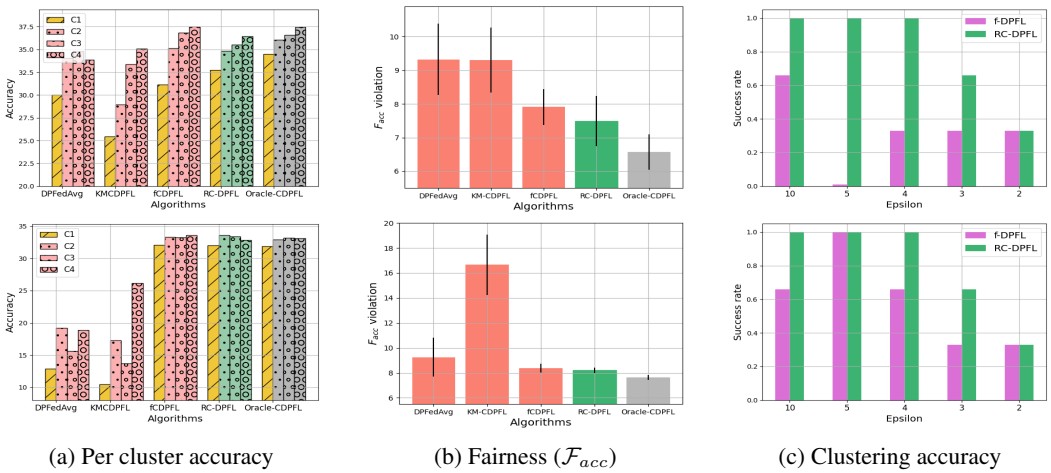

(a) Per cluster accuracy        (b) Fairness ($\mathcal{F}_{acc}$)        (c) Clustering accuracy

Figure 6: Comparison of RC-DPFL with the defined baselines on CIFAR10 with covariate shift (Top row) and concept-shift (bottom row) with C1 being the minority cluster. All results are for $\epsilon = 5$.

## 6   CONCLUSION

We proposed the first `DP` clustered `FL` algorithm, which addresses high data heterogeneity in privacy-sensitive `FL` environments. By clustering clients based on their model updates and training loss values, and mitigating noise impacts with larger batch sizes, our approach enhances utility and fairness with minimal computational overhead, while maintaining `DP`. Moreover, the robustness to noise, and the ability to handle various types of distribution shifts shows the applicability of our approach.

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
