# Appendix for *A Provably Robust Algorithm for Differentially Private Clustered Federated Learning*

## A   NOTATIONS

Table 1: Used notations

| | |
|---|---|
| $n$ | number of clients, which are indexed by $i$ |
| $x_{ij}, y_{ij}$ | $j$-th data point of client $i$ and its label |
| $\mathcal{D}_i, N_i$ | local train set of client $i$ and its size |
| $\mathcal{D}_{i,aug}$ | augmented local train set of client $i$ |
| $\mathcal{B}_i^{e,t}$ | the train data batch used by client $i$ in round $e$ and at the $t$-th gradient update |
| $b_i^e$ | batch size of client $i$ in round $e$: $|\mathcal{B}_i^{e,t}| = b_i^e$ |
| $b_i^1$ | batch size of client $i$ in the first round $e = 1$ |
| $b_i^{>1}$ | set of batch sizes of client $i$ in the rounds $e > 1$ |
| $\epsilon, \delta$ | desired DP privacy parameters |
| $E$ | total number of global communication rounds in the DPFL system, indexed by $e$ |
| $\boldsymbol{\theta}_m^e$ | model parameter for cluster $m$, at the beginning of global round $e$ |
| $K$ | number of local train epochs performed by clients during each global round $e$ |
| $h$ | predictor function, e.g., CNN model, with parameter $\boldsymbol{\theta}$ |
| $\ell$ | cross entropy loss |
| $s(i)$ | the true cluster of client $i$ |
| $R^e(i)$ | the cluster assigned to client $i$ in round $e$ |
| $\boldsymbol{\theta}_i^{e,0}$ | the model parameter passed to client $i$ at the beginning of round $e$ to start its local training |
| $\Delta\tilde{\boldsymbol{\theta}}_i^e$ | the noisy model update of client $i$ at the end of round $e$, starting from $\boldsymbol{\theta}_i^{e,0}$ |
| $\sigma_i^{e^2}$ | conditional variance of the noisy model update $\Delta\tilde{\boldsymbol{\theta}}_i^e$ of client $i$: $\mathrm{Var}(\Delta\tilde{\boldsymbol{\theta}}_i^e|\boldsymbol{\theta}_i^{e,0})$ |
| $\mu_m^*(b^1)$ | the center of the $m$-th cluster (when all clients use batch size $b^1$ in the first round) |
| $\Sigma_m^*(b^1)$ | the covariance matrix of the $m$-th cluster (when all clients use batch size $b^1$ in the first round) |
| $\alpha_m^*$ | the prior probability of the $m$-th cluster |

## B   EXPERIMENTAL SETUP

### B.1   DATASETS

**Data split:**   We use three datasets MNIST, FMNIST and CIFAR10, and consider a distributed setting with 21 clients. In order to create majority and minority clusters, we consider 4 clusters with different number of clients $\{3, 6, 6, 6\}$ (21 clients in total). The first cluster with the minimum number of clients is the "minority" cluster, and the last three are the "majority" ones. The data distribution $P(x, y)$ varies across clusters. We use two methods for making such data heterogeneity: 1. **covariate shift** 2. **concept shift**. In covariate shift we assume that features marginal distribution $P(x)$ differs from one cluster to another cluster. In order to create this variation, we first allocate samples to all clients in an *uniform* way. Then we rotate the data points (images) belonging to the clients in cluster $k$ by $k * 90$ degrees. For concept shift, we assume that conditional distribution $P(y|x)$ differs from one cluster to another cluster, and we first allocate data samples to clients in a uniform way, and flip the labels of the points allocated to clients: we flip $y_{ij}$ (label of the $j$-th data point of client $i$, which belongs to cluster $k$) to $(y_{ij} + k)$ *mod* 10, The local datasets are balanced–all users have the same amount of training samples. The local data is split into train and test sets with ratios $80\%$, and $20\%$, respectively. In the reported experimental results, all users participate in each communication round.

Table 2: CNN model for classification on MNIST/FMNIST datasets

| Layer | Output Shape | # of Trainable Parameters | Activation | Hyper-parameters |
|---|---|---|---|---|
| Input | $(1, 28, 28)$ | 0 | | |
| Conv2d | $(16, 28, 28)$ | 416 | ReLU | kernel size =5; strides=$(1, 1)$ |
| MaxPool2d | $(16, 14, 14)$ | 0 | | pool size=$(2, 2)$ |
| Conv2d | $(32, 14, 14)$ | 12,832 | ReLU | kernel size =5; strides=$(1, 1)$ |
| MaxPool2d | $(32, 7, 7)$ | 0 | | pool size=$(2, 2)$ |
| Flatten | 1568 | 0 | | |
| Dense | 10 | 15,690 | ReLU | |
| Total | | 28,938 | | |

## B.2 MODELS AND OPTIMIZATION

We use a simple 2-layer CNN model with ReLU activation, the detail of which can be found in Table 2 for MNIST and FMNIST. Also, we use the residual neural network (ResNet-18) defined in He et al. (2015), which is a large model. To update the local models allocated to each client during each round, we apply `DPSGD` (Abadi et al., 2016) with a noise scale $z$ which depends on some parameters, as in Equation (2)

Table 3: Details of the experiments and the used datasets in the main body of the paper. ResNet-18 is the residual neural networks defined in He et al. (2015). CNN: Convolutional Neural Network defined in Table 2.

| Datasets | Train set size | Test set size | Data Partition method | # of clients | Model | # of parameters |
|---|---|---|---|---|---|---|
| MNIST | 48000 | 12000 | cov. shift | $\{3, 6, 6, 6\}$ | CNN | 28,938 |
| FMNIST | 50000 | 10000 | cov. shift | $\{3, 6, 6, 6\}$ | CNN | 28,938 |
| CIFAR10 | 50000 | 10000 | cov. and con. shift | $\{3, 6, 6, 6\}$ | ResNet-18 | 11,181,642 |

## B.3 DP PARAMETERS

For each dataset, 5 different values of $\epsilon$ from set $\{2, 3, 4, 5, 10\}$ are used. We fix $\delta$ for all experiments to $10^{-4}$. We also set the clipping threshold $c$ equal to 3, as it results in better test accuracy for the considered datasets, as reported in (Abadi et al., 2016). We use the Renyi DP (RDP) privacy accountant (TensorFlow privacy implementation) during the training time. This accountant is able to handle the difference in the batch size between the first round $e = 1$ and the next rounds $e > 1$ by accounting the composition of the corresponding *heterogeneous* private mechanisms.

## B.4 ALGORITHMS TO COMPARE AND TUNING HYPERPARAMETERS

We compare our RC-DPFL algorithm, which benefits from robust clustering, with four baseline algorithms, including: 1) "DPFedAvg" (Noble et al., 2021), which learns one global model for all clients 2) An extension of the IFCA algorithm (Ghosh et al., 2020) to `DPFL` systems, which we call "f-CDPFL" 3) An extension of the gradient based clustering algorithm (algorithm 1 in (Werner et al., 2023)) to `DPFL` systems, which we call "KM-CDPFL" 4) An oracle algorithm, which has the knowledge of the true underlying clients' clusters, which we call "O-CDPFL". For each algorithm and each dataset, we find the best learning rate from a grid: the one which is small enough to avoid divergence of the `DP` federated optimization, and results in the lowest average loss (across clients) at the end of `FL` training on a "validation set" with size $10,000$ samples. Here are the grids we use for each dataset:

- MNIST: `{1e-3, 2e-3, 5e-3, 1e-2}`;

- FMNIST: `{1e-3, 2e-3, 5e-3, 1e-2}`;

- CIFAR10: `{1e-3, 2e-3, 5e-3, 1e-2}`.

### B.5 GAUSSIAN MIXTURE MODEL

We use the Gaussian Mixture Model of Scikitlearn, which can be found here: https://scikit-learn.org/dev/modules/generated/sklearn.mixture.GaussianMixture.html. The GMM model has three hyper-parameters:

1) parameter initialization, which we set to "k-means++". This is because this type of initialization leads to both low time to initialize and low number of EM iterations for the GMM to converge

2) Type of the covariance matrix, which we set to "spherical", i.e., each component has a diagonal covariance matrix with a single value as its diagonal elements. This is in accordance with Equation (18) and that we know *the covariance matrices should be diagonal*.

3) Finally, the number of components (clusters) is either known or it is unknown. In the latter case, we have explained in Appendix F.2 how we can find the true number of clusters by using the confidence level (MSS) of the GMM model.

### B.6 HYPERPARAMETERS OF THE RC-DPFL

As explained in the paper, RC-DPFL has four hyperparameters, which we know how to set, as explained in Section 4.4:

1) $b_i^1$ is set locally by clients to $N_i$ (their dataset size), i.e., full batch size for easier clustering at the end of the first round

2) $b_i^{>1}$, which is the batch size used during the rounds $e > 1$, and has to be set to a small value, as observed in Figure 3 right, and as explained in Appendix F.1. As observed in Figure 10, RC-DPFL is not sensitive to this parameter, as long as a small value is chosen for it. For the results in the paper, we have set $b_i^{>1} = 32$

3) $E_c$ which is the time of changing the clustering strategy, and as explained in Section 4.4, it has to be related to the uncertainty level (MPO) of the learned GMM such that $E_c$ decreases as MPO of the learned GMM increases. In our experiments, we have used $E_c = (1 - \text{MPO})\frac{E}{2}$. In this way, if the learned GMM is uncertain about its clusterings, RC-DPFL does trust it for many rounds and instead, uses loss-based hard clustering from the early rounds by using a smaller $E_c$. Therefore, RC-DPFL gradually reduces to the loss-based clustering method in the baseline "f-CDPFL" as the MPO of the learned GMM increases.

4) Finally, if the true number of clusters $M$ is not known, we can find it by using the very intuitive method explained in Appendix F.2.

Therefore, all the hyperparameters of RC-DPFL can be set efficiently and easily.

## C  EXAMPLES

In this section, we explain an example to elaborate that why clustering clients based on their losses (model updates) is prone to errors in the first (last) rounds. For example, consider Figure 7, where there are $M = 2$ clusters (red and blue) and $n = 4$ clients. The clients in the red cluster have loss functions $f_1(\theta) = 4(\theta + 6)^2$ and $f_2(\theta) = 4(\theta + 5)^2$ with optimum cluster parameter $\theta_1^\infty = -5.5$. Also, the the clients in the blue cluster have loss functions $f_3(\theta) = 4(\theta - 5)^2$ and $f_4(\theta) = 4(\theta - 6)^2$ with optimum cluster parameter $\theta_2^\infty = 5.5$. Clustering algorithms, which cluster clients based on their loss values on clusters' models, are vulnerable to model initialization. For example, in Figure 7, if we initialize the clusters' parameters with $\theta_1^0 = -11$ and $\theta_2^0 = 0$ (shown in the figure), all four clients will initially select cluster 2, since they have smaller losses on its parameter. At $\theta_2^0 = 0$, the average of clients' gradients (model updates) is zero, so all clients will remain stuck at $\theta_2^0$ and will always select cluster 2.

On the other hand, clustering clients based on their model updates (gradients) (Werner et al., 2023; Briggs et al., 2020; Sattler et al., 2019) have clearly issues. One of these issues appears after some rounds of training. For instance, even if we assume these algorithms can initially cluster clients "perfectly" in each round $e$, the clients' model updates (gradients) will approach zero as the clusters' models converge to their optimum parameters. Hence, clients from different clusters may appear to

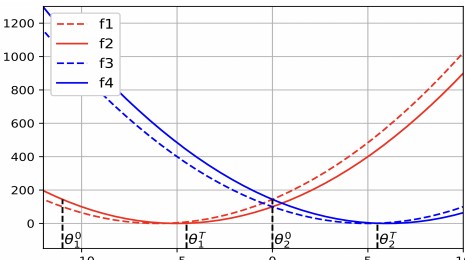

Figure 7: Loss-based clustering algorithms miscluster in the initial rounds, due to model initialization. Also, even with the assumption of perfect clustering of clients in the first rounds, clustering algorithms based on gradients (model updates) lead to clustering errors in the last rounds, due to the gradients approaching to zero.

belong to the same cluster, which results in clustering mistakes. For example, as shown in Figure 1, right, let us assume after $T$ rounds of "correct" clustering of clients, the clusters' parameters get to $\theta_1^T = -4.5$ and $\theta_2^T = 5.5$ (shown in the figure). At this parameters, clients 1 and 2 (which have been "correctly" assigned to cluster 1 so far) will have gradients $f_1'(\theta_1^T) = 12$ and $f_2'(\theta_1^T) = 4$. Similarly, clients 3 and 4 (which have been "correctly" assigned to cluster 2 so far) will have $f_3'(\theta_2^T) = 4$ and $f_4'(\theta_2^T) = -4$. We see that $f_2'$ is closer to $f_3'$ and $f_4'$ than to $f_1'$, and in the next round it will wrongly be assigned to wrong cluster 2. This happens while the clients are clearly distinguishable based on their losses, as some progress in training has been made after $T$ rounds: $f_1(\theta_1^T) = 9$, while $f_1(\theta_2^T) = 23^2$, which clearly means that client 1 correctly belongs to cluster 1. Therefore, after making some progress in training the clusters' models, it makes more sense to use a loss-based clustering strategy than using a strategy based on clients' gradients (model updates).

## D  PROOFS

**Lemma 4.1.** *Let us assume $\boldsymbol{\theta}_i^{e,0}$ is the model parameter that client $i$ is assigned at the beginning of round $e$. At the end of round, the client generates the noisy* DP *model update $\Delta\tilde{\boldsymbol{\theta}}_i^e(b_i^e)$ after $K$ local epochs with step size $\eta_l$. The amount of noise in the resulting model update can be found as:*

$$\sigma_i^{e^2}(b_i^e) := Var(\Delta\tilde{\boldsymbol{\theta}}_i^e(b_i^e)|\boldsymbol{\theta}_i^{e,0}) \approx K \cdot N_i \cdot \eta_l^2 \cdot \frac{pc^2 z_i^2(\epsilon, \delta, b_i^1, b_i^{>1}, N_i, K, E)}{b_i^{e3}}. \tag{2}$$

*Proof.* The proof is restated from (Malekmohammadi et al., 2024). We consider two illustrative scenarios:

**Scenario 1: the clipping threshold $c$ is effective for all samples in a batch:**  in this case we have: $\forall j \in \mathcal{B}_i^{e,t} : c < \|g_{ij}(\boldsymbol{\theta})\|$. Also, we know that the two sources of randomness (i.e. stochastic and Gaussian noise) are independent, thus their variances can be summed up. Let us assume that $E[\bar{g}_{ij}(\boldsymbol{\theta})] = G_i(\boldsymbol{\theta})$ for all samples $j$. From Equation (1), we can find the mean of each *batch gradient* $\tilde{g}_i^{e,t}(\boldsymbol{\theta})$ (of client $i$ in round $e$ and gradient step $t$) as follows:

$$\mathbb{E}[\tilde{g}_i^{e,t}(\boldsymbol{\theta})] = \frac{1}{b_i^e} \sum_{j \in \mathcal{B}_i^{e,t}} \mathbb{E}[\bar{g}_{ij}(\boldsymbol{\theta})] = \frac{1}{b_i^e} \sum_{j \in \mathcal{B}_i^{e,t}} G_i(\boldsymbol{\theta}) = G_i(\boldsymbol{\theta}). \tag{6}$$

Also, from Equation (1), we can find the variance of each *batch gradient* $\tilde{g}_i^{e,t}(\boldsymbol{\theta})$ (of client $i$ in round $e$ and gradient step $t$) as follows:

$$
\sigma_{i,\tilde{g}}^2(b_i^e) := \mathrm{Var}[\tilde{g}_i^{e,t}(\boldsymbol{\theta})] = \mathrm{Var}\left[\frac{1}{b_i^e}\sum_{j\in\mathcal{B}_i^{e,t}}\bar{g}_{ij}(\boldsymbol{\theta})\right] + \frac{p\sigma_{i,\mathrm{DP}}^2}{b_i^{e^2}}
$$

$$
= \frac{1}{b_i^{e^2}}\left(\mathbb{E}\left[\left\|\sum_{j\in\mathcal{B}_i^{e,t}}\bar{g}_{ij}(\boldsymbol{\theta})\right\|^2\right] - \left\|\mathbb{E}\left[\sum_{j\in\mathcal{B}_i^{e,t}}\bar{g}_{ij}(\boldsymbol{\theta})\right]\right\|^2\right) + \frac{pc^2 z_i^2(\epsilon_i,\delta_i,b_i^1,b_i^{>1},N_i,K,E)}{b_i^{e^2}}
$$

$$
= \frac{1}{b_i^{e^2}}\left(\mathbb{E}\left[\left\|\sum_{j\in\mathcal{B}_i^{e,t}}\bar{g}_{ij}(\boldsymbol{\theta})\right\|^2\right] - \left\|\sum_{j\in\mathcal{B}_i^{e,t}}G_i(\boldsymbol{\theta})\right\|^2\right) + \frac{pc^2 z_i^2(\epsilon_i,\delta_i,b_i^1,b_i^{>1},N_i,K,E)}{b_i^{e^2}}
$$

$$
= \frac{1}{b_i^{e^2}}\Big(\underbrace{\mathbb{E}\left[\left\|\sum_{j\in\mathcal{B}_i^{e,t}}\bar{g}_{ij}(\boldsymbol{\theta})\right\|^2\right]}_{\mathcal{A}} - b_i^{e^2}\|G_i(\boldsymbol{\theta})\|^2\Big) + \frac{pc^2 z_i^2(\epsilon_i,\delta_i,b_i^1,b_i^{>1},N_i,K,E)}{b_i^{e^2}}, \tag{7}
$$

where:

$$
\mathcal{A} = \mathbb{E}\left[\left\|\sum_{j\in\mathcal{B}_i^{e,t}}\bar{g}_{ij}(\boldsymbol{\theta})\right\|^2\right] = \sum_{j\in\mathcal{B}_i^{e,t}}\mathbb{E}\left[\|\bar{g}_{ij}(\boldsymbol{\theta})\|^2\right] + \sum_{m\neq n\in\mathcal{B}_i^{e,t}}2\mathbb{E}\left[[\bar{g}_{im}(\boldsymbol{\theta})]^\top[\bar{g}_{in}(\boldsymbol{\theta})]\right]
$$

$$
= \sum_{j\in\mathcal{B}_i^{e,t}}\mathbb{E}\left[\|\bar{g}_{ij}(\boldsymbol{\theta})\|^2\right] + \sum_{m\neq n\in\mathcal{B}_i^{e,t}}2\mathbb{E}\left[\bar{g}_{im}(\boldsymbol{\theta})\right]^\top\mathbb{E}\left[\bar{g}_{in}(\boldsymbol{\theta})\right]
$$

$$
= b_i^e c^2 + 2\binom{b_i^e}{2}\|G_i(\boldsymbol{\theta})\|^2. \tag{8}
$$

The last equation has used Equation (6) and that we clip the norm of sample gradients $\bar{g}_{ij}(\boldsymbol{\theta})$ with an "effective" clipping threshold $c$. By replacing $\mathcal{A}$ into eq. 7, we can rewrite it as:

$$
\sigma_{i,\tilde{g}}^2(b_i^e) := \mathrm{Var}[\tilde{g}_i^{e,t}(\boldsymbol{\theta})] = \frac{1}{b_i^{e^2}}\left(\mathbb{E}\left[\left\|\sum_{j\in\mathcal{B}_i^{e,t}}\bar{g}_{ij}(\boldsymbol{\theta})\right\|^2\right] - b_i^{e^2}\|G_i(\boldsymbol{\theta})\|^2\right) + \frac{pc^2 z_i^2(\epsilon_i,\delta_i,b_i^1,b_i^{>1},N_i,K,E)}{b_i^{e^2}}
$$

$$
= \frac{1}{b_i^{e^2}}\left(b_i^e c^2 + \left(2\binom{b_i^e}{2} - b_i^{e^2}\right)\|G_i(\boldsymbol{\theta})\|^2\right) + \frac{pc^2 z_i^2(\epsilon_i,\delta_i,b_i^1,b_i^{>1},N_i,K,E)}{b_i^{e^2}}
$$

$$
= \frac{c^2 - \|G_i(\boldsymbol{\theta})\|^2}{b_i^e} + \frac{pc^2 z_i^2(\epsilon_i,\delta_i,b_i^1,b_i^{>1},N_i,K,E)}{b_i^{e^2}} \approx \frac{pc^2 z_i^2(\epsilon_i,\delta_i,b_i^1,b_i^{>1},N_i,K,E)}{b_i^{e^2}} \tag{9}
$$

The last approximation is valid because $p \gg 1$ (it is the number of model parameters).

**Scenario 2: the clipping threshold $c$ is ineffective for all samples in a batch:**  when the clipping is ineffective for all samples, i.e., $\forall j \in \mathcal{B}_i^{e,t} : c > \|g_{ij}(\boldsymbol{\theta})\|$, we have a noisy version of the batch gradient $g_i^{e,t}(\boldsymbol{\theta}) = \frac{1}{b_i^e}\sum_{j\in\mathcal{B}_i^{e,t}}g_{ij}(\boldsymbol{\theta})$, which is unbiased with variance bounded by $\sigma_{i,g}^2(b_i^e)$ (see Assumption 3.2). We note that $\sigma_{i,g}^2(b_i^e)$ is a constant that depends on the used batch size $b_i^e$. The larger the batch size $b_i^e$ used during round $e$, the smaller the constant. Hence, in this case:

$$
\mathbb{E}[\tilde{g}_i^{e,t}(\boldsymbol{\theta})] = \mathbb{E}[g_i^{e,t}(\boldsymbol{\theta})] = \nabla f_i(\boldsymbol{\theta}), \tag{10}
$$

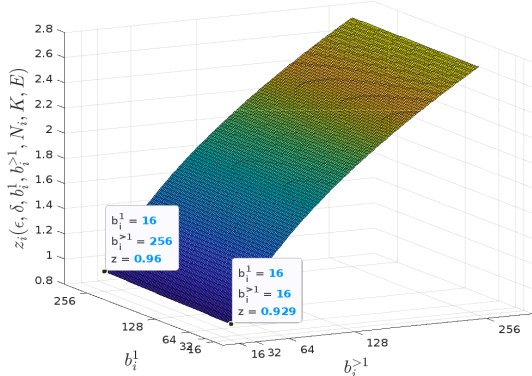

Figure 8: Plot of $z_i(\epsilon, \delta, b_i^1, b_i^{>1}, N_i, K, E)$ v.s. $b_i^1$ and $b_i^{>1}$ obtained from Renyi-DP Accountant (Mironov et al., 2019) in a setting with $N_i = 6600, \epsilon = 5, \delta = 10^{-4}, K = 1, E = 200$. It is clearly observed that the effect of $b_i^{>1}$ is much more than the effect of $b_i^1$. The reason is that $b_i^{>1}$ is used in $E-1$ rounds, while $b_i^1$ is used only in the first round. So it is the value of $b_i^{>1}$ that affects $z_i$ the most.

and

$$\sigma_{i,\tilde{g}}^2(b_i^e) = \text{Var}[\tilde{g}_i^{e,t}(\boldsymbol{\theta})] = \text{Var}[g_i^{e,t}(\boldsymbol{\theta})] + \frac{p\sigma_{i,\text{DP}}^2}{b_i^{e2}} \le \sigma_{i,g}^2(b_i^e) + \frac{p\sigma_{i,\text{DP}}^2}{b_i^{e2}}$$

$$= \sigma_{i,g}^2(b_i^e) + \frac{pc^2 z_i^2(\epsilon, \delta, b_i^1, b_i^{>1}, N_i, K, E)}{b_i^{e2}}$$

$$\approx \frac{pc^2 z_i^2(\epsilon, \delta, b_i^1, b_i^{>1}, N_i, K, E)}{b_i^{e2}}. \tag{11}$$

The approximation is valid because $p \gg 1$ (number of model parameters). Also, note that $\sigma_{i,g}^2(b_i^e)$ decreases with $b_i^e$. Therefore, we got to the same result as in Equation (9).

As observed in see Figure 8, $z_i$ grows with $b_i^1$ and $b_i^{>1}$ *sub-linearly* (especially with $b_i^1$). Therefore, the variance of the client $i$'s DP batch gradients $\tilde{g}_i^{e,t}(\boldsymbol{\theta})$ during communication round $e$, decreases with $b_i^e$ fast. The larger the batch size $b_i^e$, the less the noise existing in its batch gradients during the same round.

With the findings above, we now investigate the effect of batch size $b_i^e$ on **the noise level in clients' model updates at the end of round** $e$. During the global communication round $e$, a participating client $i$ performs $E_i^e = K \cdot \lceil \frac{N_i}{b_i^e} \rceil$ batch gradient updates locally with step size $\eta_l$:

$$\boldsymbol{\theta}_i^{e,k} = \boldsymbol{\theta}_i^{e,k-1} - \eta_l \tilde{g}_i(\boldsymbol{\theta}_i^{e,k-1}), \ k = 1, \ldots, E_i^e. \tag{12}$$

Hence,

$$\Delta \tilde{\boldsymbol{\theta}}_i^e = \boldsymbol{\theta}_i^{e,E_i^e} - \boldsymbol{\theta}_i^{e,0} \tag{13}$$

In each update, it adds a Gaussian noise from $\mathcal{N}(0, \frac{c^2 z_i^2(\epsilon, \delta, b^1, b^{>1}, N_i, K, E)}{b^{e2}} \mathbb{I}_p)$ to its batch gradients independently (see Equation (1)). Hence:

$$\text{Var}[\Delta \tilde{\boldsymbol{\theta}}_i^e | \boldsymbol{\theta}_i^{e,0}] = E_i^e \cdot \eta_l^2 \cdot \sigma_{i,\tilde{g}}^2(b_i^e), \tag{14}$$

where $\sigma_{i,\tilde{g}}^2(b_i^e)$ was computed in Equation (9) and Equation (11), and was a decreasing function of $b_i^e$. Therefore:

$$\text{Var}[\Delta \tilde{\boldsymbol{\theta}}_i^e | \boldsymbol{\theta}_i^{e,0}] \approx K \cdot N_i \cdot \eta_l^2 \cdot \frac{pc^2 z_i^2(\epsilon, \delta, b_i^1, b_i^{>1}, N_i, K, E)}{b_i^{e3}}. \tag{15}$$

$\square$

## E    EFFECT OF BATCH SIZE INCREMENT ON CLUSTERING

**Lemma 4.2.** *Let us assume $\Delta_{m,m'}(b^1) := \|\mu_m^*(b^1) - \mu_{m'}^*(b^1)\|$ when $\forall i : b_i^1 = b^1$. Then, the overlap between the pair $\mathcal{N}\left(\mu_m^*(b^1), \Sigma_m^*(b^1)\right)$ and $\mathcal{N}\left(\mu_{m'}^*(b^1), \Sigma_{m'}^*(b^1)\right)$ is $O_{m,m'} = 2Q(\frac{\sqrt{p}\Delta_{m,m'}(b^1)}{2\sigma^1(b^1)})$, where $\sigma^{1^2}(b^1) := Var[\Delta\tilde{\theta}_i^1 | \theta^{init}, b_i^1 = b^1]$ and $Q(\cdot)$ is the tail distribution function of the standard normal distribution. Furthermore, if we increase $b_i^1 = b^1$ to $b_i^1 = kb^1 \le N$ (for all i), we have $O_{m,m'} \le 2Q(\frac{\sqrt{kp}\Delta_{m,m'}(b^1)}{2\rho\sigma^1(b^1)})$, where $1 \le \rho \in \mathcal{O}(1)$ is a small constant.*

*Proof.* We first find the overlap between two arbitrary Gaussian distributions. Without loss of generality, lets assume we are in 1-dimensional space and that we have two Gaussian distributions both with variance $\sigma^2$ and with means $\mu_1 = 0$ and $\mu_2 = \mu$ ($\|\mu_1 - \mu_2\| = \mu$), respectively. Based on symmetry of the distributions, the two components start to overlap at $x = \frac{\mu}{2}$. Hence, we can find the overlap between the two gaussians as follows:

$$O := 2\int_{\frac{\mu}{2}}^\infty \frac{1}{\sqrt{2\pi}\sigma} e^{-\frac{x^2}{2\sigma^2}} dx = 2\int_{\frac{\mu}{2\sigma}}^\infty \frac{1}{\sqrt{2\pi}} e^{-\frac{x^2}{2}} dx = 2Q(\frac{\mu}{2\sigma}), \tag{16}$$

where $Q(\cdot)$ is the tail distribution function of the standard normal distribution. Now, lets consider the 2-dimensional space, and consider two similar symmetric distributions centered at $\mu_1 = (0,0)$ and $\mu_2 = (\mu, 0)$ ($\|\mu_1 - \mu_2\| = \mu$) and with $\Sigma_1 = \Sigma_2 = \begin{bmatrix} \sigma^2 & 0 \\ 0 & \sigma^2 \end{bmatrix}$. The overlap between the two gaussians can be found as:

$$O = 2\int_{-\infty}^\infty \int_{\frac{\mu}{2}}^\infty \frac{1}{2\pi\sigma^2} e^{-\frac{x^2+y^2}{2\sigma^2}} dxdy = 2\int_{\frac{\mu}{2}}^\infty \frac{1}{\sqrt{2\pi}\sigma} e^{-\frac{x^2}{2\sigma^2}} dx \cdot \int_{-\infty}^\infty \frac{1}{\sqrt{2\pi}\sigma} e^{-\frac{y^2}{2\sigma^2}} dy = 2Q(\frac{\mu}{2\sigma}). \tag{17}$$

If we compute the overlap for two similar symmetric $p$-dimensional distributions with $\|\mu_1 - \mu_2\| = \mu$ and variance $\sigma^2$ in every direction, we will get to the same result $2Q(\frac{\mu}{2\sigma})$.

In the lemma, when using batch size $b^1$, we have two Gaussian distributions $\mathcal{N}\left(\mu_m^*(b^1), \Sigma_m^*(b^1)\right)$ and $\mathcal{N}\left(\mu_{m'}^*(b^1), \Sigma_{m'}^*(b^1)\right)$, where

$$\Sigma_m^*(b^1) = \Sigma_{m'}^*(b^1) = \begin{bmatrix} \frac{\sigma^{1^2}(b^1)}{p} & & \\ & \ddots & \\ & & \frac{\sigma^{1^2}(b^1)}{p} \end{bmatrix}. \tag{18}$$

Therefore, from Equation (17), we can immediately conclude that the overlap between the two Gaussians, which we denote with $O_{m,m'}(b^1)$, is:

$$O_{m,m'}(b^1) = 2Q(\frac{\sqrt{p}\Delta_{m,m'}(b^1)}{2\sigma^1(b^1)}), \tag{19}$$

which proves the first part of the lemma.

Now, lets see the effect of increasing batch size. First, note that we had:

$$\Delta\tilde{\theta}_i^1 = \theta_i^{1,E_i^1} - \theta_i^{1,0},$$
$$\theta_i^{1,k} = \theta_i^{1,k-1} - \eta_l \tilde{g}_i(\theta_i^{1,k-1}), \; k = 1, \dots, E_i^1, \tag{20}$$

where $E_i^1 = K \cdot \lceil \frac{N}{b^1} \rceil$ is the total number of gradients steps taken by client $i$ during communication round $e = 1$. Therefore, considering that DP batch gradients are clipped with a bound $c$, we have:

$$\|\mathbb{E}[\Delta \tilde{\boldsymbol{\theta}}_i^1(b^1)]\| \leq E_i^1 \cdot \eta_l \cdot c. \tag{21}$$

When we increase batch size $b_i^1$ for all clients from $b^1$ to $kb^1$, the upperbound in Equation (21) gets $k$ times smaller. In fact by doing so, the number of local gradient updates that client $i$ performs during round $e = 1$, which is equal to $E_i^1$, decreases $k$ times. As such, we can write:

$$\Delta \tilde{\boldsymbol{\theta}}_i^1(b^1) = k \cdot \Delta \tilde{\boldsymbol{\theta}}_i^1(kb^1) + \upsilon_i, \tag{22}$$

where $\upsilon_i \in \mathbb{R}^p$ is a vector capturing the discrepancies between $\Delta \tilde{\boldsymbol{\theta}}_i^1(b^1)$ and $k \cdot \Delta \tilde{\boldsymbol{\theta}}_i^1(kb^1)$. Therefore, we have:

$$\mu_m^*(b^1) = \mathbb{E}[\Delta \tilde{\boldsymbol{\theta}}_i^1(b^1)|s(i) = m] = \mathbb{E}[k \cdot \Delta \tilde{\boldsymbol{\theta}}_i^1(kb^1) + \upsilon_i|s(i) = m]$$
$$= k \cdot \mathbb{E}[\Delta \tilde{\boldsymbol{\theta}}_i^1(kb^1)] + \mathbb{E}[\upsilon_i|s(i) = m] = k \cdot \mu_m^*(kb^1) + \mathbb{E}[\upsilon_i|s(i) = m]. \tag{23}$$

Therefore, we have:

$$\|\mu_m^*(b^1) - \mu_{m'}^*(b^1)\| = \left\| k\mu_m^*(kb^1) - k\mu_{m'}^*(kb^1) + \left( \mathbb{E}[\upsilon_i|s(i) = m] - \mathbb{E}[\upsilon_i|s(i) = m'] \right) \right\|. \tag{24}$$

Based on our experiments, the last term above, in parenthesis, is small and we can have the following approximation for the equation above:

$$\|\mu_m^*(b^1) - \mu_{m'}^*(b^1)\| \approx \|k\mu_m^*(kb^1) - k\mu_{m'}^*(kb^1)\|, \tag{25}$$

or equivalently:

$$\|\mu_m^*(kb^1) - \mu_{m'}^*(kb^1)\| \approx \frac{\|\mu_m^*(b^1) - \mu_{m'}^*(b^1)\|}{k}. \tag{26}$$

Figure 9 (left) shows the validity of the approximation above with some experimental results. On the other hand, from Equation (2) and also noting that a client, with dataset size $N$ and batch size $b^1$, takes $\frac{N}{b^1}$ gradient steps during each epoch of the first round, we have:

$$\forall m \in [M] : \sigma_m^2(b^1) = \sigma^2(b^1) \approx K \cdot N \cdot \eta_l^2 \cdot \frac{pc^2 z^2(\epsilon, \delta, b^1, b^{>1}, N, K, E)}{b^{1^3}}. \tag{27}$$

When we change the batch size used during the first communication round $e = 1$ from $b^1$ to $kb^1$ and we fix the batch size of rounds $e > 1$, then the noise scale $z$ changes from $z(\epsilon, \delta, b^1, b^{>1}, N_i, K, E)$ to $z(\epsilon, \delta, kb^1, b^{>1}, N_i, K, E)$. Confirmed by our experimental analysis (see Figure 9, right), the amount of change in $z$ due to this is small, as we have changed the batch size only in the first round $e = 1$ from $b^1$ to $kb^1$, while the batch sizes in the other $E - 1$ rounds are unchanged and $E \gg 1$. Therefore, supported by the results in Figure 9, we can always establish an upper bound on the amount of change in $z$ as $b^1$ increases: $z(\epsilon, \delta, kb^1, b^{>1}, N, K, E) \leq \rho z(\epsilon, \delta, b^1, b^{>1}, N, K, E)$, where $\rho$ is a small constant (e.g. $\rho = 2.5$ in Figure 9). So we have:

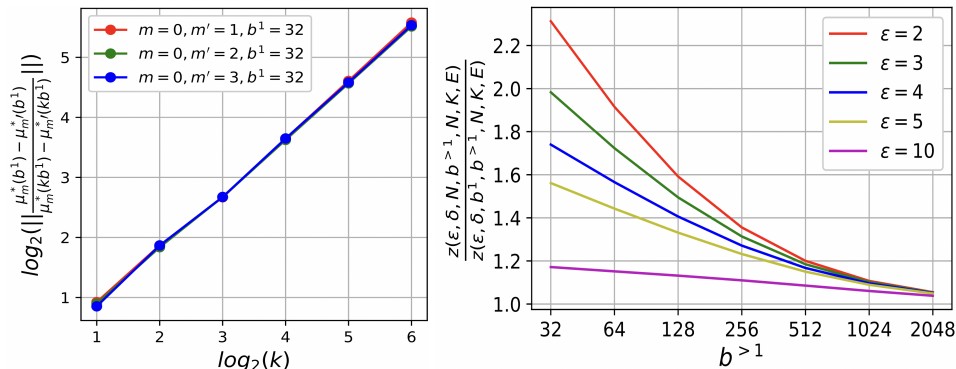

Figure 9: **Left:** Distance between the centers of different clusters, i.e., the distance between $\mu_m^*(b^1)$ and $\mu_{m'}^*(b^1)$, decreases $k$ times as $b^1$ increases $k$ times. The three curves in the plot are obtained on CIFAR10 with 4 clusters $m \in \{0, 1, 2, 3\}$ obtained from covariate shift (rotation). The curves are overlapping all with slope 0.95, which is very close to 1. This shows the validity of the approximation in Equation (26). **Right:** Effect of changing batch size $b^1$ to full batch size in the first round on the noise scale $z$. In the denominator, $b^1$ is equal to $b^{>1}$. Results are obtained from Renyi-DP accountant (Mironov et al., 2019) with $N = 50000$, $K = 1$ and $E = 200$. For each value of $\epsilon$, we have shown the results for seven values of $b^{>1}$.

$$\forall m \in [M] : \sigma_m^2(kb^1) = \sigma^2(kb^1) \approx K \cdot N \cdot \eta_l^2 \cdot \frac{pc^2 z^2(\epsilon, \delta, kb^1, b^{>1}, N, K, E)}{(kb^1)^3}$$

$$\leq K \cdot N \cdot \eta_l^2 \cdot \frac{pc^2 \rho^2 z^2(\epsilon, \delta, b^1, b^{>1}, N, K, E)}{(kb^1)^3}$$

$$= \frac{\rho^2 \sigma^2(b^1)}{k^3}. \tag{28}$$

From Equation (26) and Equation (28), we have:

$$O_{m,m'}(kb^1) = 2Q\left(\frac{\sqrt{p}\Delta_{m,m'}(kb^1)}{2\sigma(kb^1)}\right) \leq 2Q\left(\frac{\sqrt{p}\frac{\Delta_{m,m'}(b^1)}{k}}{2\frac{\rho\sigma(b^1)}{k^{\frac{3}{2}}}}\right) = 2Q(\frac{\sqrt{kp}\Delta_{m,m'}(b^1)}{2\rho\sigma(b^1)}), \tag{29}$$

which completes the proof. □

**Theorem 4.3.** *(Ma et al., 2000) Given model updates $\{\Delta\tilde{\theta}_i^1(b^1)\}_{i=1}^n$, which are samples from a true mixture of Gaussians $\{\mathcal{N}(\mu_m^*(b^1), \Sigma_m^*(b^1)), \alpha_m^*\}_{m=1}^M$, if $O^{max}(\psi^*(b^1))$ is small enough, then:*

$$\lim_{r \to \infty} \frac{\|\psi^{r+1} - \psi^*(b^1)\|}{\|\psi^r - \psi^*(b^1)\|} = o\left([O^{max}(\psi^*(b^1))]^{0.5-\gamma}\right), \tag{5}$$

*as $n$ increases. $\psi^r$ is the GMM parameters returned by EM after $r$ iterations. $\gamma$ is an arbitrary small positive number, and $o(x)$ means it is a higher order infinitesimal as $x \to 0 : \lim_{x \to 0} \frac{o(x)}{x} = 0$.*

*Proof.* The proof directly follows from the proof of Theorem 1 in Ma et al. (2000) by considering $\{\Delta\tilde{\theta}_i^1(b^1)\}_{i=1}^n$ as the samples of Gaussian mixture $\{\mathcal{N}(\mu_m^*(b^1), \Sigma_m^*(b^1)), \alpha_m^*\}_{m=1}^M$. □

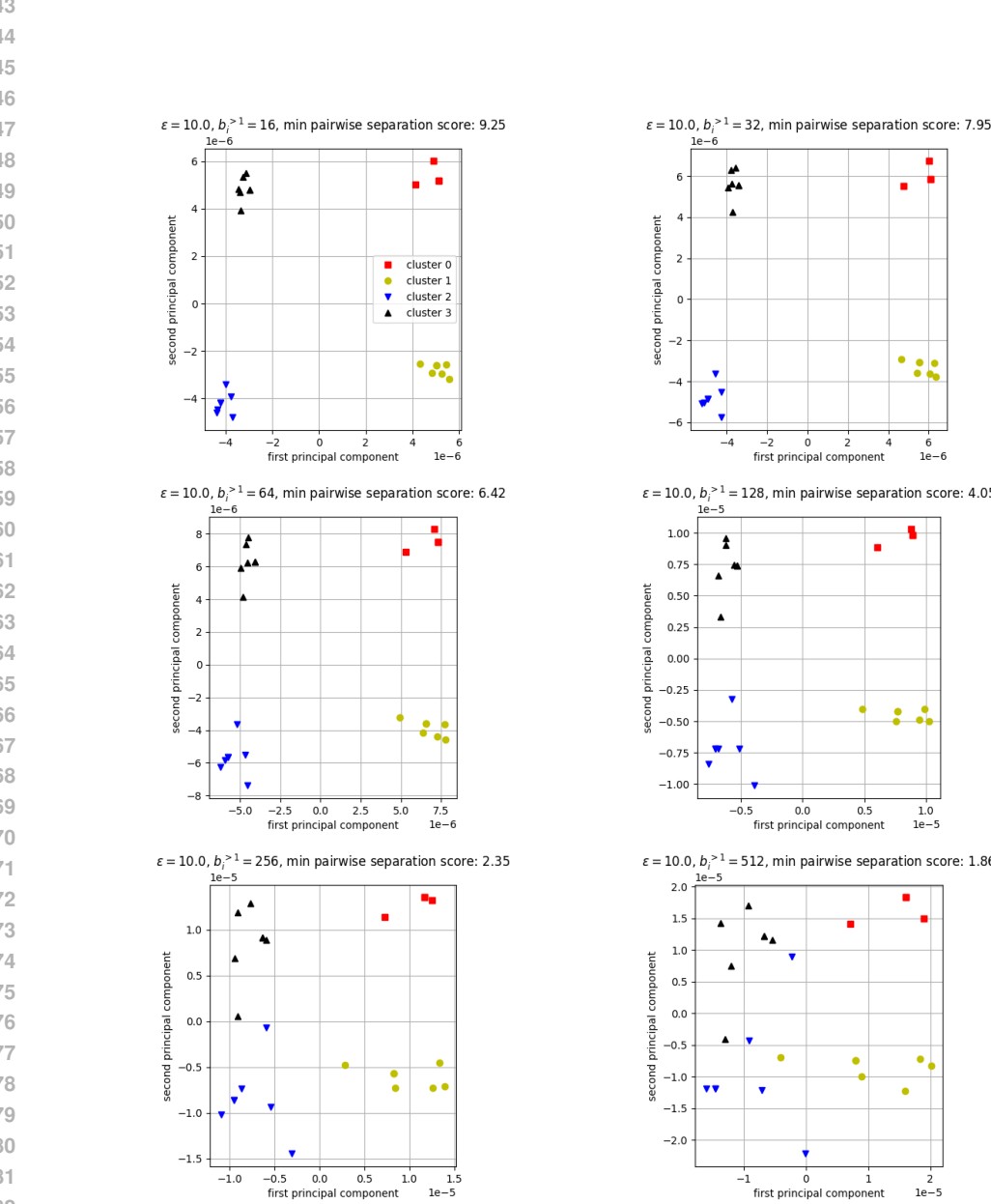

Figure 10: The effect of increasing the batch size after the first round, i.e., $b_i^{>1}$, on the model updates $\{\Delta\tilde{\boldsymbol{\theta}}_i^1\}_{i=1}^n$ at the end of the first round. All clients have used full batch sizes in the first round, i.e., $\forall i : b_i^1 = N_i$. The level of noise in $\{\Delta\tilde{\boldsymbol{\theta}}_i^1\}_{i=1}^n$ affects the quality and confidence of the client clustering that the server performs at the end of the first round. As can be observed, for a fixed $\epsilon = 10$, the model updates scatter further in space as $b_i^{>1}$ increases and different clusters get less separated. This leads to an decrement in the confidence level or the "minimum pairwise separation score (MSS)" of the resulting GMM, as $b_i^{>1}$ increases (see Section 4.2.2 for explanations about the score). The MSS score for each $b_i^{>1}$ is mentioned on the top of the corresponding plot. All the results are obtained on CIFAR10 with covariate shift (rotation) across clusters.

# F  SETTING HYPER-PARAMETERS OF RC-DPFL

## F.1  THE EFFECT OF $b_i^{>1}$ ON CLUSTERING

As we observed in Lemma 4.1 and Figure 3 left, $\text{Var}(\Delta\tilde{\boldsymbol{\theta}}_i^1(b_i^1)|\boldsymbol{\theta}^{init})$, which affects the clustering done at the end of the first round, is an increasing function of $b_i^{>1}$. More generally, we can describe the effect of increasing $b_i^{>1}$ in three things: 1) increasing the noise variance in $\text{Var}(\Delta\tilde{\boldsymbol{\theta}}_i^1(b_i^1)|\boldsymbol{\theta}^{init})$ (as shown in Figure 3, right) 2) decreasing the noise variance in $\text{Var}(\Delta\tilde{\boldsymbol{\theta}}_i^1(b_i^e)|\boldsymbol{\theta}_i^{e,0})$ (as shown in Figure 3, right) 3) decreasing the number of gradients steps during each round $e$ for $e > 1$.

While first one is only limited two the first round $e = 1$, while the last two affect the remaining $E - 1$ rounds $e > 1$ and have conflicting effects on the final accuracy. However, an important point about the problem of clustered DPFL is that finding the true structure of clusters in the first round is a prerequisite for making progress in the next rounds. Therefore, the first effect of increasing the noise variance in $\text{Var}(\Delta\tilde{\boldsymbol{\theta}}_i^1(b_i^1)|\boldsymbol{\theta}^{init})$ is more important and is the most undesirable result. We have demonstrated this effect in Figure 10, which shows that how increasing $b_i^{>1}$ adversely affects the clustering done at the end of the first round. Note how MSS of the learned GMM increases as $b_i^{>1}$ increases. Therefore, in order to have a reliable client clustering at the end of the first round, we need to keep the value of $b_i^{>1}$ as small as possible. Following this observation, we have fixed $b_i^{>1}$ to 32 in all our experimental results, and RC-DPFL was able to outperform the exisitng baseline algorithms.

## F.2  FINDING THE NUMBER OF CLUSTERS ($M$)

Knowing the number of clusters is broadly accepted and applied in the clustered FL literature (Ghosh et al., 2020; Ruan & Joe-Wong, 2021; Briggs et al., 2020). This is the assumption of our baseline algorithms too. Yet, techniques to determine the number of clusters can enable our approach to be more widely adopted. In this section, we show that how we can find the true number of clusters ($M$) when it is not given. Our method, which results in a high accuracy, relies on the MSS score (confidence level) defined in Section 4.2.2: $\text{MSS} = \min_{m,m'} \hat{\text{SS}}(m, m') \in [0, +\infty)$, where $\hat{\text{SS}}(m, m') \approx \frac{\sqrt{p}\Delta_{m,m'}(b^1)}{2\sigma^1(b^1)} = \frac{\Delta_{m,m'}(b^1)}{2\sigma^1(b^1)/\sqrt{p}}$ (please see the detailed explanations in Section 4.2.2). Consider the Figure 2 right as an example. There is a good separation between the $M = 4$ existing clusters, thanks to using full batch size $b^1$ in the first round. Fitting a GMM with 4 components to the model updates results in the highest MSS for the learned GMM model: remember that MSS was the maximum pairwise separation score between the different components of the learned GMM. In contrast, if we fit a GMM with 3 components (less than the true number of components) to the same model updates in the figure, then two clusters will be merged into one component (for examples clusters 0 and 1) leading to a high radius ($\sigma^1(b^1)/\sqrt{p}$) for one of the three components of the resulting GMM. This leads to a low MSS (confidence level) for the resulting GMM. Similarly, if we fit a GMM with 5 components, one of the four clusters (for example cluster 1) will be split between two of the 5 components (call them $m$ and $m'$), which leads to a low inter-component distance ($\Delta_{m,m'}(b^1)$) for the pair of components. This also leads to a low MSS for the resulting GMM. However, fitting a GMM with $M = 4$ components leads to a well separation between all the true components and maximizes the resulting MSS. Based on this very intuitive observation, we propose the following method for setting $m$ at the end of the first round. We select the number of clusters/components, which leads to the maximum MSS for the resulting GMM. More specifically:

$$M = \arg\max_{m \in S} \text{MSS}\Big(\textbf{GMM}(\Delta\tilde{\boldsymbol{\theta}}_1^1, \dots, \Delta\tilde{\boldsymbol{\theta}}_n^1; m)\Big), (\textit{line 9 of Algorithm 1}) \qquad (30)$$

where $S$ is a set of candidate values for $M$: At the end of the first round and on the server, we learn one GMM for each candidate value in $S$ on the same received model updates $\{\Delta\tilde{\boldsymbol{\theta}}_i^1\}_{i=1}^n$. Finally, we choose the value resulting in the GMM with the highest MSS (confidence). It is noteworthy that we know from Lemma 4.2 that learning the GMM does not incur much computational cost.

We have evaluated this method on multiple data splits and different privacy budgets ($\epsilon$) on CIFAR10, MNIST and FMNIST, and it worked perfectly, as shown in Figure 11. As can be observed, the method has made only one mistake for $\epsilon = 4$ (seed 1) and two mistakes for $\epsilon = 3$ (seeds 0 and 1), out of 20 total experiments. Even in those three cases, it has predicted $M$ as 5, which is closest to the

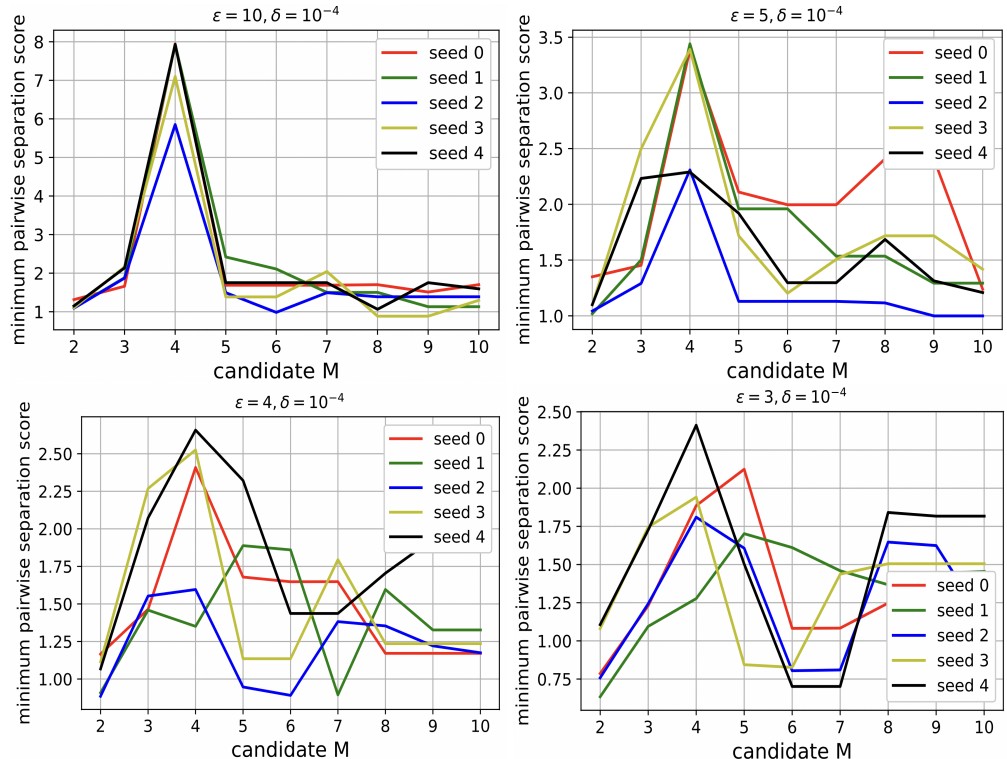

Figure 11: The minimum pairwise separation score (MSS) or confidence of the GMM learned on $\{\Delta\tilde{\boldsymbol{\theta}}_i^1\}_{i=1}^n$ peaks at the true cluster number, which is equal to 4 in all the plots above. Each figure is for a different value of $\epsilon$ (mentioned on top of each figure), and are obtained on CIFAR10 with covariate shift (rotation) across clusters, and 5 different random data splits (5 seeds). All the results are obtained with full batch sizes in the first round and $b_i^{>1} = 32$ for all $i$. We can use this observation as a method to find the true number of clusters ($M$) when it is not given. For larger $\epsilon$, this method work perfectly and even when $\epsilon$ is too small, e.g., $\epsilon = 3$, this method works well and predicts the true number of clusters correctly most of the times: 3 out of the 5 curves in the bottom right plot have a peak at $M = 4$ (the true cluster number). and the other 2 curves predict 5 as the true number, which is the closest and the best alternative for the true value $M = 4$.

true value ($M = 4$) and does not lead to much performance drop. This is because having $M = 5$ splits an existing cluster into two and it is better than predicting for example $M = 3$, which results in "mixing" two clusters with heterogeneous data. **Same method could predict the number of underlying clusters with 100% accuracy for the MNIST and FMNIST datasets for all values of** $\epsilon$. Finally, note that none of the existing baseline algorithms has such an easy and applicable strategy for finding $M$. This shows another useful feature of the proposed RC-DPFL.

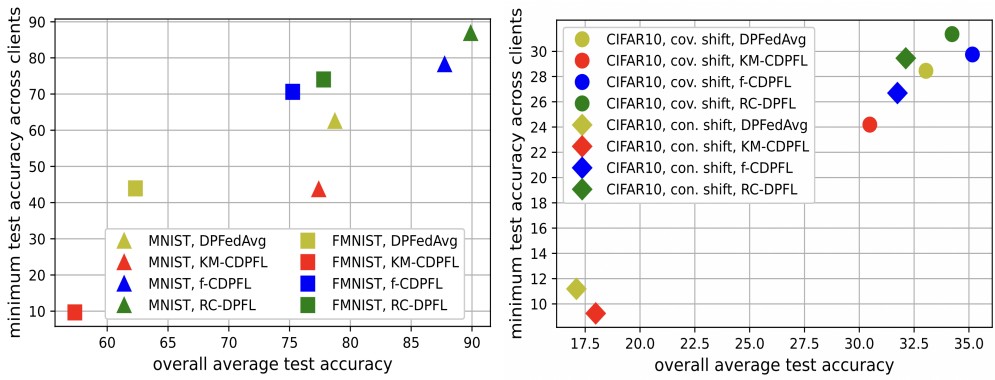

Figure 12: Pareto frontier for overall utility and fairness (in terms of the minimum test accuracy across clients). All the results are for $\epsilon = 4$. The results on MNIST and FMNIST (left) are both on data splits with covariate shift. For other values of $\epsilon$, we observe similar results, as observed in Table 4 to Table 11.

## G  DETAILED EXPERIMENTAL RESULTS

### G.1  MNIST DATA SPLIT WITH COVARIATE SHIFT

Table 4: Average test accuracy of all (All), majority (Maj) and minority (Min) clients on the split of MNIST dataset with covariate shift (rotation). The results are averaged over 3 seeds.

| | algorithm | | $\epsilon = 10$ | $\epsilon = 5$ | $\epsilon = 4$ | $\epsilon = 3$ | $\epsilon = 2$ |
|---|---|---|---|---|---|---|---|
| MNIST | DPFedAvg | All: | $80.92_{\pm 0.83}$ | $79.37_{\pm 0.76}$ | $78.73_{\pm 1.04}$ | $72.88_{\pm 0.79}$ | $72.29_{\pm 0.77}$ |
| | | Maj: | $82.47_{\pm 0.01}$ | $81.16_{\pm 0.01}$ | $80.35_{\pm 0.05}$ | $75.45_{\pm 0.02}$ | $75.02_{\pm 0.05}$ |
| | | Min: | $67.16_{\pm 1.37}$ | $64.04_{\pm 2.06}$ | $63.29_{\pm 1.83}$ | $52.80_{\pm 2.24}$ | $52.50_{\pm 2.76}$ |
| | f-CDPFL | All: | $90.64_{\pm 1.45}$ | $89.12_{\pm 1.69}$ | $87.75_{\pm 0.99}$ | $84.56_{\pm 3.91}$ | $85.01_{\pm 2.71}$ |
| | | Maj: | $\mathbf{92.72}_{\pm 1.42}$ | $\mathbf{91.25}_{\pm 0.01}$ | $85.93_{\pm 0.02}$ | $81.75_{\pm 0.3}$ | $88.80_{\pm 0.03}$ |
| | | Min: | $86.98_{\pm 5.77}$ | $79.12_{\pm 8.94}$ | $81.36_{\pm 6.03}$ | $88.12_{\pm 1.02}$ | $76.07_{\pm 17.79}$ |
| | KM-CDPFL | All: | $81.42_{\pm 3.03}$ | $77.80_{\pm 0.47}$ | $77.40_{\pm 2.06}$ | $71.91_{\pm 2.29}$ | $68.89_{\pm 1.20}$ |
| | | Maj: | $81.55_{\pm 0.01}$ | $79.20_{\pm 0.01}$ | $80.60_{\pm 0.07}$ | $74.38_{\pm 0.02}$ | $73.11_{\pm 0.02}$ |
| | | Min: | $78.78_{\pm 0.08}$ | $59.01_{\pm 5.07}$ | $52.57_{\pm 9.51}$ | $42.33_{\pm 5.2}$ | $35.13_{\pm 1.81}$ |
| | RC-DPFL | All: | $\mathbf{92.28}_{\pm 0.40}$ | $\mathbf{90.7}_{\pm 0.47}$ | $\mathbf{89.89}_{\pm 0.43}$ | $\mathbf{89.00}_{\pm 0.44}$ | $\mathbf{87.92}_{\pm 0.36}$ |
| | | Maj: | $92.06_{\pm 0.01}$ | $90.63_{\pm 1.42}$ | $\mathbf{89.95}_{\pm 0.01}$ | $\mathbf{88.78}_{\pm 0.02}$ | $\mathbf{87.76}_{\pm 0.03}$ |
| | | Min: | $\mathbf{90.45}_{\pm 0.19}$ | $\mathbf{89.33}_{\pm 0.24}$ | $\mathbf{87.44}_{\pm 0.54}$ | $\mathbf{88.21}_{\pm 0.67}$ | $\mathbf{85.83}_{\pm 0.26}$ |
| | Oracle-CDPFL | All: | $92.71_{\pm 0.27}$ | $91.06_{\pm 0.06}$ | $91.09_{\pm 0.53}$ | $89.33_{\pm 0.44}$ | $88.94_{\pm 0.47}$ |
| | | Maj: | $92.70_{\pm 0.01}$ | $91.10_{\pm 0.03}$ | $90.91_{\pm 0.02}$ | $89.10_{\pm 0.05}$ | $88.71_{\pm 1.42}$ |
| | | Min: | $91.11_{\pm 0.25}$ | $90.28_{\pm 0.02}$ | $89.87_{\pm 0.46}$ | $89.10_{\pm 0.089}$ | $88.23_{\pm 0.52}$ |

Table 5: Fairness evaluation in terms of different metrics on the split of MNIST dataset with covariate shift (rotation). The results are averaged over 3 seeds.

| | algorithm | | $\epsilon = 10$ | $\epsilon = 5$ | $\epsilon = 4$ | $\epsilon = 3$ | $\epsilon = 2$ |
|---|---|---|---|---|---|---|---|
| MNIST | DPFedAvg | $\mathcal{F}_{acc}$: | $18.41_{\pm 0.80}$ | $20.81_{\pm 2.18}$ | $20.74_{\pm 1.46}$ | $27.93_{\pm 1.68}$ | $27.57_{\pm 1.89}$ |
| | | $\mathcal{F}_{loss}$: | $1.18_{\pm 0.16}$ | $1.49_{\pm 0.26}$ | $1.54_{\pm 0.25}$ | $1.53_{\pm 0.36}$ | $1.61_{\pm 0.32}$ |
| | | Min Acc: | $66.33_{\pm 1.28}$ | $63.26_{\pm 2.25}$ | $62.58_{\pm 2.20}$ | $51.55_{\pm 2.88}$ | $51.46_{\pm 3.04}$ |
| | | Accuracy Disparity: | $18.56_{\pm 0.57}$ | $21.06_{\pm 2.26}$ | $21.00_{\pm 1.50}$ | $28.25_{\pm 1.75}$ | $27.93_{\pm 1.87}$ |
| | f-CDPFL | $\mathcal{F}_{acc}$: | $8.70_{\pm 5.52}$ | $13.39_{\pm 9.28}$ | $13.60_{\pm 3.07}$ | $10.15_{\pm 5.75}$ | $17.81_{\pm 15.77}$ |
| | | $\mathcal{F}_{loss}$: | $0.60_{\pm 0.21}$ | $0.81_{\pm 0.19}$ | $0.86_{\pm 0.11}$ | $0.57_{\pm 0.11}$ | $0.89_{\pm 0.17}$ |
| | | Min Acc: | $84.81_{\pm 4.98}$ | $78.76_{\pm 9.02}$ | $78.25_{\pm 2.71}$ | $79.75_{\pm 6.44}$ | $72.20_{\pm 15.79}$ |
| | | Accuracy Disparity: | $9.05_{\pm 5.33}$ | $13.70_{\pm 9.12}$ | $13.83_{\pm 3.07}$ | $10.21_{\pm 5.70}$ | $17.88_{\pm 15.70}$ |
| | KM-CDPFL | $\mathcal{F}_{acc}$: | $18.67_{\pm 6.54}$ | $41.43_{\pm 10.58}$ | $46.69_{\pm 9.40}$ | $46.68_{\pm 4.56}$ | $54.17_{\pm 2.67}$ |
| | | $\mathcal{F}_{loss}$: | $1.48_{\pm 0.31}$ | $2.73_{\pm 0.24}$ | $3.33_{\pm 0.23}$ | $2.33_{\pm 0.24}$ | $3.26_{\pm 0.58}$ |
| | | Min Acc: | $73.33_{\pm 6.95}$ | $48.66_{\pm 10.04}$ | $43.71_{\pm 9.25}$ | $40.58_{\pm 5.24}$ | $32.86_{\pm 2.57}$ |
| | | Accuracy Disparity: | $18.81_{\pm 6.45}$ | $41.58_{\pm 10.67}$ | $47.08_{\pm 9.58}$ | $46.98_{\pm 4.43}$ | $54.48_{\pm 2.69}$ |
| | RC-DPFL | $\mathcal{F}_{acc}$: | $\mathbf{4.04}_{\pm 0.90}$ | $\mathbf{3.13}_{\pm 0.36}$ | $\mathbf{4.27}_{\pm 0.80}$ | $\mathbf{2.95}_{\pm 0.84}$ | $\mathbf{4.40}_{\pm 1.22}$ |
| | | $\mathcal{F}_{loss}$: | $\mathbf{0.55}_{\pm 0.15}$ | $\mathbf{0.46}_{\pm 0.16}$ | $\mathbf{0.68}_{\pm 0.06}$ | $\mathbf{0.35}_{\pm 0.14}$ | $\mathbf{0.52}_{\pm 0.15}$ |
| | | Min Acc: | $\mathbf{89.46}_{\pm 0.28}$ | $\mathbf{88.71}_{\pm 0.14}$ | $\mathbf{86.91}_{\pm 0.52}$ | $\mathbf{87.43}_{\pm 0.81}$ | $\mathbf{85.23}_{\pm 0.54}$ |
| | | Accuracy Disparity: | $\mathbf{4.41}_{\pm 0.81}$ | $\mathbf{3.36}_{\pm 0.23}$ | $\mathbf{4.53}_{\pm 0.59}$ | $\mathbf{2.99}_{\pm 0.64}$ | $\mathbf{4.48}_{\pm 1.00}$ |
| | Oracle-CDPFL | $\mathcal{F}_{acc}$: | $3.35_{\pm 0.81}$ | $3.10_{\pm 0.73}$ | $2.93_{\pm 0.49}$ | $2.92_{\pm 0.75}$ | $2.85_{\pm 0.66}$ |
| | | $\mathcal{F}_{loss}$: | $0.43_{\pm 0.12}$ | $0.36_{\pm 0.08}$ | $0.43_{\pm 0.08}$ | $0.32_{\pm 0.05}$ | $0.36_{\pm 0.08}$ |
| | | Min Acc: | $90.33_{\pm 0.45}$ | $89.36_{\pm 0.37}$ | $89.13_{\pm 0.28}$ | $87.73_{\pm 0.81}$ | $87.43_{\pm 0.86}$ |
| | | Accuracy Disparity: | $3.71_{\pm 0.79}$ | $3.13_{\pm 0.51}$ | $3.21_{\pm 0.38}$ | $2.91_{\pm 0.74}$ | $2.93_{\pm 0.47}$ |

## G.2 FMNIST DATA SPLIT WITH COVARIATE SHIFT

Table 6: Average test accuracy of all (All), majority (Maj) and minority (Min) clients on the split of FMNIST dataset with covariate shift (rotation). The results are averaged over 3 seeds.

| | algorithm | | $\epsilon = 10$ | $\epsilon = 5$ | $\epsilon = 4$ | $\epsilon = 3$ | $\epsilon = 2$ |
|---|---|---|---|---|---|---|---|
| **FMNIST** | **DPFedAvg** | All: | $62.62_{\pm 0.39}$ | $62.22_{\pm 0.35}$ | $62.32_{\pm 0.93}$ | $59.58_{\pm 0.25}$ | $59.78_{\pm 0.25}$ |
| | | Maj: | $66.27_{\pm 0.05}$ | $65.86_{\pm 0.05}$ | $65.86_{\pm 0.03}$ | $62.90_{\pm 7.10}$ | $62.96_{\pm 0.03}$ |
| | | Min: | $42.32_{\pm 1.55}$ | $41.61_{\pm 1.18}$ | $44.84_{\pm 6.62}$ | $38.15_{\pm 0.73}$ | $43.23_{\pm 3.91}$ |
| | **f-CDPFL** | All: | $78.19_{\pm 1.53}$ | $74.37_{\pm 0.46}$ | $75.26_{\pm 2.38}$ | $70.10_{\pm 1.30}$ | $70.42_{\pm 3.84}$ |
| | | Maj: | $76.12_{\pm 0.05}$ | $73.43_{\pm 0.03}$ | $72.48_{\pm 0.03}$ | $67.82_{\pm 0.04}$ | $67.35_{\pm 0.06}$ |
| | | Min: | $76.21_{\pm 4.92}$ | $73.43_{\pm 6.62}$ | $\mathbf{77.89_{\pm 0.96}}$ | $\mathbf{69.85_{\pm 7.48}}$ | $69.88_{\pm 7.63}$ |
| | **KM-CDPFL** | All: | $60.17_{\pm 2.15}$ | $57.65_{\pm 1.09}$ | $57.34_{\pm 0.19}$ | $54.82_{\pm 2.95}$ | $53.55_{\pm 0.47}$ |
| | | Maj: | $57.79_{\pm 0.02}$ | $58.75_{\pm 0.03}$ | $63.22_{\pm 7.10}$ | $53.31_{\pm 0.03}$ | $58.30_{\pm 0.06}$ |
| | | Min: | $59.81_{\pm 4.34}$ | $31.72_{\pm 20.48}$ | $15.95_{\pm 2.90}$ | $32.33_{\pm 2.78}$ | $19.70_{\pm 7.14}$ |
| | **RC-DPFL** | All: | $\mathbf{80.26_{\pm 0.22}}$ | $\mathbf{78.15_{\pm 0.23}}$ | $\mathbf{77.80_{\pm 0.50}}$ | $\mathbf{74.04_{\pm 1.54}}$ | $\mathbf{74.84_{\pm 0.43}}$ |
| | | Maj: | $\mathbf{80.50_{\pm 0.02}}$ | $\mathbf{78.41_{\pm 0.02}}$ | $\mathbf{78.12_{\pm 0.06}}$ | $\mathbf{75.11_{\pm 0.06}}$ | $\mathbf{74.93_{\pm 0.04}}$ |
| | | Min: | $\mathbf{79.03_{\pm 1.57}}$ | $\mathbf{77.23_{\pm 1.30}}$ | $76.21_{\pm 1.96}$ | $69.35_{\pm 7.17}$ | $\mathbf{73.29_{\pm 1.55}}$ |
| | **Oracle-CDPFL** | All: | $80.46_{\pm 0.10}$ | $78.44_{\pm 0.18}$ | $78.31_{\pm 0.22}$ | $75.19_{\pm 0.16}$ | $75.11_{\pm 0.17}$ |
| | | Maj: | $80.76_{\pm 0.02}$ | $78.42_{\pm 0.03}$ | $78.31_{\pm 0.3}$ | $75.02_{\pm 0.05}$ | $75.21_{\pm 0.03}$ |
| | | Min: | $79.76_{\pm 0.96}$ | $78.35_{\pm 0.62}$ | $77.81_{\pm 1.02}$ | $75.33_{\pm 0.53}$ | $75.00_{\pm 1.01}$ |

Table 7: Fairness evaluation in terms of different metrics on the split of FMNIST dataset with covariate shift (rotation). The results are averaged over 3 seeds.

| | algorithm | | $\epsilon = 10$ | $\epsilon = 5$ | $\epsilon = 4$ | $\epsilon = 3$ | $\epsilon = 2$ |
|---|---|---|---|---|---|---|---|
| **FMNIST** | **DPFedAvg** | $\mathcal{F}_{acc}$: | $25.07_{\pm 2.16}$ | $24.86_{\pm 1.44}$ | $23.31_{\pm 6.22}$ | $26.53_{\pm 2.35}$ | $22.97_{\pm 4.59}$ |
| | | $\mathcal{F}_{loss}$: | $1.26_{\pm 0.05}$ | $1.02_{\pm 0.08}$ | $0.99_{\pm 0.24}$ | $0.83_{\pm 0.08}$ | $0.77_{\pm 0.18}$ |
| | | Min Acc: | $41.60_{\pm 1.55}$ | $40.69_{\pm 1.34}$ | $43.93_{\pm 6.45}$ | $37.30_{\pm 1.02}$ | $42.22_{\pm 4.08}$ |
| | | Accuracy Disparity: | $27.86_{\pm 0.98}$ | $28.11_{\pm 1.27}$ | $25.83_{\pm 5.02}$ | $29.58_{\pm 0.70}$ | $25.12_{\pm 3.31}$ |
| | **f-CDPFL** | $\mathcal{F}_{acc}$: | $11.59_{\pm 1.16}$ | $11.43_{\pm 1.22}$ | $12.00_{\pm 0.80}$ | $18.13_{\pm 2.15}$ | $12.87_{\pm 1.59}$ |
| | | $\mathcal{F}_{loss}$: | $0.32_{\pm 0.09}$ | $0.33_{\pm 0.05}$ | $0.37_{\pm 0.03}$ | $0.42_{\pm 0.07}$ | $0.33_{\pm 0.04}$ |
| | | Min Acc: | $72.31_{\pm 4.33}$ | $66.73_{\pm 2.89}$ | $70.61_{\pm 4.12}$ | $61.19_{\pm 2.26}$ | $64.93_{\pm 6.69}$ |
| | | Accuracy Disparity: | $9.36_{\pm 4.22}$ | $13.11_{\pm 2.64}$ | $9.60_{\pm 3.85}$ | $15.08_{\pm 2.01}$ | $11.08\pm 4.86$ |
| | **KM-CDPFL** | $\mathcal{F}_{acc}$: | $43.55_{\pm 2.16}$ | $59.20_{\pm 6.77}$ | $63.41_{\pm 1.25}$ | $48.46_{\pm 4.18}$ | $54.50_{\pm 6.54}$ |
| | | $\mathcal{F}_{loss}$: | $2.44_{\pm 0.25}$ | $3.36_{\pm 0.99}$ | $3.33_{\pm 0.35}$ | $1.89_{\pm 0.42}$ | $2.28_{\pm 0.40}$ |
| | | Min Acc: | $33.33_{\pm 2.08}$ | $12.41_{\pm 7.07}$ | $9.66_{\pm 0.23}$ | $21.03_{\pm 2.39}$ | $14.03_{\pm 7.47}$ |
| | | Accuracy Disparity: | $43.83_{\pm 2.48}$ | $62.78_{\pm 7.73}$ | $66.08_{\pm 0.87}$ | $51.40_{\pm 2.53}$ | $57.91_{\pm 8.10}$ |
| | **RC-DPFL** | $\mathcal{F}_{acc}$: | $\mathbf{7.37_{\pm 2.70}}$ | $\mathbf{7.25_{\pm 3.29}}$ | $\mathbf{7.73_{\pm 2.95}}$ | $\mathbf{10.55_{\pm 5.18}}$ | $\mathbf{6.92_{\pm 2.69}}$ |
| | | $\mathcal{F}_{loss}$: | $\mathbf{0.28_{\pm 0.06}}$ | $\mathbf{0.24_{\pm 0.05}}$ | $\mathbf{0.30_{\pm 0.08}}$ | $\mathbf{0.22_{\pm 0.07}}$ | $\mathbf{0.27_{\pm 0.02}}$ |
| | | Min Acc: | $\mathbf{77.31_{\pm 1.07}}$ | $\mathbf{75.16_{\pm 1.04}}$ | $\mathbf{74.03_{\pm 1.29}}$ | $\mathbf{67.66_{\pm 6.64}}$ | $\mathbf{71.76_{\pm 1.62}}$ |
| | | Accuracy Disparity: | $\mathbf{5.18_{\pm 1.43}}$ | $\mathbf{5.5_{\pm 1.14}}$ | $\mathbf{6.46_{\pm 1.10}}$ | $\mathbf{9.21_{\pm 6.10}}$ | $\mathbf{5.66_{\pm 1.29}}$ |
| | **Oracle-CDPFL** | $\mathcal{F}_{acc}$: | $7.37_{\pm 2.90}$ | $6.27_{\pm 2.96}$ | $6.50_{\pm 3.19}$ | $6.51_{\pm 3.01}$ | $6.55_{\pm 3.41}$ |
| | | $\mathcal{F}_{loss}$: | $0.24_{\pm 0.03}$ | $0.24_{\pm 0.07}$ | $0.24_{\pm 0.06}$ | $0.20_{\pm 0.08}$ | $0.20_{\pm 0.08}$ |
| | | Min Acc: | $77.70_{\pm 0.64}$ | $76.61_{\pm 0.59}$ | $75.98_{\pm 0.87}$ | $73.11_{\pm 0.33}$ | $72.71_{\pm 0.68}$ |
| | | Accuracy Disparity: | $5.05_{\pm 1.32}$ | $3.91_{\pm 0.49}$ | $4.61_{\pm 0.77}$ | $4.08_{\pm 0.02}$ | $4.75_{\pm 0.35}$ |

## G.3 CIFAR10 DATA SPLIT WITH COVARIATE SHIFT

Table 8: Average test accuracy of all (All), majority (Maj) and minority (Min) clients on the split of CIFAR10 dataset with covariate shift (rotation). The results are averaged over 3 seeds.

| | algorithm | | $\epsilon = 10$ | $\epsilon = 5$ | $\epsilon = 4$ | $\epsilon = 3$ | $\epsilon = 2$ |
|---|---|---|---|---|---|---|---|
| **CIFAR10 (covariate shift)** | **DPFedAvg** | All: | $36.50_{\pm 0.10}$ | $33.64_{\pm 0.01}$ | $33.04_{\pm 0.07}$ | $28.55_{\pm 0.01}$ | $28.12_{\pm 0.18}$ |
| | | Maj: | $37.39_{\pm 0.02}$ | $34.18_{\pm 0.02}$ | $33.69_{\pm 0.05}$ | $29.09_{\pm 0.01}$ | $29.00_{\pm 0.04}$ |
| | | Min: | $32.10_{\pm 0.03}$ | $30.03_{\pm 0.14}$ | $29.68_{\pm 0.15}$ | $25.84_{\pm 0.24}$ | $25.48_{\pm 0.47}$ |
| | **f-CDPFL** | All: | $\mathbf{40.20_{\pm 0.50}}$ | $\mathbf{35.72_{\pm 0.01}}$ | $\mathbf{35.16_{\pm 0.01}}$ | $\mathbf{30.38_{\pm 0.21}}$ | $\mathbf{29.17_{\pm 0.07}}$ |
| | | Maj: | $\mathbf{41.00_{\pm 0.04}}$ | $\mathbf{36.62_{\pm 0.02}}$ | $\mathbf{36.03_{\pm 0.05}}$ | $\mathbf{30.81_{\pm 0.08}}$ | $\mathbf{30.05_{\pm 0.11}}$ |
| | | Min: | $36.80_{\pm 1.53}$ | $31.16_{\pm 0.61}$ | $30.83_{\pm 1.32}$ | $27.58_{\pm 1.47}$ | $26.05_{\pm 0.13}$ |
| | **KM-CDPFL** | All: | $34.86_{\pm 0.23}$ | $31.46_{\pm 0.12}$ | $30.48_{\pm 0.84}$ | $26.48_{\pm 0.01}$ | $26.10_{\pm 0.08}$ |
| | | Maj: | $36.05_{\pm 0.12}$ | $32.47_{\pm 0.15}$ | $32.02_{\pm 0.04}$ | $27.16_{\pm 0.01}$ | $26.79_{\pm 0.12}$ |
| | | Min: | $27.69_{\pm 0.23}$ | $25.43_{\pm 0.08}$ | $25.20_{\pm 0.26}$ | $22.43_{\pm 0.01}$ | $21.93_{\pm 0.13}$ |
| | **RC-DPFL** | All: | $39.97_{\pm 0.09}$ | $35.20_{\pm -.25}$ | $34.23_{\pm 0.23}$ | $29.92_{\pm 0.50}$ | $27.78_{\pm 0.69}$ |
| | | Maj: | $40.52_{\pm 0.12}$ | $35.97_{\pm 0.04}$ | $35.05_{\pm 0.05}$ | $30.52_{\pm 0.05}$ | $28.80_{\pm 0.05}$ |
| | | Min: | $\mathbf{37.75_{\pm 0.10}}$ | $\mathbf{32.76_{\pm 0.26}}$ | $\mathbf{32.26_{\pm 0.54}}$ | $\mathbf{27.85_{\pm 1.12}}$ | $\mathbf{26.13_{\pm 0.05}}$ |
| | **Oracle-CDPFL** | All: | $40.43_{\pm 0.17}$ | $36.39_{\pm 0.20}$ | $35.46_{\pm 0.43}$ | $30.79_{\pm 0.19}$ | $29.34_{\pm 0.61}$ |
| | | Maj: | $41.07_{\pm 0.05}$ | $36.95_{\pm 0.05}$ | $36.47_{\pm 0.11}$ | $31.12_{\pm 0.05}$ | $30.41_{\pm 3.55}$ |
| | | Min: | $37.84_{\pm 0.32}$ | $34.49_{\pm 0.33}$ | $33.07_{\pm 0.39}$ | $29.63_{\pm 0.76}$ | $27.38_{\pm 1.12}$ |

Table 9: Fairness evaluation in terms of different metrics on the split of CIFAR10 dataset with covariate shift (rotation). The results are averaged over 3 seeds.

| | algorithm | | $\epsilon = 10$ | $\epsilon = 5$ | $\epsilon = 4$ | $\epsilon = 3$ | $\epsilon = 2$ |
|---|---|---|---|---|---|---|---|
| **CIFAR10 (covariate shift)** | **DPFedAvg** | $\mathcal{F}_{acc}$: | $9.31_{\pm 0.76}$ | $9.32_{\pm 1.06}$ | $9.81_{\pm 0.59}$ | $9.13_{\pm 2.13}$ | $9.11_{\pm 2.15}$ |
| | | $\mathcal{F}_{loss}$: | $0.16_{\pm 0.01}$ | $0.14_{\pm 0.01}$ | $0.12_{\pm 0.01}$ | $0.08_{\pm 0.01}$ | $0.07_{\pm 0.01}$ |
| | | Min Acc: | $31.65_{\pm 0.02}$ | $28.81_{\pm 0.05}$ | $28.45_{\pm 0.28}$ | $24.79_{\pm 0.19}$ | $24.65_{\pm 0.33}$ |
| | | Accuracy Disparity: | $8.57_{\pm 0.17}$ | $8.14_{\pm 0.45}$ | $8.09_{\pm 1.10}$ | $7.63_{\pm 0.81}$ | $7.07_{\pm 0.33}$ |
| | **f-CDPFL** | $\mathcal{F}_{acc}$: | $7.53_{\pm 0.73}$ | $7.90_{\pm 0.53}$ | $7.79_{\pm 0.75}$ | $7.71_{\pm 1.36}$ | $7.61_{\pm 0.9}$ |
| | | $\mathcal{F}_{loss}$: | $0.16_{\pm 0.04}$ | $0.19_{\pm 0.02}$ | $0.18_{\pm 0.05}$ | $0.12_{\pm 0.01}$ | $\mathbf{0.09_{\pm 0.02}}$ |
| | | Min Acc: | $35.97_{\pm 1.47}$ | $30.31_{\pm 0.56}$ | $29.75_{\pm 0.76}$ | $27.43_{\pm 1.04}$ | $25.01_{\pm 0.93}$ |
| | | Accuracy Disparity: | $7.71_{\pm 1.50}$ | $8.29_{\pm 1.13}$ | $7.57_{\pm 0.76}$ | $6.55_{\pm 1.30}$ | $6.89_{\pm 1.52}$ |
| | **KM-CDPFL** | $\mathcal{F}_{acc}$: | $10.88_{\pm 0.28}$ | $9.30_{\pm 0.97}$ | $8.90_{\pm 0.421}$ | $9.18_{\pm 0.61}$ | $8.76_{\pm 0.64}$ |
| | | $\mathcal{F}_{loss}$: | $0.32_{\pm 0.07}$ | $0.24_{\pm 0.01}$ | $0.22_{\pm 0.02}$ | $0.13_{\pm 0.01}$ | $0.12_{\pm 0.02}$ |
| | | Min Acc: | $26.87_{\pm 0.72}$ | $24.65_{\pm 0.08}$ | $24.19_{\pm 0.82}$ | $21.05_{\pm 0.23}$ | $20.87_{\pm 0.12}$ |
| | | Accuracy Disparity: | $14.21_{\pm 0.65}$ | $11.63_{\pm 0.07}$ | $10.79_{\pm 2.05}$ | $9.71_{\pm 0.23}$ | $9.35_{\pm 0.13}$ |
| | **RC-DPFL** | $\mathcal{F}_{acc}$: | $\mathbf{7.4_{\pm 0.68}}$ | $\mathbf{7.49_{\pm 0.74}}$ | $\mathbf{7.05_{\pm 1.10}}$ | $\mathbf{7.04_{\pm 1.76}}$ | $\mathbf{7.03_{\pm 1.17}}$ |
| | | $\mathcal{F}_{loss}$: | $\mathbf{0.10_{\pm 0.03}}$ | $\mathbf{0.10_{\pm 0.01}}$ | $\mathbf{0.10_{\pm 0.01}}$ | $\mathbf{0.10_{\pm 0.01}}$ | $0.68_{\pm 0.01}$ |
| | | Min Acc: | $\mathbf{37.29_{\pm 0.11}}$ | $\mathbf{31.95_{\pm 0.28}}$ | $\mathbf{31.37_{\pm 0.33}}$ | $\mathbf{27.81_{\pm 0.98}}$ | $\mathbf{25.37_{\pm 0.14}}$ |
| | | Accuracy Disparity: | $\mathbf{6.35_{\pm 0.34}}$ | $\mathbf{6.21_{\pm 0.56}}$ | $\mathbf{5.19_{\pm 1.15}}$ | $\mathbf{5.87_{\pm 0.48}}$ | $\mathbf{6.05_{\pm 0.84}}$ |
| | **Oracle-CDPFL** | $\mathcal{F}_{acc}$: | $6.78_{\pm 0.46}$ | $6.53_{\pm 0.61}$ | $6.43_{\pm 0.72}$ | $6.89_{\pm 0.07}$ | $6.62_{\pm 0.28}$ |
| | | $\mathcal{F}_{loss}$: | $0.10_{\pm 0.02}$ | $0.09_{\pm 0.01}$ | $0.10_{\pm 0.02}$ | $0.07_{\pm 0.01}$ | $0.08_{\pm 0.01}$ |
| | | Min Acc: | $37.41_{\pm 0.39}$ | $33.29_{\pm 0.42}$ | $31.91_{\pm 0.16}$ | $28.25_{\pm 0.76}$ | $26.13_{\pm 0.98}$ |
| | | Accuracy Disparity: | $6.01_{\pm 0.28}$ | $5.29_{\pm 0.22}$ | $4.95_{\pm 0.53}$ | $4.63_{\pm 0.65}$ | $4.75_{\pm 0.25}$ |

## G.4 CIFAR10 DATA SPLIT WITH CONCEPT SHIFT

Table 10: Average test accuracy of all (All), majority (Maj) and minority (Min) clients on the split of CIFAR10 dataset with concept shift (label flip). The results are averaged over 3 seeds.

| | algorithm | | $\epsilon = 10$ | $\epsilon = 5$ | $\epsilon = 4$ | $\epsilon = 3$ | $\epsilon = 2$ |
|---|---|---|---|---|---|---|---|
| | **DPFedAvg** | All: | $17.98_{\pm 0.31}$ | $17.19_{\pm 0.20}$ | $17.10_{\pm 0.14}$ | $17.06_{\pm 0.19}$ | $15.28_{\pm 0.27}$ |
| | | Maj: | $19.26_{\pm 0.02}$ | $18.17_{\pm 0.02}$ | $17.85_{\pm 0.11}$ | $17.80_{\pm 0.19}$ | $15.86_{\pm 0.25}$ |
| | | Min: | $12.51_{\pm 0.58}$ | $12.86_{\pm 0.32}$ | $12.56_{\pm 0.43}$ | $12.65_{\pm 0.40}$ | $11.55_{\pm 0.88}$ |
| | **f-CDPFL** | All: | $35.27_{\pm 2.83}$ | $33.21_{\pm 0.26}$ | $31.74_{\pm 1.56}$ | $29.70_{\pm 1.84}$ | $\mathbf{25.18_{\pm 1.32}}$ |
| | | Maj: | $35.12_{\pm 3.30}$ | $33.56_{\pm 0.02}$ | $\mathbf{32.42_{\pm 0.92}}$ | $30.11_{\pm 2.22}$ | $\mathbf{25.89_{\pm 0.65}}$ |
| CIFAR10 (concept shift) | | Min: | $\mathbf{36.17_{\pm 0.37}}$ | $32.12_{\pm 0.21}$ | $27.68_{\pm 5.58}$ | $\mathbf{27.25_{\pm 4.70}}$ | $\mathbf{20.95_{\pm 5.37}}$ |
| | **KM-CDPFL** | All: | $19.67_{\pm 0.70}$ | $17.80_{\pm 0.30}$ | $17.98_{\pm 0.58}$ | $17.70_{\pm 0.67}$ | $15.50_{\pm 0.30}$ |
| | | Maj: | $20.95_{\pm 0.72}$ | $19.49_{\pm 0.03}$ | $19.27_{\pm 0.65}$ | $18.99_{\pm 0.80}$ | $16.44_{\pm 0.30}$ |
| | | Min: | $12.01_{\pm 0.91}$ | $10.46_{\pm 0.55}$ | $10.31_{\pm 0.46}$ | $9.91_{\pm 0.52}$ | $9.89_{\pm 1.01}$ |
| | **RC-DPFL** | All: | $\mathbf{37.16_{\pm 0.26}}$ | $\mathbf{33.41_{\pm 0.49}}$ | $\mathbf{32.12_{\pm 0.44}}$ | $\mathbf{30.25_{\pm 1.45}}$ | $23.85_{\pm 1.40}$ |
| | | Maj: | $\mathbf{37.45_{\pm 0.23}}$ | $\mathbf{33.62_{\pm 0.02}}$ | $32.39_{\pm 0.45}$ | $\mathbf{30.98_{\pm 0.91}}$ | $24.72_{\pm 0.80}$ |
| | | Min: | $35.38_{\pm 0.44}$ | $\mathbf{32.22_{\pm 0.60}}$ | $\mathbf{30.54_{\pm 0.51}}$ | $25.90_{\pm 4.70}$ | $18.61_{\pm 5.20}$ |
| | **Oracle-CDPFL** | All: | $37.40_{\pm 0.30}$ | $33.93_{\pm 0.20}$ | $32.98_{\pm 0.31}$ | $32.36_{\pm 0.24}$ | $26.39_{\pm 0.23}$ |
| | | Maj: | $37.64_{\pm 0.27}$ | $33.81_{\pm 0.05}$ | $33.20_{\pm 0.28}$ | $32.67_{\pm 0.28}$ | $26.59_{\pm 0.12}$ |
| | | Min: | $35.98_{\pm 0.56}$ | $32.91_{\pm 0.28}$ | $31.65_{\pm 0.45}$ | $30.56_{\pm 0.12}$ | $25.16_{\pm 0.88}$ |

Table 11: Fairness evaluation in terms of different metrics on the split of CIFAR10 dataset with concept shift (label flip). The results are averaged over 3 seeds.

| | algorithm | | $\epsilon = 10$ | $\epsilon = 5$ | $\epsilon = 4$ | $\epsilon = 3$ | $\epsilon = 2$ |
|---|---|---|---|---|---|---|---|
| | **DPFedAvg** | $\mathcal{F}_{acc}$: | $12.14_{\pm 0.92}$ | $9.25_{\pm 1.55}$ | $10.03_{\pm 1.38}$ | $9.59_{\pm 2.26}$ | $8.59_{\pm 0.99}$ |
| | | $\mathcal{F}_{loss}$: | $0.29_{\pm 0.01}$ | $0.24_{\pm 0.03}$ | $0.21_{\pm 0.02}$ | $0.20_{\pm 0.01}$ | $0.13_{\pm 0.02}$ |
| | | Min Acc: | $11.11_{\pm 0.45}$ | $11.63_{\pm 0.25}$ | $11.19_{\pm 0.31}$ | $11.55_{\pm 0.76}$ | $11.17_{\pm 0.86}$ |
| | | Accuracy Disparity: | $11.89_{\pm 0.27}$ | $9.49_{\pm 0.05}$ | $10.53_{\pm 1.06}$ | $9.83_{\pm 1.62}$ | $6.95_{\pm 1.82}$ |
| | **f-CDPFL** | $\mathcal{F}_{acc}$: | $11.40_{\pm 4.66}$ | $8.38_{\pm 0.34}$ | $9.44_{\pm 1.13}$ | $11.54_{\pm 5.45}$ | $9.54_{\pm 0.98}$ |
| | | $\mathcal{F}_{loss}$: | $0.14_{\pm 0.07}$ | $0.08_{\pm 0.01}$ | $0.16_{\pm 0.12}$ | $0.22_{\pm 0.09}$ | $\mathbf{0.09_{\pm 0.05}}$ |
| | | Min Acc: | $32.15_{\pm 4.49}$ | $\mathbf{31.07_{\pm 0.17}}$ | $26.69_{\pm 5.44}$ | $23.45_{\pm 4.48}$ | $\mathbf{19.97_{\pm 5.13}}$ |
| CIFAR10 (concept shift) | | Accuracy Disparity: | $7.33_{\pm 4.44}$ | $4.26_{\pm 0.17}$ | $8.45_{\pm 5.56}$ | $11.01_{\pm 4.73}$ | $8.59_{\pm 5.34}$ |
| | **KM-CDPFL** | $\mathcal{F}_{acc}$: | $20.99_{\pm 1.67}$ | $16.66_{\pm 2.41}$ | $16.41_{\pm 2.92}$ | $15.95_{\pm 3.23}$ | $11.15_{\pm 2.63}$ |
| | | $\mathcal{F}_{loss}$: | $0.59_{\pm 0.07}$ | $0.46_{\pm 0.01}$ | $0.47_{\pm 0.02}$ | $0.45_{\pm 0.03}$ | $0.24_{\pm 0.01}$ |
| | | Min Acc: | $10.61_{\pm 1.30}$ | $9.43_{\pm 0.61}$ | $9.25_{\pm 0.39}$ | $8.89_{\pm 0.66}$ | $9.07_{\pm 1.103}$ |
| | | Accuracy Disparity: | $21.35_{\pm 2.44}$ | $17.85_{\pm 1.47}$ | $18.21_{\pm 2.03}$ | $18.71_{\pm 2.63}$ | $12.75_{\pm 1.81}$ |
| | **RC-DPFL** | $\mathcal{F}_{acc}$: | $\mathbf{8.09_{\pm 1.18}}$ | $\mathbf{8.20_{\pm 0.20}}$ | $\mathbf{7.88_{\pm 0.70}}$ | $\mathbf{8.37_{\pm 1.08}}$ | $\mathbf{8.03_{\pm 0.03}}$ |
| | | $\mathcal{F}_{loss}$: | $0.09_{\pm 0.01}$ | $0.07_{\pm 0.02}$ | $0.06_{\pm 0.01}$ | $0.16_{\pm 0.08}$ | $0.12_{\pm 0.07}$ |
| | | Min Acc: | $\mathbf{34.69_{\pm 0.11}}$ | $30.97_{\pm 0.82}$ | $\mathbf{29.45_{\pm 0.71}}$ | $24.57_{\pm 4.99}$ | $17.61_{\pm 5.09}$ |
| | | Accuracy Disparity: | $\mathbf{4.45_{\pm 0.79}}$ | $\mathbf{4.21_{\pm 0.08}}$ | $\mathbf{5.06_{\pm 0.81}}$ | $9.53_{\pm 5.49}$ | $10.07_{\pm 4.82}$ |
| | **Oracle-CDPFL** | $\mathcal{F}_{acc}$: | $8.77_{\pm 1.29}$ | $8.88_{\pm 0.38}$ | $8.00_{\pm 0.44}$ | $7.80_{\pm 0.45}$ | $8.28_{\pm 0.46}$ |
| | | $\mathcal{F}_{loss}$: | $0.09_{\pm 0.01}$ | $0.07_{\pm 0.01}$ | $0.06_{\pm 0.01}$ | $0.08_{\pm 0.01}$ | $0.05_{\pm 0.01}$ |
| | | Min Acc: | $34.83_{\pm 0.63}$ | $30.59_{\pm 0.32}$ | $30.91_{\pm 0.53}$ | $29.63_{\pm 0.30}$ | $23.83_{\pm 0.96}$ |
| | | Accuracy Disparity: | $4.97_{\pm 0.88}$ | $4.91_{\pm 0.22}$ | $4.49_{\pm 0.21}$ | $5.17_{\pm 0.51}$ | $5.17_{\pm 0.96}$ |

## H    PRIVACY GUARANTEES OF RC-DPFL

The privacy guarantee of RC-DPFL for each client $i$ in the system comes from the fact that the client runs DPSGD with a fixed DP noise variance $\sigma_{i,\text{DP}}^2 = c^2 \cdot z_i^2(\epsilon, \delta, b_i^1, b_i^{>1}, N_i, K, E)$ in each of its batch gradient computations. We provide a formal privacy guarantee for the algorithm to show the record-level DP privacy guarantees provided to each client $i$ with respect to its local dataset $\mathcal{D}_i$.

**Theorem H.1.** *The set of model updates $\{\Delta\tilde{\boldsymbol{\theta}}_i^e\}_{e=1}^E$, which are uploaded to the server by each client $i \in \{1, \cdots, n\}$ during the training time, satisfies $(\epsilon, \delta)$-DP with respect to the client's local dataset $\mathcal{D}_i$, where the parameters $\epsilon$ and $\delta$ depend on the amount of DP noise $\sigma_{i,\text{DP}}^2$ used by the client.*

*Proof.* We remember that the sensitivity of the batch gradient in Equation (1) to every data sample is $c$. Therefore, based on Proposition 7 in (Mironov, 2017) each of the batch gradient computations by client $i$ (in the first round $e = 1$ as well as the next rounds $e > 1$) is $(\alpha, \frac{\alpha c^2}{2\sigma_{i,\text{DP}}^2})$-RDP w.r.t the local dataset $\mathcal{D}_i$. Therefore, if the client runs $E_i^{\text{tot}}$ total number of gradient updates during the training time, which results in the model updates $\{\Delta\tilde{\boldsymbol{\theta}}_i^e\}_{e=1}^E$ uploaded to the server, the set of model updates will be $(\alpha, \frac{E_i^{\text{tot}}\alpha c^2}{2\sigma_{i,\text{DP}}^2})$-RDP w.r.t $\mathcal{D}_i$, according to Proposition 1 in (Mironov, 2017). Finally, according to Proposition 3 in the same work, this guarantee is equivalent to $(\frac{E_i^{\text{tot}}\alpha c^2}{2\sigma_{i,\text{DP}}^2} + \frac{log(1/\delta)}{\alpha-1}, \delta)$-DP (for any $\delta > 1$). The RDP-based guarantee is computed over a bunch of orders $\alpha$ and the best result among them is chosen as the privacy guarantee. Therefore, the proof is complete and the set $\{\Delta\tilde{\boldsymbol{\theta}}_i^e\}_{e=1}^E$ satisfies $(\epsilon, \delta)$-DP w.r.t $\mathcal{D}_i$, with $\epsilon = \frac{E_i^{\text{tot}}\alpha c^2}{2\sigma_{i,\text{DP}}^2} + \frac{log(1/\delta)}{\alpha-1}$ derived above, and $\delta > 0$. Tighter bounds for $\epsilon$ can be derived by using the numerical procedure, proposed in (Mironov et al., 2019), for accounting sampled Gaussian mechanism. □

As a complement to the above theorem, which focuses on the privacy guarantee of the uploaded model updates, we note that the local model selection by clients in the third stage of RC-DPFL has also an effectively zero "record-level" privacy leakage. We explain about this important point as follows. The sensitivity of a client's "average train loss value" (which is not shared with the server) to the addition or removal of a sample in the client's local dataset is relatively small, as this value is resulted from an "averaging over the whole dataset". However, even if the "average train loss value" of a client changes slightly as a result of addition or removal of a sample (a record-level change), the client's selected model remains the same, especially in the third stage of RC-DPFL that one of the $M$ existing cluster models results in a smaller loss by a large margin. Therefore, in the third stage, the sensitivity of the local model selection of a client to a record-level change is effectively zero (in contrast to the sensitivity of the average train loss, which was "small"). This is true especially for our algorithm, which transitions to a loss-based strategy after some training progress in the first $E_c$ rounds. Our claim of "effectively zero record-level privacy leakage of local model selections" is an immediate consequence of the their "zero sensitivity to record-level changes" discussed above. Please remember that the Definition 3.1 considered in this work is concerned with "record-level" changes.

## I    GRADIENT ACCUMULATION

When training large models with DPSGD, increasing the batch size results in memory exploding during training or finetuning. This might happen even when we are not using DP training. On the other hand, using a small batch size results in larger stochastic noise in batch gradients. Also, in the case of DP training, using a small batch size results in fast increment of DP noise (as explained in Lemma 4.1 in details). Therefore, if the memory budget of devices allow, we prefer to avoid using small batch sizes. But what if there is a limited memory budget? A solution for virtually increasing batch size is "gradient accumulation", which is very useful when the available physical memory of GPU is insufficient to accommodate the desired batch size. In gradient accumulation, gradients are computed for smaller batch sizes and summed over multiple batches, instead of updating model parameters after computing each batch gradient. When the accumulated gradients reach the target logical batch size, the model weights are updated with the accumulated batch gradients. The page in https://opacus.ai/api/batch_memory_manager.html shows the implementation of gradient accumulation for DP training.