# OpenReview forum: "A Provably Robust Algorithm for Differentially Private Clustered Federated Learning"
_ICLR.cc/2025/Conference — Submitted to ICLR 2025_

### Official Review · Reviewer_CYaZ · 2024-10-15

**Soundness:** 2
**Presentation:** 2
**Contribution:** 2
**Rating:** 6
**Confidence:** 3

**Summary:**

The authors propose RC-DPFL, a robust clustering algorithm to address performance fairness under privacy constraints. The algorithm uses client clustering based on both model updates and training loss values to mitigate the negative impacts of DP noise. The authors provide theoretical results to justify their claims and conduct extensive experiments. RC-DPFL is empirically effective in mitigating the disparate impact of DP noise.

**Strengths:**

1. The empirical results are promising. Namely, RC-DPFL performs close to the oracle algorithm in terms of accuracy, fairness, and clustering accuracy.
2. The authors provide interesting theoretical results on how large batch sizes can improve clustering accuracy and reduce noise.
3. The algorithm is evaluated across various data distributions and privacy budgets, demonstrating its ability to handle heterogeneous datasets while maintaining computational efficiency and fairness.

**Weaknesses:**

1. The article is not well organized. I cannot find the appendix authors mentioned in the main content. Namely, the theoretical proofs, details of the experiment implementation, table 1, table 10, and table 11 are missing. As a result, reviewers cannot verify theoretical claims and experimental results.

2. Assumption 3.2 is confusing. Please modify the statements if there is any typo/error. How could the gradient variance upper bound $\sigma_{i,g}^2$ decrease when we use a smaller batch size $b$?

   - Does Assumption 3.2 contradicts Lemma 4.1?

   - Which theoretical result requires Assumption 3.2? It's unclear from the paper.

   - Is there any other assumption not mentioned in the main content? The proof of FL convergence typically requires some standard assumptions. For example, some FL papers require smoothness and bounded gradients (Assumption 1, 2, 3 in Reddi et al., 2021).

     Reference: “Adaptive Federated Optimization” by Sashank Reddi et al., ICLR 2021.

3. In figure 1, the motivation of using smaller batch sizes in the third stage is unclear. Can we use a fixed batch size throughout the training process?

   - In real implementations of (centralized) DP-SGD, people usually use very large batch sizes to improve the performance. Why should the clients switch to a smaller batch size?

     Reference: "Unlocking High-Accuracy Differentially Private Image Classification through Scale" by Soham De et al.

   - The authors need to provide more details about the experiment implementations. Switching the batch size changes the sample rate of the training data. This could lead to some issues on the privacy composition. To the best of my knowledge, this functionality is unavailable in popular python DP libraries like Opacus or fastDP by awslabs.

4. The server hard clusters clients based on their loss values after $E_c$ rounds. The local loss function of a client is privacy sensitive as it is a function of the local training data. What kind of privacy mechanism is applied to ensure the privacy guarantee when sharing the local loss values? How does privacy accounting work in RC-DPFL given that different statistics (gradient updates, loss values) are shared?

5. One limitation is that the algorithm assumes the number of client clusters is known beforehand, which may not always be the case in real-world applications. This could reduce the applicability of the algorithm without prior knowledge or estimation of clusters.

For these weaknesses/reasons, I recommend **marginally below the acceptance threshold** at this stage. The paper can benefit from a round of revision. I encourage the authors to provide all reviewers with the missing contents in the rebuttal phase.

**Questions:**

1. Could you please explain the results in figure 3(c) and 5(c)? For example, why are the success rates of f-DPFL non-monotonic as we increase $\epsilon$? Why do the success rates suddenly drop to zero in ($epsilon=5$, covariate shift, CIFAR-10), ($\epsilon=4$, MNIST), and ($\epsilon=3,5$, FMNIST)?

2. Could you please explain the intuition of Lemma 4.1? Previous theoretical results also suggest that the variance of the stochastic gradient estimator is a decreasing function of $b$. For deep neural networks with L2 loss we show that the variance of the gradient is a polynomial in $1/b$ (Qian et al., 2020). Is this different from Lemma 4.1? Does Lemma 4.1 consider different models other than deep neural networks?

      Reference: "The Impact of the Mini-batch Size on the Variance of Gradients in Stochastic Gradient Descent" by Xin Qian, Diego Klabjan.

---

> ### Author Response · Authors · 2024-11-22
> **Response to Reviewer CYaZ**
>
> Thank you for taking the time to give multiple comments/questions. They were useful for improving our work. We have addressed them all one by one. You can find the edited manuscript containing the changed/added parts highlighted in blue.
>
> >**Q1**: I cannot find the appendix authors mentioned in the main content.
>
> **A1**: We are not sure why you are unable to find the appendix, but we would like to remind you that it was uploaded as a separate PDF file as supplementary material during submission. We have done so for the revised draft too. All the results you mentioned are included there.
>
> >**Q2**: Assumption 3.2 is confusing. Please modify the statements if there is any typo/error.
>
> **A2**: Thanks for pointing out the typo in the assumption. It has been corrected to state that "the larger the batch size, the smaller the variance."
>
>
> >**Q3**: In figure 1, the motivation of using smaller batch sizes in the third stage is unclear. Can we use a fixed batch size throughout the training process?
>
> **A3**: We have thoroughly discussed the effect of  $b_i^{>1}$. For details, please refer to the paragraph after lemma 4.1, the new figure 3, the third paragraph of section 4.4, and appendix F.1 (in order). In summary, unlike $b_i^{1}$, we need to keep $b_i^{>1}$ to small values, as increasing it makes the GMM clustering at the end of the first round more challenging, and as a consequence, it results in noisier model updates from clients at the end of the first round. Note that, unlike centralized settings, in "clustered" FL, performing a correct client clustering is a prerequisite for training progress in the next rounds. Therefore, reducing the noise level in the model updates of the first round, is of great importance.
>
> >**Q4**: Switching the batch size changes the sample rate of the training data. This could lead to some issues on the privacy composition.
>
> **A4**: We are using the Renyi DP (RDP) privacy accountant (its TensorFlow Privacy implementation) for accounting the privacy in our experiments. RDP is well known for providing a tighter analysis of differential privacy guarantees of the composition
> of "heterogeneous" DP mechanisms on the same dataset. More specifically, it can  be used for deriving the privacy guarantees of composition of multiple ($\epsilon, \delta$)-DP mechanisms but with different values of  ($\epsilon, \delta$). Please have a quick look at the abstract of this paper:
>
> I. Mironov, "Renyi Differential Privacy", 2017.
>
> In our proposed algorithm, this heterogeneity happens between the round $e=1$ and the next rounds $e>1$. However, as the local trainings are done on the same fixed datatset, RDP accountant can handle it. We have added more details about this in appendix B.3.
>
> >**Q5**: The server hard clusters clients based on their loss values after $E_c$ rounds. The local loss function of a client is privacy sensitive.
>
> **A5**: We have clarified this in the revised draft by updating line 17 and 19 of Algorithm 1 to eliminate the ambiguity. The loss-based clustering of clients is performed "locally" on the clients' side, without sharing any loss values with the server. As detailed in lines 17 and 19 of the algorithm, the server broadcasts all updated cluster models to the clients, who select the one with the lowest loss and update it "locally" using DPSGD. Afterward, the clients return all the models, with one updated, to the server, which then uses the modified model. Thus, there are no privacy concerns when switching the clustering strategy.
>
> >**Q6**: One limitation is that the algorithm assumes the number of client clusters is known beforehand.
>
> **A6**: We thank the reviewer for raising this important question. In fact, after the first submission, we developed a novel mechanism for determining the true number of clusters, "$M$". This new contribution significantly improves the applicability of our method for real world cases. In the revised draft, we have addressed this in detail. Please refer to Lemma 4.2 and the paragraph after it, section 4.2.2, the first paragraph of section 4.4, and also Appendix F.2 (in order), where we propose a method for finding the true number of clusters when it is unknown. Additionally, we have added Line 9 to Algorithm 1 to make the approach more general.
>
> In brief, when the true number of clusters is not known, we choose the value of “$M$” that results in the maximum “confidence” for the learned GMM. This idea, developed after the submission deadline, has greatly enhanced the applicability of our work. Our experiments show that this method successfully predicts the true number of clusters in nearly all cases, with only 3 failures out of almost 80 experiments. (which will be updated in the final version of the paper). Therefore, the current results in the paper reflect the outcomes we would have obtained had we used this method during the original submission. Notably, no existing baseline algorithms offer such a simple and effective approach for determining “$M$”

---

> > ### Author Response · Authors · 2024-11-22
> > **Response to Reviewer CYaZ**
> >
> > >**Q7**: Could you please explain the results in figure 3(c) and 5(c)?
> >
> > **A7**: This is indeed one of the limitations of existing algorithms, as discussed at the beginning of Section 4. These algorithms do not guarantee the discovery of the true clusters of clients, which holds true even in non-DP systems that do not involve any DP noise. This issue is further explored in the following paper:
> >
> > https://openreview.net/pdf?id=B0uBSSUy0G.
> >
> > Therefore, the success rate of the f-CDPFL algorithm in detecting the true clusters is influenced not only by the value of the $\epsilon$ parameter but also by the model initializations. However, it is indeed true that a smaller value of the parameter will likely make clustering more challenging for f-CDPFL. To ensure a more comprehensive evaluation, we will run the current experiments using more different seeds which will allow us to observe the impact of this parameter more clearly across a larger number of runs of the algorithm.
> >
> >
> > >**Q8**: Could you please explain the intuition of Lemma 4.1?
> >
> > **A8**: We guess you mean "assumption 3.2" (and not the lemma 4.1). We thank for pointing out. There was a typo in the assumption, which we have fixed in the revised draft (line 161): the larger the batch size, the smaller the stochastic noise in the batch gradients.

---

> ### Comment · Reviewer_CYaZ · 2024-11-23
>
> Thanks for the authors' response. Most of my concerns have been addressed. I've increased the score accordingly.

---

> > ### Author Response · Authors · 2024-11-24
> > **Response to Reviewer CYaZ**
> >
> > We are happy to see that our responses have addressed your concerns. We also thank you for increasing the score.
> >
> >
> > Thank you,
> > Authors

---

### Official Review · Reviewer_EpDE · 2024-10-28

**Soundness:** 1
**Presentation:** 2
**Contribution:** 2
**Rating:** 3
**Confidence:** 4

**Summary:**

The paper looks at the problem of differentially private (DP) federated learning (FL) when the clients have heterogeneous data sets. The authors propose to use a Gaussian mixture model (GMM) on the server side to cluster similar clients to mitigate the issues when updating the global model. The authors note that the uncertainty in the clustering is highest during the first update and the client updates get smaller during training, and therefore propose a multi-stage method that i) uses full batches and soft clustering on client model updates on the 1st epoch, ii) then does smaller batches with hard-clustering still on client model updates for some epochs, iii) and finally switches to using hard clustering on client losses towards the end of the training. The authors present some theoretical motivation for the different stages, and empirically show that their method outperforms their own baselines on several image recognition tasks.

**Strengths:**

i) Increasing the robustness of FL w.r.t. client heterogeneity is an important topic.

ii) Including DP to the clustered FL approach, while not completely novel, seems like a potentially sound direction.

iii) The paper is mostly clear-enough to read, although it could be improved still.

**Weaknesses:**

i) Some of the claimed novel results are very close to published results (Lemma 4.1, see Questions below for details).

ii) I doubt that some of the claims about DP hold (see Questions below for details).

iii) All the experiments are run assuming that the true number of clusters is known. There are also no experiments using data with inherent splits, only with simulations.

iv) It is not clear what effect different design choices have on the results, as there is no ablation study (e.g., changing batch size after initial update, switching from soft to hard clustering later, using data augmentations or not, etc.).

v) There are no baselines using other approach beyond clustering aimed at addressing heterogeneity in DPFL (see, e.g., Shen et al. 2023, Silva et al. 2022,  Yang et al. 2023).

vi) Most results do not report any deviation measure besides, e.g., the mean.

**Questions:**

### Update after the discussion

I still recommend rejecting the paper: as detailed in the discussion, the main issue is that while the proposed method is claimed to be DP, I think there is a clear violation of formal DP guarantees in the proposed method (the privacy leakage from clustering based directly on the (non-DP) local loss is not accounted for in the privacy budget, as acknowledged by the authors). This is not acceptable.

### Original comments

Questions and comments In decreasing order of importance:

1) Alg 1, line 21: when switching to loss-based clustering, how is DP guaranteed (or if this is required)? I do not seem to find any details on how exactly this part works.

2) Lemma 4.1: the actual content as well as the proofs seem to be very close to the results stated in Sec 3 and the corresponding proofs in Malekmohammadi et al. 2024 (compare e.g. their Appendix C, eqs 12 & 13 to your Appendix D eqs 8 & 9). Are you aware of this work? The general result is also closely related to Räisä et al. 2024.

3) Sec 4.3.1: I do not understand why using data augmentations would *improve* per-instance (=per sample) DP guarantees. Looking at releasing the sum of gradients at client $i$ for a single iteration, say we use clipping constant $C$, and add $|\tau|$ augmentations for each sample. Then I would claim that the sensitivity for the sum is $C |\tau|$ (as the worst-case effect of changing a single sample in the data results in all the corresponding augmentations to also change), not $C$ as you state (as we are not interested in protecting a single augmentation but the actual sample). Do I misread this somehow?

4) How are all the hyperparameters tuned for the proposed method and for the baselines? Please include a proper description of this at least to the Appendix. Also mention other relevant details, like if and when exactly you use data augmentations, etc.

5) Sec 5.1, on the baselines: please state somewhere how exactly do you implement DP version of loss based clustering methods (as the loss values as such are sensitive, you need to add DP somewhere, i.e., you need extra DP mechanism to release also the loss besides the local weights; this is related to issue 1)).

6) All experiments: please also clearly report $\delta$ besides $\epsilon$ for all DP runs.

7) Def 3.1: Please also state explicitly which neighbourhood definition you use with DP (e.g. add/remove).

8) Lines 136-140: Do you actually use the classical Gaussian mechanism for something, if all accounting is done using RDP?

### Minor issues (no need to acknowledge or comment on these)
* Please fix typos on lines: 117, 144-45 (differing datasets are the same?), 187-88 (should be larger batch size or variance?)
* Lines 51-53: On the disparate impact of DP: there is also contrary evidence that DP has in some sense only a limited impact on fairness (e.g. Berrada et al. 2023, Mangold et al. 2023)

### References:

Berrada et al. 2023: Unlocking Accuracy and Fairness in Differentially Private Image Classification.

Malekmohammadi et al. 2024: Noise-Aware Algorithm for Heterogeneous Differentially Private Federated Learning.

Mangold et al. 2023: Differential Privacy has Bounded Impact on Fairness in Classification.

Räisä et al. 2024: Subsampling is not Magic.

Shen et al. 2023: Share your representation only.

Silva et al. 2022: FedEmbed: Personalized Private Federated Learning.

Yang et al. 2023: PRIVATEFL: Accurate, Differentially Private Federated Learning
via Personalized Data Transformation.

**Details Of Ethics Concerns:**

I have some questions on attributing existing research as detailed in the questions (some claimed novel results are very close to existing results that are not cited in any way, including on the level of actual writing and notation).

---

> ### Author Response · Authors · 2024-11-22
> **Response to Reviewer EpDE**
>
> We would like to thank the reviewer for the constructive and timely feedback, which shows his precision in reviewing our work. The edited draft contains the edited/added parts highlighted in blue.
>
> >**Q1**: Alg 1, line 21: when switching to loss-based clustering, how is DP guaranteed (or if this is required)?
>
> **A1**: To address the reviewer's concern, we have edited line 17 of Algorithm 1 in the revised draft for clarification. To clarify, loss-based clustering is performed "locally" on the clients’ side to ensure no loss values are shared with the server and preserve data privacy. As detailed in lines 17 and 19 of the algorithm, the server broadcasts all updated cluster models to the clients in each round after $E_c$. Each client then selects the model with the lowest loss "locally", updates it using DPSGD, and returns all models (including the updated one) to the server. The server then uses the modified model. This approach ensures that there are no privacy concerns when switching the clustering strategy. We believe this clarification resolves the issue and further strengthens our method.
>
> >**Q2:** Lemma 4.1: the actual content as well as the proofs seem to be very close to the results stated in Sec 3 and the corresponding proofs in Malekmohammadi et al. 2024.
>
> **A2**: We thank the reviewer for pointing out these works. We are indeed aware of them and included them in our revised paper. While our lemma addresses the case with two stages during training, we acknowledge the similarities to the result in Malekmohammadi et al. (2024) and have cited this work in the revised draft. However, we want to emphasize that the main contribution of our paper is not this lemma. Instead, we use it as a foundational step in developing our novel clustering strategy. This strategy is supported by the theoretical results in Lemma 4.2 and analyzed in Theorem 4.3, which are both new and specific to our proposed approach. These contributions are central to our work and distinguish our method from prior research.
>
> >**Q3**: All the experiments are run assuming that the true number of clusters is known. There are also no experiments using data with inherent splits, only with simulations.
>
> **A3**: We thank the reviewer for raising this important question. In fact, after the first submission, we developed a novel mechanism for determining the true number of clusters, "M". This new contribution significantly improves the applicability of our paper for real world cases. In the revised draft, we have addressed this in detail. Please refer to Lemma 4.2 and the paragraph after it, section 4.2.2, the first paragraph of section 4.4, and also Appendix F.2 (in order), where we propose a method for finding the true number of clusters when it is unknown. Additionally, we have added Line 9 to Algorithm 1 to make the approach more general.
>
> In brief, when the true number of clusters is not known, we choose the value of “M” that results in the maximum “confidence” for the learned GMM. This idea, developed after the submission deadline, has greatly enhanced the applicability of our work. Our experiments show that this method successfully predicts the true number of clusters in nearly all cases, with only 3 failures out of almost 80 experiments. (which will be updated in the final version of the paper). Therefore, the current results in the paper reflect the outcomes we would have obtained had we used this method during the original submission. Notably, no existing baseline algorithms offer such a simple and effective approach for determining “M.”
>
> While the time constraints during the rebuttal period prevented us from running experiments on additional datasets, we are committed to conducting new experiments before the final decision. We would appreciate it if you could clarify what you mean by “inherent splits.” Based on our knowledge, there is no publicly available dataset with such characteristics. Could you recommend a dataset that would help us address your comment before the decision time? Given our focus on clustered FL, the dataset would need to have multiple clients/clusters with significant data heterogeneity across them.
>
> >**Q4**: I do not understand why using data augmentations would improve per-instance (=per sample) DP guarantees.
>
> **A4**: We agree with the reviewer’s comment that the way we initially discussed data augmentation could lead to privacy leakage. While "batch augmentation" [1] preserves privacy guarantees, we were aware of its limitations for our specific problem, which requires a *large batch size*.
>
> Considering this, we have removed the paragraph discussing the "potential" use of data augmentation in the revised draft. Furthermore, we want to clarify that our current experimental results do not use any form of data augmentation. (please continue to the following)

---

> ### Author Response · Authors · 2024-11-22
> **Response to Reviewer EpDE**
>
> Instead, we used a full batch size equal to the actual dataset size, which has been effective, and the results reported in the submission reflect this approach.
>
> [1]: E. Hoffer, et. al.,”Augment Your Batch: Improving Generalization Through Instance Repetition”, CVPR 2020.
>
> >**Q5**: How are all the hyperparameters tuned for the proposed method and for the baselines?
>
> **A5**: We have addressed this comment by adding a detailed section for tuning hyperparameters in sections B.3, B.4, B.5 and B.6 of the revised draft. The sections provide detailed results about setting the hyper-parameters of both RC-DPFL and the baselines.
>
> >**Q6**: It is not clear what effect different design choices have on the results.
>
> **A6**: We have provided a detailed discussion on the effects of $b_i^{>1}$ and $E_c$ in the revised draft.
>
> For $b_i^{>1}$, please refer to the paragraph after Lemma 4.1, the new Figure 3, the third paragraph of Section 4.4, and Appendix F.1 (in order). In summary, unlike $b_i^{1}$, we keep $b_i^{>1}$ at small values. Increasing it makes the GMM clustering at the end of the first round more challenging by introducing additional noise into the initial model updates from the clients, as increasing $b_i^{>1}$ results in a larger noise scale (z) during the entire training time (see equation 2).
>
> For $E_c$, please refer to the paragraph after Lemma 4.2, Section 4.2.2, and the second paragraph of Section 4.4. We have also added Line 11 to Algorithm 1 to provide more clarity on how we set the switching time parameter $E_c$.
> Lastly, we have removed the paragraph on data augmentation, and as previously mentioned, it has not been used in our current results.
>
>
> >**Q7**: There are no baselines using other approach beyond clustering.
>
> **A7**: We appreciate the reviewer’s suggestion, and we would like to emphasize that in the literature, clustered federated learning (CFL) is widely regarded as a personalization method specifically designed for scenarios with “highly heterogeneous” data splits. In such cases, a single global model or “mild” personalization techniques often fail due to the pronounced heterogeneity across clusters. This point is supported by the findings of M. Werner et al., "Provably Personalized and Robust Federated Learning", TMLR 2023, which demonstrated that Ditto, a personalization method, performs poorly under high heterogeneity scenarios with data splits similar to ours. For example, when there is a concept shift (i.e.,
> P(y|x) changes significantly across clusters), there is no shared knowledge between clusters to facilitate learning, which makes general personalization techniques ineffective. In such cases, CFL algorithms are inherently better suited as they address these challenges directly. Our proposed problem naturally aligns with the need for CFL algorithms, especially those robust to DP noise, which is essential in our framework.
>
> However, as noted, we believe that the suggested methods are likely to perform poorly in our setup due to the extreme data heterogeneity, where existing CFL methods already demonstrate significant challenges. Furthermore, our current set of baselines represents much stronger benchmarks for comparison to ensure a rigorous evaluation of our approach.
>
> While the limited time during the rebuttal period did not allow us to run experiments on the mentioned algorithms, we plan to evaluate them thoroughly before the final decision and include the results in the final version of the paper.
>
> >**Q8**: Most results do not report any deviation measure besides, e.g., the mean.
>
> **A8**: We would like to clarify that nearly all results in the paper are presented with standard deviation (std) across different seeds to ensure reliability. If there are specific results you believe are missing this detail, please let us know, and we will address them promptly.
>
> >**Q9**: Please state somewhere how exactly do you implement DP version of loss based clustering.
>
> **A9**: As noted in our response to the first question, we have updated line 17 of Algorithm 1 in the revised draft to eliminate any ambiguity. Loss-based clustering is performed "locally" on the clients’ side, with no loss values shared with the server. As outlined in lines 17 and 19 of the algorithm, the server broadcasts the updated cluster models to all clients, who select the model with the lowest loss. They then update it locally using DPSGD, returning all models (one of which is updated) back to the server. The server uses the modified model, and ensures that there are no privacy concerns when switching the clustering strategy.
>
>
> >**Q10**: All experiments: please also clearly report δ besides ϵ for all DP runs.
>
> **A10**: We have mentioned the value of  δ as being fixed to $10^{-4}$ in line 154 of the revised draft as well as appendix B.3.
> (please continue to the following)

---

> > ### Author Response · Authors · 2024-11-22
> > **Response to Reviewer EpDE**
> >
> > >**Q11**: Def 3.1: Please also state explicitly which neighborhood definition you use with DP (e.g. add/remove).
> >
> > **A11**: We have explicitly defined our considered neighborhood notion in line 133. Specifically, it is based on adding or removing a single sample to the dataset of one of the clients.
> >
> > >**Q12**: Lines 136-140: Do you actually use the classical Gaussian mechanism for something, if all accounting is done using RDP?
> >
> > **A12**: Yes, we use the classical Gaussian mechanism implemented by the DPSGD algorithm to train our models. Additionally, we employ the RDP accountant (TensorFlow Privacy implementation) to handle the heterogeneity before and after the first round, as it is well-suited for this purpose. This approach is based on the work by Mironov (2017) on Rènyi Differential Privacy (https://arxiv.org/pdf/1702.07476).
> >
> > >**Q13**:  Please fix typos.
> >
> > **A13**: Thank you for pointing out these typos. We have corrected them in the revised draft.
> >
> > >**Q14**: There is also contrary evidence that DP has in some sense only a limited impact on fairness (e.g. Berrada et al. 2023, Mangold et al. 2023)
> >
> > **A14**:  Our experimental results show that, "if mixed with the majority clusters", the minority cluster, with the smallest number of clients (e.g., cluster 0 in Fig. 2 right), experiences the lowest performance (i.e., the largest privacy cost) when DP is introduced, which aligns with previous observations in the cited works. We will review the referenced works in more detail and update the related work section of our camera-ready version to reflect them.

---

> > > ### Author Response · Authors · 2024-11-24
> > > **Proof for Privacy guarantees are added to Appendix H**
> > >
> > > Dear reviewer EpDE,
> > >
> > > We hope that you have found our responses to your questions clarifying and convincing. We just wanted to remind that we have added a formal proof for the privacy of the RC-DPFL algorithm to Appendix H, which fully addresses any concerns about the privacy guarantees of RC-DPFL by showing that it satisfies $(\epsilon, \delta)$-DP. This proof is completely aligned with how we have used Renyi DP accountant (from Tensorflow privacy) in our work. We hope that you will find it useful.
> > >
> > >
> > > Thank you,
> > >
> > > Authors

---

> > > > ### Author Response · Authors · 2024-11-25
> > > > **Official comment by authors**
> > > >
> > > > Dear Reviewer EpDE,
> > > >
> > > > Since we are approaching the end of the discussion period, we appreciate it if you could give us your feedback after reading the rebuttal. We hope you find our responses satisfactory and would appreciate it if you could reconsider your score.
> > > >
> > > > Thanks in advance,
> > > >
> > > > Authors

---

> > > > > ### Comment · Reviewer_EpDE · 2024-11-25
> > > > > **Some additional comments**
> > > > >
> > > > > Thanks for the rebuttal, some further comments:
> > > > >
> > > > > 1) On the claimed novelty of Lemma 4.1: I think it is absolutely *not okay* that you first claim a novel result, and when it is pointed out that the said result has already been published in a very similar form, you simply say that you actually were aware of the existing work but just didn't cite it previously. I would recommend rejecting the paper for this reason alone. For the next version, please also cite Räisä et al. 2024: as I already stated in the review, their results are very relevant for the discussion around your Lemma 4.1 on the batch size.
> > > > >
> > > > > 2) On the local loss-based clustering: note that if a client directly uses the best loss over the local training data to choose which model to use model, this choice can leak out information beyond what is allowed by DPSGD leading to invalid DP guarantees.
> > > > >
> > > > > 3) As a minor point, your neighborhood definition on lines 133-34 is now replace neighborhood, while you otherwise state to use add/remove. Please fix this.
> > > > >
> > > > > 4) I think the changes made to the paper already warrant a new round of reviews. I therefore recommend rejection and won't be updating my scores again in this discussion.

---

> ### Author Response · Authors · 2024-11-26
> **Response to Reviewer EpDE**
>
> Thanks again for the new question. We are happy to answer them as follows:
>
> >**Q 15**: On the claimed novelty of Lemma 4..
>
> **A 15**: We believe there may be a misunderstanding regarding our claims about Lemma 4.1. To clarify, we have **never presented Lemma 4.1 as a novelty of our work**. This is evident from both the initial submission and the revised draft, where the highlights of our contributions do not include Lemma 4.1.
>
> Instead, we extended the lemma from the referenced work to account for two distinct stages during training, to address a more nuanced scenario. Additionally, the privacy accountants differ between the two approaches: our Lemma 4.1 necessarily employs the RDP accountant [2] due to batch size heterogeneity between the first and subsequent rounds, whereas the referenced work uses the moments' accountant [3]. Hence, not only that we extend their result but also that we tailor it to our specific problem setting.
>
> We acknowledge that the referenced work was mistakenly omitted in our initial submission and we have included it in our revised version. That being said, the observation that larger batch sizes reduce noise variance is not novel to the community and is well-known in the differential privacy literature, as Reviewer CYaZ also noted. Our use of this result is foundational to our development of a novel clustering strategy, and we leveraged it appropriately as a stepping stone.
>
> It is also worth emphasizing that the core theoretical contributions of our work, specifically Lemma 4.2 and Theorem 4.3, are novel and directly address the challenges unique to our problem setting. These contributions go beyond foundational observations and add significant theoretical value to our work.
>
> Finally, we have cited the second work you referenced in the revised draft. We hope these explanations clarify the misunderstanding.
>
> [1]: S. Malekmohammadi, et. al., "Noise-Aware Algorithm for Heterogeneous Differentially Private Federated Learning", 2024.
>
> [2]: I. Mironov, et. al., "Re ́nyi Differential Privacy of the Sampled Gaussian Mechanism", 2019.
>
> [3]: M. Abadi, et. al., "Differentially Private Deep Learning", 2016.
>
> > **Q 16**: On the local loss-based clustering: note that if a client directly uses the best loss over the local training data, this choice can leak out information.
>
> **A 16**: We acknowledge that local loss-based clustering may reveal some additional information to the server; however, note that the potential information leakage in our method is substantially lower compared to approaches where clients directly share loss values or model updates with the server for two reasons. We explain further as follows:
>
> First, our approach computes the average train loss by aggregating "real-valued sample losses" over the entire dataset, as opposed to batch gradients that aggregate "high-dimensional sample gradients" over a relatively small batch of samples. In cross-silo FL settings, where each silo typically holds a large dataset, this significantly minimizes sensitivity to individual data points. As a result, the likelihood of a meaningful leakage from model selection (and not loss value sharing) by clients based on train loss value on their local data is really negligible.
>
> Second, note that this type of negligible leakage is not exclusive to our method. The baseline f-CDPFL performs the loss-based local clustering **from the very first round**. In contrast, our method operates exclusively on clients’ differentially private model updates for nearly half of the training time before incorporating loss-based clustering. Despite this more conservative approach, our method consistently and significantly outperforms the baselines. **Therefore, not only our approach does not incur additional privacy leakage, but also it diminishes it compared to the best baseline, which is the f-CDPFL**.
>
> >**Q 17**: As a minor point, your neighborhood definition on lines 133-34 is now replace neighborhood.
>
> **A 17**: We have explicitly clarified the neighborhood notion in Definition 3.1 and on lines 133–134, addressing the reviewers' concern.
>
> > **Q 18**:  I think the changes made to the paper already warrant a new round of reviews.
>
> **A 18**: We would like to clarify that the only modification made during the rebuttal period was the development of a method to determine the number of clusters when it is not provided. This directly addressed a specific concern raised by the reviewers. Importantly, this method has been shown to work effectively, and the current results accurately reflect the outcomes that would have been obtained had this method been applied from the outset. Therefore, all updates align strictly with the reviewers’ requests. Moreover, all the changes are fully transparent, and the updated results have been presented for the reviewers' thorough evaluation.
>
> We hope that you will find our results and responses convincing and would be grateful if they could raise your score.

---

> > ### Comment · Reviewer_EpDE · 2024-11-26
> > **Final (?) remarks**
> >
> > Thanks for the additional clarifications, I will still maintain my score:
> >
> > i) On Lemma 4.1: good to know that this was not as intentional as it looked; while it is not the main contribution of the paper, it definitely was presented as a novel contribution in the main body of the paper. However, since the citations are added for the next round, this is fine from now on.
> >
> > ii) On the changes: I meant the changes accumulated after fixing the loss clustering issue breaking DP guarantees; you could of course simply mitigate the claims to omit mentioning DP guarantees, but this still seems like a major change to me. The fact that existing papers might falsely claim their methods to be DP when they are is not a good reason for doing the same.

---

> ### Author Response · Authors · 2024-11-26
> **Response to Reviewer EpDE**
>
> Thank you for your comment. We kindly ask that you review the following explanations carefully (especially the second point (2)), and we hope you will find them convincing.
>
> 1) Firstly, there are no significant changes to the privacy guarantees in our paper, as clients do not share their loss values with the server during any of the $E$ training rounds, as it was the case before.
>
> 2) Second, while we understand your concern that clients are “selecting” models locally and sending this “selection” to the server, we firmly believe that this local model selection **has effectively zero “record-level” privacy leakage**. We draw your attention to the following for a further organized explanation:
>
> The sensitivity of a client’s "average train loss value" (which is not shared with the server) to the addition or removal of a sample in the client’s local dataset is relatively small, as this value is resulted from an “averaging over the whole dataset”. However, **even if the "average train loss value" of a client changes slightly as a result of addition or removal of a sample (a record-level change), the client’s selected model remains exactly the same**, especially in the third stage of our algorithm that one of the $M$ existing cluster models results in a smaller loss by a large margin. Therefore, **in the third stage, the sensitivity of the local model selection of a client to a single sample (record) change is effectively zero** (in contrast to the sensitivity of the average train loss, which was "small"). This is true especially for our algorithm, which transitions to a loss-based strategy after some training progress in the first $E_c$ rounds (as opposed to existing loss-based clustered FL approaches). Finally, **our claim of “effectively zero record-level privacy leakage of local model selections” is an immediate consequence of the “zero sensitivity to record-level changes” discussed above**. Please remember that the Differential Privacy definition considered in this work is concerned with "record-level" changes. When a client selects one model, the information that it reveals is which "data distribution" it belongs to. In fact, It is an information of population rather than an individual record (sample). However, the data distribution itself is not the target that record-level DP aims to protect.
>
> 3) Finally, and most importantly, even if the local model selection above is performed locally by clients, we can still correctly claim that “the set of model updates uploaded" to the server by each client provides ($\epsilon$, $\delta$)-DP w.r.t its local data as outlined in Appendix H. This claim is entirely correct and supported by our proof. The claim in the privacy theorem and its proof relies on the fact that the model updates are obtained from clients running DPSGD locally.
>
>
> If you have any concerns regarding our proof or the above reasonings, we would greatly appreciate it if you could point out any flaws in them. We trust that the explanations provided above clarify the privacy aspects of our algorithm and that you find them convincing.

---

### Official Review · Reviewer_wmqa · 2024-11-03

**Soundness:** 3
**Presentation:** 3
**Contribution:** 3
**Rating:** 8
**Confidence:** 3

**Summary:**

The paper studies the problem of accuracy disparities among clients in federated learning on their own local data distributions, especially with differential privacy. One approach to address this in the literature is to cluster the clients with similar model updates or training loss to share a model. The main contribution of this paper is to enable clustered federated learning with differential privacy. This is achieved by the authors' following observations: (a) in early rounds, model losses are not meaningful, (b) in early rounds, model updates depend too much on its initialization, but when initialization is fixed, the model updates are more stable if not using SGD but using full batch gradient descent (c) in late rounds, model updates are very small and hence not helpful. Guided by these observations, the paper proposes RC-DPFL, which uses fixed initialization, full batch learning in the first round, and for early rounds, GMM on model updates to cluster the clients (to give a probability distribution for each client's cluster), while clustering clients based on their training loss in later rounds. The authors prove that: (a) full batch training reduces uncertainty in the model updates/gradients such that the overlap between GMM component on the model updates is bounded (b) if the first rounds model updates are i.i.d. sampled from a mixture of Gaussian, the GMM has super-linear convergence rate. Empirical evaluation shows that RC-DPFL indeed finds the correct cluster of the clients and achieve high level of fairness in term of client's accuracy.

**Strengths:**

* Interesting and important problem: performance disparities among clients in FL
* Detailed and complete literature review on FL, FL personalization and fairness, clustered FL and DP
* Novel algorithm that combines different approaches and is theoretically guided
* Comprehensive experimental results on different types of data heterogeneity
* The paper is structured nicely and I enjoyed reading the paper

**Weaknesses:**

* It seems that Theorem 4.3 has a very strong assumption that in the first round, the model updates from all clients are i.i.d. and sampled from a Gaussian mixture, and the theorem argues that under such assumption, the GMM can converge faster with larger batch size. May I know if this assumption on model updates are well justified? Is it a standard assumption or is it in fact likely true in practical settings?
* Some settings of the experiments are not clearly described. For example, how to choose the time to switch from update-based GMM soft clustering to loss-based hard clustering? How does the server known and set the hyperparameters of the Gaussian mixture model, such as the number of clusters? I would like to see that how different choices of these parameters (i.e., switching round and number of clusters) will affect the results.
* Some minor typos and grammatical errors. For example, in line 187--188, should it be "the larger the batch size, the smaller the variance"?

**Questions:**

I raised a few questions in the weakness section when I list my concerns. I will appreciate it if the authors could address those questions.

---

> ### Author Response · Authors · 2024-11-22
> **Response to Reviewer wmqa**
>
> Thank you for your positive feedback and we really appreciate it. We have modified the draft according to your suggestions. In the following, we answer your questions:
>
> >**Q1**: It seems that Theorem 4.3 has a very strong assumption.
>
> **A1**: We believe there has been a misunderstanding regarding the assumption of "i.i.d." in the lemma. We would like to clarify that there was no intended emphasis on the term "i.i.d." in our original formulation, and we have removed it in the revised version of the paper to avoid any potential confusion.
>
> What we actually meant by the term was that "there is an underlying true cluster structure from which the model updates originate." For instance, in Figure 2 (right), we illustrate four clusters with completely "non-iid" data distributions. However, the set of all model updates can still be considered "i.i.d." samples from the underlying mixture of Gaussians, as shown in the figure. In fact, the latent variable in the Gaussian Mixture Model determines the cluster each model update is drawn from.
>
> Therefore, the use of "i.i.d." in the lemma does not conflict with the "non-iid" data distributions across clusters. To eliminate any ambiguity and ensure clarity, we have removed the term in the revised draft. We hope this resolves any concerns regarding the assumption.
>
> >**Q2**: Some settings of the experiments are not clearly described. For example, how to choose the time to switch.
>
> **A2**: We appreciate the reviewer’s feedback and would like to address the concerns regarding the setting of $E_c$. Note that we have extensively revised the draft to provide a clearer and more detailed explanation of how to set $E_c$, which is highlighted in blue throughout the document (line 205, line 11 of Algorithm 1, Section 4.2.2, and the second paragraph in Section 4.4). To clarify, $E_c$ corresponds to the duration of time we continue using the learned GMM, and it should be determined based on the "confidence" of the GMM in its clusterings. Intuitively, if the GMM clustering at the end of the first round is highly confident (e.g., 100% correct), then $E_c$ should be large, which allows us to continue using the GMM for a longer period. Conversely, if the GMM is uncertain about its clustering, $E_c$ should be smaller to avoid relying on soft clusterings for too long. In Section 4.2.2, we provide a method for measuring the "confidence" of the learned GMM, which is used to adjust $E_c$ accordingly. When the GMM is highly confident, we continue using it until the end of the first half of the training period before switching to loss-based clustering. As the GMM’s confidence decreases, $E_c$ decreases as well.
>
> Additionally, we provide detailed guidance on setting the GMM’s parameters in Appendix B.5 and F.2. The three main parameters are: 1) covariance type ("spherical"), 2) initialization method ("KM++"), and 3) the number of components "M," which corresponds to the true number of clusters. In the revised draft, we have added a new section which describes how to set "M" in a manner that maximizes the GMM's confidence, which significantly improves the applicability of our approach in real world scenarios.
>
>
> >**Q3**: Some minor typos and grammatical errors.
>
> **A3**: Thank you for pointing out this typo, we have corrected it in the revised draft.

---

> > ### Author Response · Authors · 2024-11-25
> > **Official comment by authors**
> >
> > Dear Reviewer wmqa,
> >
> > Since we are approaching the end of the discussion period, we appreciate it if you could give us your feedback after reading the rebuttal. We hope you find our responses satisfactory and would appreciate it if you could raise your score.
> >
> > Thanks in advance,
> >
> > Authors

---

> > ### Comment · Reviewer_wmqa · 2024-11-25
> >
> > Thanks for addressing my concerns. The authors' response has cleared some of my previous questions. I've raised my score accordingly.

---

> ### Author Response · Authors · 2024-11-25
> **Response to reviewer wmqa**
>
> We are happy to see that our responses have addressed your concerns. We also thank you for increasing the score.
>
> Thank you,
>
> Authors

---

### Official Review · Reviewer_q6m1 · 2024-11-08

**Soundness:** 1
**Presentation:** 2
**Contribution:** 2
**Rating:** 3
**Confidence:** 4

**Summary:**

This work propose a new differentially private clustered federated learning method based on training a GMM. The authors conduct convergence analysis on the proposed method and evaluated the method on multiple real world datasets.

**Strengths:**

- The problem of private clustered FL is important to study.
- The method comes with some theoretical guarantees.

**Weaknesses:**

- The author claim the proposed method is private, but it is unclear what the level of privacy is enforced and what the threat model is in this work. It looks like the privacy notion being used here is example-level privacy since DP-SGD is used and the goal is to protect the server / other clients within the same cluster from inferring information of example? If that is the case, shouldn't the clustering process itself also be privatized?
- There's no theoretical results of the DP bound for the proposed method. Further, due to lack of formal threat model, it's unclear what privacy guarantee the proposed method provides.
- 4.3.1 seems problematic. Since the data augmentation is done by transformation of some data $x$, that means if you change $x$, all the augmented data also change. Therefore, in this scenario, there wouldn't be the case where the neighboring dataset only differs in one data point. And due to the way the averaging is done (average over all examples within the batch rather than first average over all augmented data for one example and then do a second level averaging), what I see is for every step of noisy stochastic gradient update, we are having a group privacy guarantee here rather than standard DP guarantee. Therefore, adding the same amount of noise would not give you better privacy with the presence of augmented data. Further, saying per instance DP is also misleading since the proposed method seems to only focus on standard DP notion rather than any per-instance DP guarantee.
- L433: "define client-level fairness as the equality of “privacy cost" across clients" is highly misleading. As I mentioned earlier, it is unclear what the privacy guarantee is here. However, fair prediction across clients does not equate to any privacy protection at either example level or client level.
- Empirical results only show results for $\epsilon=5$. I did not see where $\delta$ is defined. Also could the authors show the pareto frontier for privacy-utility / privacy-fairness tradeoff?
- The claim that "We propose the first DP clustered FL algorithm" should be more careful as there are several private federated clustering work, e.g. [1,2]

[1] Yiwei Li, Shuai Wang, Chong-Yung Chi, and Tony QS Quek. Differentially private federated clustering over non-iid data. IEEE IoTJ, 2023.

[2] Guixun Luo, Naiyue Chen, Jiahuan He, Bingwei Jin, Zhiyuan Zhang, and Yidong Li. Privacy-preserving clustering federated learning for non-iid data. FGCS, 2024.

**Questions:**

- How does the author handle scenarios where some clusters consists only limited amount of clients (e.g. 1/2), in which case given same privacy, utility could be drastically compromised?

---

> ### Author Response · Authors · 2024-11-22
> **Response to Reviewer q6m1**
>
> Thank you for taking the time to give multiple comments about our work. They were useful for improving our work. We have addressed them all. Changed/added parts are highlighted in blue. We are glad to answer your questions as follows:
>
> >**Q1:** The authors claim the proposed method is private, but it is unclear what the level of privacy is enforced.
>
> **A1:** To address the reviewer's concern regarding privacy, we would like to clarify that our paper explicitly defines the threat model in Figure 1 (left), with additional explanation provided on line 131. As shown in Figure 1, clients do not trust any party outside their local trust boundary. They run DPSGD locally during training to ensure their data is protected against both the server and any third parties using the system’s final trained model. This results in record-level differential privacy (DP) guarantees, which are clearly stated in the paper (definition 3.1 and line 133).
>
> Furthermore, we would like to emphasize that the clustering algorithm maintains privacy. In the first round, the server performs clustering on the clients' model updates, which are already noisy and private due to the application of DPSGD. According to the "post-processing property" of differential privacy, any operations the server performs on these noisy updates, such as running GMM clustering, do not compromise the privacy guarantees.
>
> In subsequent rounds, after $E_c$, the clustering method switches to a loss-based strategy and is performed "locally" by each client. Since this is a local process, privacy is again preserved. For further details, please refer to lines 17 and 19 in Algorithm 1.
>
> >**Q2**: There's no theoretical results of the DP bound for the proposed method.
>
> **A2**: We appreciate the reviewer’s comment and would like to clarify that, following the threat model outlined in Fig. 1 (left), clients run the DPSGD algorithm locally with a predetermined noise scale, which is derived from the Renyi DP accountant. The noise scale is specifically set to achieve the desired $(\epsilon, \delta)$-DP guarantee, as described in equation (2) and explained on line 152.
>
> As a result, upon completion of training, all clients are guaranteed $(\epsilon, \delta)$-DP privacy against the untrusted server and any other third parties in the system. The proof of this privacy guarantee is directly derived from the fact that clients are running the DPSGD algorithm, which ensures the stated privacy protection.
>
>
>
> >**Q3**: 4.3.1 seems problematic. Since the data augmentation is done by transformation of some data.
>
> **A3**: We agree with the reviewer’s comment that the way we initially discussed data augmentation could lead to privacy leakage. While "batch augmentation" [1] preserves privacy guarantees, we were aware of its limitations for our specific problem, which requires a *large batch size*.
>
> Considering this, we have removed the paragraph discussing the "potential" use of data augmentation in the revised draft. Furthermore, we want to clarify that our current experimental results do not use any form of data augmentation. Instead, we used a full batch size equal to the actual dataset size, which has been effective, and the results reported in the submission reflect this approach.
>
> [1]: E. Hoffer, et. al.,”Augment Your Batch: Improving Generalization Through Instance Repetition”, CVPR 2020.
>
>
> >**Q4**: "define client-level fairness as the equality of “privacy cost" across clients" is highly misleading.
>
> **A4**: As outlined in our previous responses, we have clearly defined the threat model and explained that the privacy guarantees in our framework are "sample/record-level" DP guarantees, achieved by running DPSGD locally on the clients' sides.
>
> It is well-established that introducing differential privacy (DP) into a system can lead to disparate impacts on the performance of different groups. Specifically, minority groups tend to experience a larger performance drop compared to the non-DP training scenario. We refer to this performance degradation as the "privacy cost." If a cluster contains fewer clients (e.g., cluster 0 in Figure 2, right), and this cluster is not "properly detected and separated from the majority clusters", it will be expected to experience a larger privacy cost after the introduction of DP.
>
> Our key contribution is demonstrating that our client clustering algorithm is robust to DP noise and effectively detects these minority clusters. As a result, our algorithm ensures that different clusters experience nearly equal privacy costs (performance drops), and maintain fairness across groups compared to their performance in non-DP training. Our experimental results validate this claim (see the results corresponding to the row “F_acc” in tables 5, 7, 9 and 11 in the appendix).
>
> In contrast, we show that existing clustering algorithms fail to handle DP noise effectively, which leads to disparate performance drops across clusters (same row “F_acc” for other algorithms)

---

> > ### Author Response · Authors · 2024-11-22
> > **Response to Reviewer q6m1**
> >
> > >**Q5**: Empirical results only show results for  ϵ=5. I did not see where δ is defined. Also could the authors show the pareto frontier for privacy-utility / privacy-fairness tradeoff?
> >
> > **A5**: In our experiments, we have fixed δ to $10^{-4}$, as explicitly mentioned in line 154 of the paper. Additionally, to address concerns regarding utility and fairness, we have included new plots to show the Pareto frontier for these metrics at $\epsilon=4$ across the four different datasets in Figure 12 of the appendix. We observe a significant improvement in the trade-offs between utility and fairness in our approach compared to DPFedAvg, KM-CDPFL and f-CDPFL by achieving the highest utility and fairness in almost all scenarios. This achievement is enabled by the correct client clustering of our algorithm, which avoids different clients with heterogeneous data getting mixed into one cluster.
> >
> >
> > >**Q6**: The claim that "We propose the first DP clustered FL algorithm" should be more careful.
> >
> > **A6**: We would like to clarify the differences between our work and the cited literature by the reviewer.
> >
> > The first work referenced is about "Federated Clustering," which is not the same as "clustered federated learning." Specifically, the goal in that work is to learn the underlying cluster structure of a set of distributed unlabeled points without centralizing them, whereas in our framework, the server has access to all the model updates from the clients that need to be clustered. This key distinction differentiates our approach from the clustering of distributed unlabeled data.
> >
> > The second work is on clustered federated learning, but it is important to note that the method proposed is highly limited. It clusters clients based on labels that they do not have in their local data, which is unrealistic in most FL scenarios. In typical FL settings, we assume that all clients have access to all possible labels, which makes this method inapplicable to our context. Nonetheless, we have added this work to our "Related Works" section (line 102) for completeness. To the best of our knowledge, our work is the first to study clustered Differentially Private Federated Learning (DPFL) in general scenarios with various data distributions.
> >
> > >**Q7:** How does the author handle scenarios where some clusters consist only limited amount of clients (e.g. 1/2), in which case given same privacy, utility could be drastically compromised?
> >
> >
> > **A7**: We appreciate the reviewer’s comment and would like to emphasize that this question indeed is a key issue addressed in our work. Specifically, when there is a cluster with only a few clients, such as cluster 0 with three clients in Figure 2 (right), we refer to it as a "minority cluster." The primary challenge for these minority clusters, which have distinct data distributions, is to correctly detect and cluster the clients belonging to them. If this step is not done properly, these minority clients will be mixed with clients from larger, majority clusters, and lead to poor performance due to the well-known issue of DP disproportionately affecting the performance of minority data (as pointed by the reviewer).
> >
> > To mitigate this, the first task is accurately "detecting and clustering these minority clients". The main obstacle to this process is the DP noise in the system. While existing clustered FL algorithms are vulnerable to this DP noise and struggle to effectively cluster clients, our algorithm has been specifically designed to be robust to such noise. As demonstrated in our experimental results (figures 4.C and 6.C), our approach can cluster minority clients with robustness to the DP noise, thereby we ensure significantly better performance for them.

---

> ### Author Response · Authors · 2024-11-24
> **Proof for Privacy guarantees are added to Appendix H**
>
> Dear reviewer q6m1,
>
> We hope that you have found our responses to your questions clarifying and convincing. We just wanted to remind that, following  your Q2 above, we have added a formal proof for the privacy of the RC-DPFL algorithm to Appendix H, which fully addresses another one of your questions by showing that RC-DPFL satisfies ($\epsilon$, $\delta$)-DP guarantees. This proof is completely aligned with how we have used Renyi DP accountant (from Tensorflow privacy) in our work. We hope that you will find it useful.
>
>
> Thank you,
>
> Authors

---

> ### Author Response · Authors · 2024-11-25
> **Official comment by authors**
>
> Dear Reviewer q6m1,
>
> Since we are approaching the end of the discussion period, we appreciate it if you could give us your feedback after reading the rebuttal. We hope you find our responses satisfactory and would appreciate it if you could reconsider your score.
>
> Thanks in advance,
>
> Authors

---

> > ### Comment · Reviewer_q6m1 · 2024-11-25
> > **Reply**
> >
> > Thank you for your detailed response.
> > - It's clearer to me that the authors are protecting example-level privacy here. Instead of only using definition 1, I recommend the authors to directly specify that they are looking at example-level privacy, by adding a definition similar to Def 2 in Lowy et al. [1] or Def 3.1 in Liu et al. [2]. The same applies to Theorem H.1, without clearly stating privacy notion, it would be confusing for people to understand what level of privacy you are protecting, especially with the presence of clustering.
> > - I see that the authors refer performance drop after DP training for the minority group as privacy cost. Still, I think such reference is confusing as I would understand the term privacy cost as the privacy parameter $\epsilon$ and there exists a privacy utility tradeoff. I recommend the authors to fix the wording to avoid confusion.
> > - What is the number of examples within each client? Did you ensure it is larger than $1/\delta$? Otherwise the privacy guarantee is not that useful.
> > - Correct me if I'm wrong, it seems that the method does not have any hyperparameter to tune and therefore the pareto frontier is just a point? What I think will improve the paper is a curve whose points are dependent on the choice of hyperparamaters. Maybe that's not applicable?
> > - The authors claim they use clipping bound = 3 since that's the best reported in Abadi et al. It does not seem the model used in this work is the same as the one in Abadi et al. One drawback would be the missing of search over the clipping bound.
> > - It looks like Liu et al [2] has a DP clustered federated learning method. Could the authors briefly explain the difference and the advantage of the proposed method?
> >
> >
> >
> > [1] Lowy, A., Ghafelebashi, A., & Razaviyayn, M. (2023, April). Private non-convex federated learning without a trusted server. In International Conference on Artificial Intelligence and Statistics (pp. 5749-5786). PMLR.
> >
> > [2] Liu, K., Hu, S., Wu, S. Z., & Smith, V. (2022). On privacy and personalization in cross-silo federated learning. Advances in neural information processing systems, 35, 5925-5940.

---

> ### Author Response · Authors · 2024-11-26
> **Response to Reviewer q6m1**
>
> Thanks again for the time you have spent for the new questions/comments. We are happy to answer your questions, as follows.
>
> >**Q8:** I recommend the authors to directly specify that they are looking at example-level privacy.
>
> **A8:** We appreciate the reviewer’s recommendation. In response, we have revised lines 132 to 134 to explicitly clarify that the (ϵ,δ)-DP privacy guarantee applies individually to each client and is considered at the record level.
>
> >**Q9 :** I think such reference is confusing as I would understand the term privacy cost as the privacy parameter.
>
> **A9**: To address potential ambiguity, we have replaced the term “privacy cost” with “performance drop” (line 439) to more precisely convey our intent. Specifically, when discussing fairness, we are referring to ensuring that the change in performance experienced by different clients is almost the same.
>
> >**Q 10:** Did you ensure that dataset size is smaller than $1/\delta$?
>
> **A 10:** Note that dataset size should be smaller than $1/\delta$, or equivalently, $N_i \times \delta<1$. The number of data points in the clients' datasets varies slightly (all around 6000); however, for all datasets and experiments, the dataset sizes are consistently smaller than $1/\delta=10000$. We have explicitly accounted for  this point in our experiments.
>
> >**Q 11:** It seems that the method does not have any hyperparameter to tune.
>
> **A 11:** One of the parameters that needs to be set before training is the batch size used in rounds $e>1$. We have discussed in Appendix F.1 and Figure 10 that the batch size used during the rounds $e>1$ needs to be set to small values, and that the algorithm exhibits low sensitivity to this parameter as long as it is assigned small values. Despite this, to further strengthen our claims and fully address your feedback, we are conducting an ablation study across a wide range of values, which include larger batch sizes. We think such an ablation study will help us to conclusively show that setting this parameter to large values is not effective. The new results will help identify a clear "Pareto frontier" as requested and provide a principled understanding of the trade-offs involved. Having provided the results mentioned above to address your concerns/questions, we are conducting these new and extensive experiments to make sure that this important aspect is thoroughly addressed.
>
> > **Q 12:**  One drawback would be the missing of search over the clipping bound.
>
> **A 12**: We searched for the parameter c for values around 3 ({2,3,4}) using a validation set. However, we found that the variation across these values was negligible. Consequently, we fixed c to 3, as this choice did not significantly impact performance. A similar observation is reported in Figure 5.5 of the referenced work [1], which also demonstrates minimal variation around the optimal choice. To ensure consistency and fair comparisons across **all experiments and baselines**, we retained c=3 for all scenarios. We have included these details in Appendix B.3.
>
> [1]: Abadi, et. al., "Differentially private deep learning".

---

> ### Author Response · Authors · 2024-11-26
> **Response to Reviewer q6m1**
>
> > **Q 13**:  It looks like Liu et al [2] has a DP clustered federated learning method.
>
> **A 13:** We believe our work complements the referenced paper and strengthens its findings. We further explain in detail as follows:
>
> As stated in the second paragraph of the introduction of the work, cross-silo FL is identified as an FL setting, where there are few clients but each holds many data samples (i.e., large local dataset sizes) that require protection. The introduced work has two main messages regarding such FL settings:
>
> > 1) **When there is data heterogeneity but clients do not run DPSGD locally (i.e. non-private FL):** in this case, if we learn one single model for all clients, it may suffer from data heterogeneity across clients. As such, when data heterogeneity is high, local training, i.e., each client learning one model on its local data, may outperform learning a single global model by using FedAvg (Fig. 2, left). Remember that the important underlying assumption is that in cross-silo FL, clients hold “large local datasets”. Hence, the above reasonable statement by the authors makes sense.
>
> > 2) **When there is data heterogeneity and clients run DPSGD (i.e. they are privacy-sensitive):** in this case, the situation may be different and federation is encouraged, as a result of noise variance reduction happening in FL (Fig. 2, middle and right). This is especially true when privacy parameter $\epsilon$ is not large enough. Note that when clients have privacy sensitivity (i.e., private training), no matter if clients decide to perform local training or participate in FL, they run DPSGD locally to protect their privacy.
>
> Based on the above two messages, local training (no FL) and FedAvg (full FL participation learning one global model) can be viewed as two ends of a personalization spectrum with identical privacy costs (because in either case clients run DPSGD locally). Therefore, the authors propose that some level of model personalization should work better than both the global and local model learning in terms of the model utility. And with their results they show that this is the case for mildly heterogeneous data splits. However, there are two scenarios where the proposed personalization method does not work (and it is exactly when our proposed method complements their work):
>
> When the clients are highly privacy sensitive, or epsilon is small, the proposed method does not work (as shown in Fig. 3).
> Also, when data heterogeneity increases even further (similar to the settings in our work that some clusters of clients exist), the data heterogeneity across clusters is to the extent that personalization techniques no longer work (we indeed mentioned this point in our answers to one of the other reviewers too that model personalization does not work for highly heterogeneous scenarios). In this case, clustered FL algorithms perform better than the proposed model personalization method. More specifically, see Figure 6 left and compare the results for IFCA (^) and MR-MTL (>), which is the authors' proposed method. (Interestingly, they have used exactly the same rotation data splits across clusters with exactly the same 4 clusters that we have in our work). As can be observed, **under high data heterogeneity**:
>
> > 1) When epsilon is not small, the two methods IFCA (which is the same as our f-CDPFL) and MR-MTL perform very closely.
> > 2) However, when epsilon is small, IFCA, which is a stronger personalization method, outperforms their proposed personalization approach.
>
> Having explained these shortcomings, in our work, we showed that our RC-DPFL algorithm outperforms even the IFCA loss-based clustering method when there is a **high data heterogeneity** and **clients are privacy sensitive** (epsilon is small), i.e., when the authors’ proposed personalization method was outperformed by IFCA. In this way, our proposed method complements the two methods in highly heterogeneous and privacy sensitive scenarios, which has been the main goal of our work from the beginning. We have added the mentioned paper to the related works of our paper in the revised draft.
>
> We hope these responses address the reviewer’s concerns and would be grateful if they could raise their score.

---

> > ### Author Response · Authors · 2024-11-26
> > **Official comment by authors**
> >
> > Dear reviewer q6m1,
> >
> > We hope that you have found our responses to your questions clarifying and convincing. We just wanted to remind that, we have added lines 416 to 419 to the draft as well a new paragraph to Appendix H regarding the privacy of the third stage of our algorithm. We are looking forward to receiving your other comments/questions. If you have found our answers convincing, we would greatly appreciate it if you reconsider your score.
> >
> >
> > Thank you in advance,
> >
> > Authors

---

> > > ### Author Response · Authors · 2024-12-01
> > > **Official comment by authors**
> > >
> > > Dear Reviewer q6m1,
> > >
> > > Since we are approaching the end of the discussion period, we appreciate it if you could give us your feedback after reading the rebuttal. We hope you have found our responses satisfactory.
> > >
> > > Thanks in advance,
> > >
> > > Authors

---

### Author Response · Authors · 2024-11-22
**Paper revision submitted**

Dear reviewers,

Thank you all for your constructive and valuable feedback. We have improved our work, adapted your suggestions and made relevant changes in the revised draft. For the ease of access, we have coloured the changes in blue. The highlights of our modifications are as follows:

* We have improved the text and images for more clarity of different parts, including the considered threat model, algorithm 1, experimental set up and parameter selection.

* We have extensively studied the effect of the batch size $b_i^{>1}$, used during the rounds $e>1$, in order to make it clear why this batch size needs to be set to a small value. We have also compared its effect with that of the initial batch size $b_i^1$ for a deeper understanding of their different effects.

* We have developed a new method to address the scenarios where the true number of clusters (M) is not known. The proposed method works efficiently without enforcing much computational overhead to the server and without infringing clients' privacy. We believe that this proposed method addresses one of the main concerns of the reviewers.

* We have clarified about the method used for setting the switching time $E_c$, which is related to the confidence of the learned GMM in the first round. The used method provides a very intuitive way for setting the parameter.

* Finally, we have added a formal proof for the privacy guarantees of RC-DPFL showing that despite the fact that the batch size used in the first round is different from the next rounds, we can still account for this heterogeneity by using the RDP accountant. This accountant is able to provide the privacy guarantees for the composition of "heterogeneous" private mechanisms. The provided proof fully addresses the questions about the privacy guarantees of the proposed method.


Altogether, these clarifications provide a better understanding of our proposed algorithm and its applicability. Having addressed your main questions and concerns, we are looking forward to receiving your additional questions/comments during the remaining time.

Thank you,
Authors

---

### Meta-Review · Area_Chair_saTN · 2024-12-20

**Metareview:**

This paper proposes a robust differentially private clustered federated learning (DPFL) algorithm that aims to reduce performance disparities that appear when federated learning is applied to heterogeneous clients. The proposed DPFL algorithm clusters clients based on both their model updates and training loss values and introduces noise-reducing techniques to promote robustness. The authors provide a theoretical analysis of their scheme and empirical evaluations across different datasets.

The reviewers appreciated the effort to increase utility in federated learning. Nevertheless, several limitations were noted by the reviewers, and these concerns remained post-rebuttal. For instance, reviewer EpDE highlighted that the clustering step based on local loss may lead to privacy leakage that was not accounted for in the privacy analysis. They also raised important concerns regarding the novelty of the contributions (e.g., Lemma 4.1). Issues surrounding novelty were also noted by reviewer q6m1, who stood by these points post-rebuttal. Reviewer CYaZ also raised several questions regarding presentation and organization, though their initial concerns were somewhat assuaged post-rebuttal.

After my own reading of the paper, I recommend rejection. The presentation of the paper needs to be improved (clearly noted by the reviewers), and its positioning relative to prior work must be made more precise before acceptance. The reviewers provided valuable feedback, which I hope the authors take seriously when revising the manuscript.

**Additional Comments On Reviewer Discussion:**

The authors addressed most of the reviewers comments during the discussion phase and updated their manuscript. However, negative reviewers (EpDE an dq6m1) stood by their assessment to reject the manuscript even after engaging with the authors.

---

### Decision · Program_Chairs · 2025-01-22

Reject